

# New heterodont odontocetes from the Oligocene Pysht Formation in Washington State, U.S.A., and a reevaluation of Simocetidae (Cetacea, Odontoceti)

Jorge Velez-Juarbe

Department of Mammalogy, Natural History Museum of Los Angeles County, Los Angeles, CA, USA

Corresponding author
Jorge Velez-Juarbe,
jvelezjuar@nhm.org

## ABSTRACT

Odontocetes first appeared in the fossil record by the early Oligocene, and their early evolutionary history can provide clues as to how some of their unique adaptations, such as echolocation, evolved. Here, three new specimens from the early to late Oligocene Pysht Formation are described further increasing our understanding of the richness and diversity of early odontocetes, particularly for the North Pacific. Phylogenetic analysis shows that the new specimens are part of a more inclusive, redefined Simocetidae, which now includes *Simocetus rayi*, *Olympicetus* sp. 1, *Olympicetus avitus*, *O. thalassodon* sp. nov., and a large unnamed taxon (Simocetidae gen. et sp. A), all part of a North Pacific clade that represents one of the earliest diverging groups of odontocetes. Amongst these, *Olympicetus thalassodon* sp. nov. represents one of the best known simocetids, offering new information on the cranial and dental morphology of early odontocetes. Furthermore, the inclusion of CCNHM 1000, here considered to represent a neonate of *Olympicetus* sp., as part of the Simocetidae, suggests that members of this group may not have had the capability of ultrasonic hearing, at least during their early ontogenetic stages. Based on the new specimens, the dentition of simocetids is interpreted as being plesiomorphic, with a tooth count more akin to that of basilosaurids and early toothed mysticetes, while other features of the skull and hyoid suggest various forms of prey acquisition, including raptorial or combined feeding in *Olympicetus* spp., and suction feeding in *Simocetus*. Finally, body size estimates show that small to moderately large taxa are present in Simocetidae, with the largest taxon represented by Simocetidae gen. et sp. A with an estimated body length of 3 m, which places it as the largest known simocetid, and amongst the largest Oligocene odontocetes. The new specimens described here add to a growing list of Oligocene marine tetrapods from the North Pacific, further promoting faunistic comparisons across other contemporaneous and younger assemblages, that will allow for an improved understanding of the evolution of marine faunas in the region.

# INTRODUCTION

The Eastern North Pacific Region is recognized as one of the most prolific sources for early marine mammals belonging to various groups, particularly desmostylians, pinnipeds, and early mysticetes (*Emlong, 1966*; *Russel, 1968*; *Domning, Ray & McKenna, 1986*; *Berta, 1991*; *Ray, Domning & McKenna, 1994*; *Barnes et al., 1995*; *Barnes & Goedert, 2001*; *Beatty, 2006*; *Beatty & Cockburn, 2015*; *Marx, Tsai & Fordyce, 2015*; *Marx et al., 2016b*; *Peredo & Uhen, 2016*; *Peredo & Pyenson, 2018*; *Peredo et al., 2018*; *Poust & Boessenecker, 2018*; *Shipps, Peredo & Pyenson, 2019*; *Solis-Añorve, Gozález-Barba & Hernández-Rivera, 2019*; *Hernández Cisneros, 2018*; *Hernández Cisneros, 2022*; *Hernández Cisneros & Nava-Sánchez, 2022*; *Everett, Deméré & Wyss, 2023*). However, while odontocetes have also been found in these Oligocene-age units, and have been remarked in the literature in non-taxonomic context (*e.g.*, *Whitmore Jr & Sanders, 1977*; *Goedert, Squires & Barnes, 1995*; *Barnes, 1998*; *Barnes, Goedert & Furusawa, 2001*; *Kiel, Kahl & Goedert, 2013*; *Hernández Cisneros, González Barba & Fordyce, 2017*, only a handful are described (*Fordyce, 2002*; *Boersma & Pyenson, 2016*; *Vélez-Juarbe, 2017*). These include *Simocetus rayi Fordyce, 2002*, from the early Oligocene Alsea Formation, in Oregon, USA, the platanistoid *Arktocara yakataga Boersma & Pyenson, 2016*, from the late Oligocene Poul Creek Fm., in Alaska, USA, and the more recently described, *Olympicetus avitus Vélez-Juarbe, 2017*, from the early to late Oligocene Oligocene Pysht Fm., in Washington State, USA. The presence of stem (*i.e., Simocetus, Olympicetus*) and crown (*Arktocara*) odontocetes in similar-aged rocks point to a complex early history for odontocetes in this region, hence the description of new material will advance our current understanding of odontocete evolution.

In this work three additional specimens of stem odontocetes collected from the early to late Oligocene Pysht Formation of Washington State are described. The morphology of these new specimens shows similarities with *Simocetus* and *Olympicetus* and provides further insight into the diversity of early odontocetes in the North Pacific. In addition, cranial and dental features of simocetids hint at different modes of prey acquisition within members of the clade, with some taxa using suction feeding, while others being raptorial or combined feeders. The Pysht Fm. has a rich fossil record of marine tetrapods, including plotopterids (*Olson, 1980*; *Dyke, Wang & Habib, 2011*; *Mayr & Goedert, 2016*), desmostylians (*Domning, Ray & McKenna, 1986*; *Ray, Domning & McKenna, 1994*), aetiocetids (*Barnes et al., 1995*; *Shipps, Peredo & Pyenson, 2019*), stem mysticetes (*Peredo & Uhen, 2016*), pinnipeds (*Everett, Deméré & Wyss, 2023*) and many others still remaining to be described (*Whitmore Jr & Sanders, 1977*; *Hunt Jr & Barnes, 1994*; *Barnes, Goedert & Furusawa, 2001*; *Marx et al., 2016b*). The fossils described in this work demonstrate that stem odontocetes were more diverse in the North Pacific Region during the Oligocene and hint at the presence of clade of stem odontocetes that were geographically confined to this region in a pattern that parallels aetiocetid mysticetes (*Hernández Cisneros & Vélez-Juarbe, 2021*).

## MATERIALS & METHODS

### Phylogenetic analysis

The phylogenetic analysis was performed using the morphological matrix of *Albright III, Sanders & Geisler (2018)* as modified recently by *Boessenecker et al. (2020)*, with modification of two characters and addition of four new ones (see Files S1–S2). Characters 328 and 329 are modified to be specific to the upper molars, while new characters 330 and 331 are related to the number of denticles on the mesial and distal edges, respectively, on the main lower molars. The third new character (c.337) refers to the presence of a transverse cleft on the apex of the zygomatic process of the squamosal (first noted by (*Racicot et al., 2019*), for CCNHM 1000). The fourth new character (c.338) relates to the morphology of the thyrohyoid/thyrohyal, adding up to a total of 338 characters (see Files S1–S2). Besides LACM 124104, LACM 124105 and LACM 158720, one additional odontocete from the Pysht Fm. was added, CCNHM 1000 (collected from the same locality as the specimens described here), based on the description from Racicot et al. (*2019*: S1). All otherwise undescribed specimens in earlier versions of this matrix were removed from this analysis because their character states cannot be independently corroborated, resulting in a total of three outgroup and 107 ingroup taxa. The matrix was analyzed using PAUP* (version 4.0a169; *Swofford, 2003*); all characters were treated as unordered and with equal weights. A heuristic search of 10,000 replicates was performed using the tree bisection-reconnection (TBR) algorithm and using a backbone constraint based on the phylogenetic tree of extant cetaceans from *McGowen et al. (2020)*; bootstrap values were obtained by performing 10,000 replicates. The terminology used for the descriptions follows *Mead & Fordyce (2009)*.

### Taxonomy

The electronic version of this article in portable document format will represent a published work according to the International Commission on Zoological Nomenclature (ICZN), and hence the new names contained in the electronic version are effectively published under that Code from the electronic edition alone. This published work and the nomenclatural acts it contains have been registered in ZooBank, the online registration system for the ICZN. The ZooBank LSIDs (Life Science Identifiers) can be resolved and the associated information viewed through any standard web browser by appending the LSID to the prefix http://zoobank.org/. The LSID for this publication is LSIDurn:lsid:zoobank.org:pub:D190F6B6-FB67-4F2B-AC24-145DF06D3FD3. The online version of this work is archived and available from the following digital repositories: PeerJ, PubMed Central, and CLOCKSS.

## Systematic Paleontology

CETACEA *Brisson, 1762*
ODONTOCETI *Flower, 1867*
SIMOCETIDAE *Fordyce, 2002*

**Type Genus**—*Simocetus Fordyce, 2002*.
**Included Genera**—*Simocetus*; *Olympicetus Vélez-Juarbe, 2017*; Simocetidae gen. et sp. A.
**Temporal and Geographic Range**—early–late Oligocene (Rupelian–early Chattian) of the eastern North Pacific.
**Emended Diagnosis**—Stem odontocetes displaying a mosaic of plesiomorphic and derived characters that sets them apart from other basal odontocetes, particularly the Xenorophidae, Patriocetidae and Agorophiidae. Characterized by the following unambiguous synapomorphies: seven to eight teeth completely enclosed by the maxilla (c.25[1]); lack of a rostral basin (c.66[0]), differing from most xenorophids which have a well-defined basin; posteriormost edge of nasals in line with the anterior half of the supraorbital processes (c.123[1]); supraoccipital at about the same level as the nasals (c.129[1]), differing from xenorophids where the supraoccipital is higher; floor of squamosal fossa thickens posteriorly (c.149[1]); distal end of postglenoid process is anteroposteriorly wide (c.152[2]); long and subconical hamular process of the pterygoid (c.173[1]); hamular processes unkeeled (c.174[0]); hamular processes extending to a point in line with the middle of the zygomatic processes (c.175[3]); cranial hiatus constricted by medial projection of the parietal (c.184[2]); absent to poorly defined rectus capitus anticus muscle fossa (c.193[0]), differing from the well-defined fossa of xenorophids; posteroventral end of basioccipital crest forming a posteriorly oriented flange (c.194[2]); anterior process of periotic with well-defined fossa for contact with tympanic (c.210[3]); lateral tuberosity of periotic forming a bulbous prominence lateral to mallear fossa (c.212[1]); tegmen tympani at the base of the anterior process unexcavated (c.232[0]), differing from the excavated surface in xenorophids; articular surface of the posterior process of periotic is smooth (c.242[0]) and concave (c.243[0]); and, posterolateral sulcus of premaxilla deeply entrenched (c.310[1]).

Additional characters present in simocetids include: rostrum fairly wide (c.7[1]; shared with *Ashleycetus planicapitis Sanders & Geisler, 2015*, *Agorophius pygmaeus* (*Müller, 1849*), and *Ankylorhiza tiedemani* (*Allen, 1887*)); palatine/maxilla suture anteriorly bowed (21[0]; shared with *Patriocetus kazakhstanicus Dubrovo & Sanders, 2000*); lacrimal restricted to below the supraorbital process of frontal (c.52[0]; shared with *A. planicapitis*, *P. kazakhstanicus* and *An. tiedemani*); relatively small ventral (orbital) exposure of the lacrimal (c.56[0]; shared with *A. planicapitis*, *Archaeodelphis patrius Allen, 1921*, and *P. kazakhstanicus*); postorbital process of frontal relatively long and oriented posterolaterally and ventrally (c.62[0]; shared with *A. planicapitis*, *Mirocetus riabinini Mchedlidze, 1970* and *P. kazakhstanicus*); presence of a long posterolateral sulcus extending from the premaxillary foramen (c.73[2]; shared with *A. planicapitis*); maxillae only partially

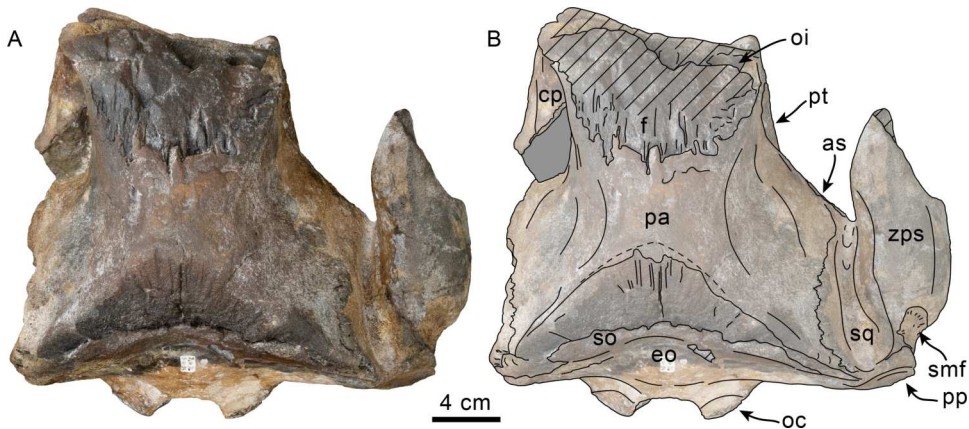

**Figure 1 Dorsal view of skull of Simocetidae gen. et sp. A (LACM 124104).** Unlabeled (A) and labeled (B) skull in dorsal view. Diagonal lines denote broken surfaces, gray shaded areas are obscured by sediment. Abbreviations: as, alisphenoid; cp, coronoid process; eo, exoccipital; f, frontal; oc, occipital condyle; oi, optic infundibulum; pa, parietal; pp, paroccipital process of exoccipital; pt, pterygoid; smf, sternomastoid fossa; so, supraoccipital; sq, squamosal; zps, zygomatic process of squamosal.

covering supraorbital processes (c.77[1]; shared with *A. planicapitis* and *Ar. patrius*); frontals slightly lower than nasals (c.125[0]; shared with *Cotylocara macei Geisler, Colbert & Carew, 2014*); intertemporal region with an ovoid cross section (c.137[1]; shared with *A. planicapitis*, *Echovenator sandersi Churchill et al., 2016*, and *C. macei*); anterior end of supraoccipital is semicircular (c.153[1]; shared with *P. kazakhstanicus*); occipital shield with distinct sagittal crest (= external occipital crest, *sensu Mead & Fordyce, 2009*) (c.156[1]; shared with *Albertocetus meffordorum Uhen, 2008*, *P. kazakhstanicus*, *Ag. pygmaeus*, and *An. tiedemani*); a nearly transverse pterygoid-palatine suture (c.163[1]; shared with *Ar. patrius*); anterior process of periotic short (c.204[2]; shared with *C. macei*).

SIMOCETIDAE GEN. ET SP. A
(Figs. 1–5; Tables 1–2)

**Material**—LACM 124104, posterior part of skull, missing most parts anterior to the frontal/parietal suture and the left squamosal; including one molariform tooth and partial atlas, axis and third cervical vertebrae. Collected by J. L. Goedert and G. H. Goedert March 21, 1984.

**Locality and horizon**—LACM Loc. 5123, Murdock Creek, Clallam Co., Washington, USA (48°09′25″N, 123°52′10″W; = locality JLG-76). At this locality specimens are found as concretions along a beach terrace about 40 m north of the mouth of Murdock Creek. Besides LACM 124104, additional specimens known from this locality include the desmostylian *Behemotops proteus Domning, Ray & McKenna, 1986* (LACM 124106; *Ray, Domning & McKenna, 1994*), additional material of the simocetid *Olympicetus* sp. 1 (LACM 124105)
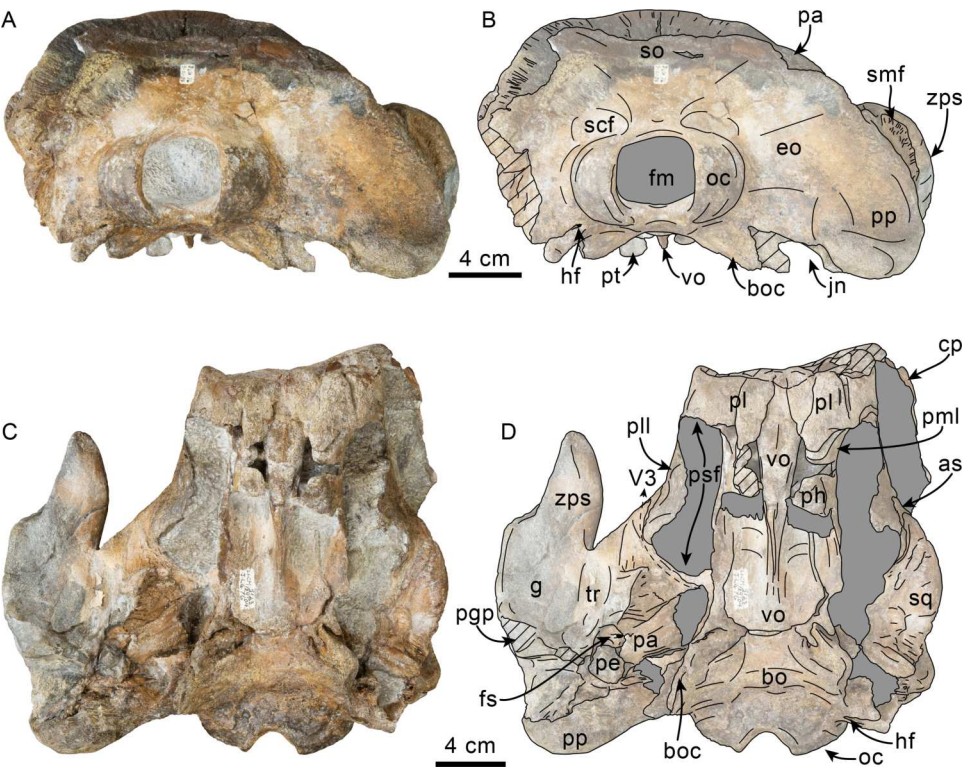

**Figure 2 Posterior and ventral views of skull of Simocetidae gen. et sp. A (LACM 124104).** Unlabeled (A) and labeled (B) skull in posterior view; unlabeled (C) and labeled (D) skull in ventral view. Diagonal lines denote broken surfaces, gray shaded areas are obscured by sediment. Abbreviations: as, alisphenoid; bo, basioccipital; bo, basioccipital crest; cp, coronoid process; eo, exoccipital; fm, foramen magnum; fs, foramen spinosum; g, glenoid fossa; hf, hypoglossal foramen; jn, jugular notch; oc, occipital condyle; pa, parietal; pe, periotic; pgp, postglenoid process; ph, pterygoid hamulus; pl, palatine; pll, pterygoid lateral lamina; pml, pterygoid medial lamina; pp, paroccipital process; psf, pterygoid sinus fossa; pt, pterygoid; scf, supracondylar fossa; smf, sternomastoid fossa; so, supraoccipital; sq, squamosal; tr, tympanosquamosal recess; V3, groove and path of mandibular branch of trigeminal nerve; vo, vomer; zps, zygomatic process of squamosal.

and *O. thalassodon* sp. nov. (LACM 158720; described below), aff. *Olympicetus* sp. (*Racicot et al., 2019*), and the aetiocetid *Borealodon osedax Shipps, Peredo & Pyenson, 2019*.

**Formation and Age**—Pysht Formation, between 30.5–26.5 Ma (Oligocene: late Rupelian-early Chattian; *Prothero, Streig & Burns, 2001a*; *Vélez-Juarbe, 2017*).

**Temporal and geographic range**—Oligocene of Washington, USA.

## Description

As preserved, the partial skull (LACM 124104; Figs. 1–4) has a pachyostotic appearance, in comparison with the other described simocetids. Based on the fused/closed sutures and heavily worn tooth, the specimen is considered to belong to an adult individual. The estimated bizygomatic width, 322 mm (c.335[2]), suggests a body length of around 3 m (based on equation "i" for stem Odontoceti from *Pyenson & Sponberg, 2011*), which is larger than any of the other described simocetids.

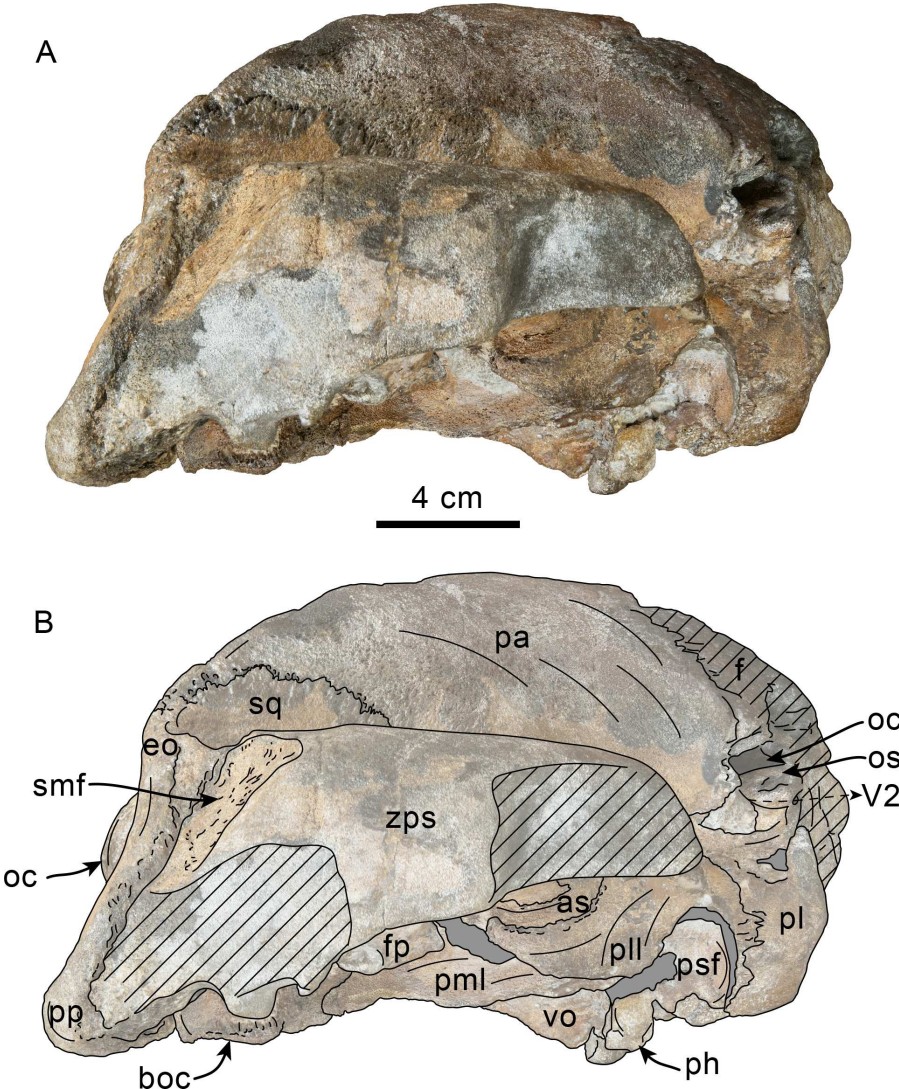

**Figure 3** **Lateral view of skull of Simocetidae gen. et sp. A (LACM 124104).** Unlabeled (A) and labeled (B) skull in right lateral view. Diagonal lines denote broken surfaces, gray shaded areas are obscured by sediment. Abbreviations: as, alisphenoid; boc, basioccipital crest; eo, exoccipital; f, frontal; fp, falciform process; oc, occipital condyle; oc, optic canal; os, orbitosphenoid; pa, parietal; ph, pterygoid hamulus; pl, palatine; pll, pterygoid lateral lamina; pml, pterygoid medial lamina; pp, paroccipital process; psf, pterygoid sinus fossa; smf, sternomastoid fossa; sq, squamosal; V2, path for maxillary nerve; vo, vomer; zps, zygomatic process of squamosal.

**Vomer**—Most of the palatal surface of the vomer is missing, as is much of the rostrum. Posteriorly, it seems to have been exposed ventrally along an elongated, diamond-shaped, window between the palatines and pterygoids as in other simocetids (Figs. 2C–2D; *Fordyce, 2002*; *Vélez-Juarbe, 2017*; see below). From this point, the vomerine keel extends posterodorsally, separating the choanae along the midline and extending to about 20 mm from the posterior edge of the bone (Figs. 2C–2D). The horizontal plate extends posteriorly to a point in line with the anterior end of the basioccipital crests, thus covering the suture

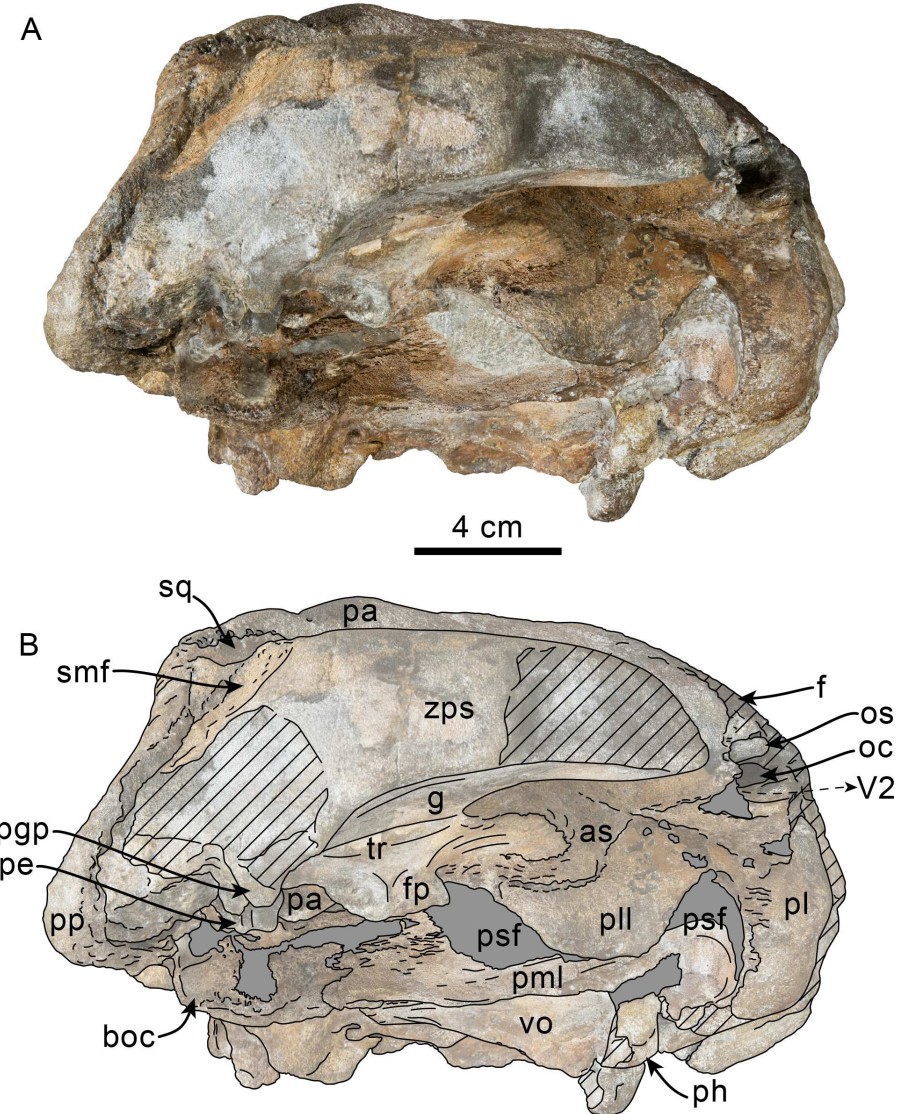

**Figure 4 Ventrolateral view of skull of Simocetidae gen. et sp. A (LACM 124104).** Unlabeled (A) and labeled (B) skull in right ventrolateral view. Diagonal lines denote broken surfaces, gray shaded areas are obscured by sediment. Abbreviations: as, alisphenoid; boc, basioccipital crest; f, frontal; fp, falciform process; g, glenoid fossa; oc, optic canal; os, orbitosphenoid; pa, parietal; pe, periotic; pgp, postglenoid process; ph, pterygoid hamulus; pl, palatine; pll, pterygoid lateral lamina; pml, pterygoid medial lamina; pp, paroccipital process; psf, pterygoid sinus fossa; smf, sternomastoid fossa; sq, squamosal; tr, tympanosquamosal recess; V2, path for maxillary nerve, vo, vomer; zps, zygomatic process of squamosal.

between the basisphenoid and basioccipital (c.191[0]; Figs. 2C–2D). The choanal surface of the horizontal plate forms a ventrally concave choanal roof, with its lateral edges slightly flared and forming a nearly continuous surface with the internal lamina of the pterygoid. **Palatine**—Only the posteriormost parts of the palatines are preserved; these are separated along the midline by the vomer, resembling the condition of other simocetids (Figs. 2C–2D; *Fordyce, 2002*; see below). In anterior view, the palatines formed the ventral and lateral

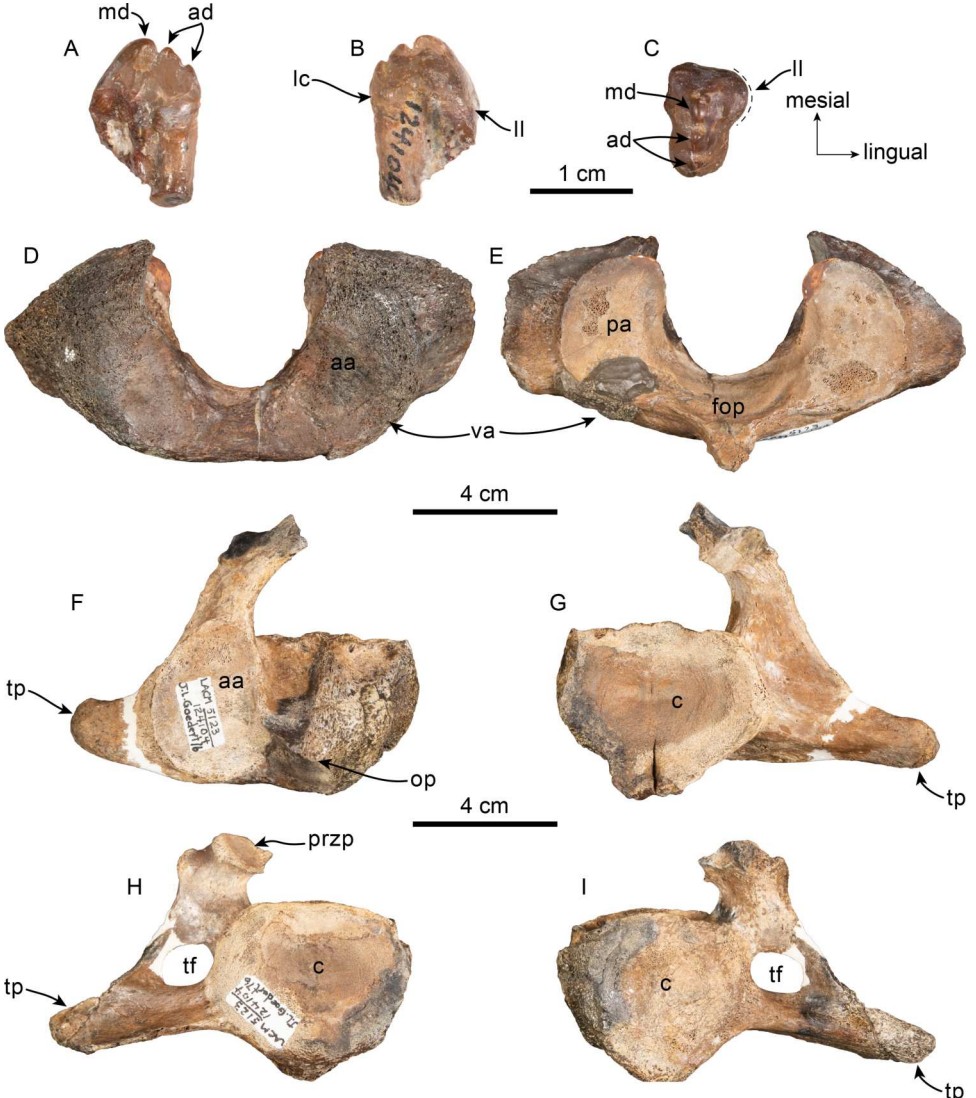

**Figure 5** **Tooth and vertebrae of Simocetidae gen. et sp. A (LACM 124104).** Upper right postcanine tooth (P4?) in buccal (A), lingual (B) and occlusal (C) views. Atlas (D, E), axis (F, G) and third cervical (H, I) vertebrae in anterior (D, F, H) and posterior (E, G, I) views. Abbreviations: aa, anterior articular facet; ad, accessory denticles; c, centrum; lc, lingual cingulum; ll, lingual lobe; fop, facet for odontoid process; hp, hypapophysis; md, main denticle; op, odontoid process; pa, posterior articular facet; przp, prezygapophysis; tf, transverse foramen; tp, transverse process; va, ventral arch.

surfaces of the internal nares, while the vomer formed the medial and dorsal surfaces. Ventrolaterally, the palatines form a vertical to semilunar contact with the pterygoids, best observed in ventral, ventrolateral and lateral views (c.163[1]; Figs. 2C–2D, 3–4), resembling the contact in *Simocetus rayi* and *Olympicetus* spp. (*Fordyce, 2002*; *Vélez-Juarbe, 2017*). An elongated groove along the ventrolateral end of the left palatine seems to have been part of the palatine foramen/canal.

**Table 1  Dimensions of simocetid skulls and mandible.** Measurements (in mm) of Simocetidae gen. et sp. A (LACM 124104), *Olympicetus thalassodon* gen. et sp. nov. (LACM 158720) and *Olympicetus* sp. 1 (LACM 124105). Modified after *Perrin (1975)*.

| | LACM 124104 | LACM 158720 | LACM 124105 |
|---|---|---|---|
| Width of rostrum at base | – | 135 | 93+ |
| Width of rostrum at 60 mm anterior to line across hindmost limits of antorbital notches | – | 105 | – |
| Greatest preorbital width (width across preorbital processes) | – | 153 | 136 |
| Greatest postorbital width | – | 187 | 150e |
| Mid-orbital width | – | 151 | 140e |
| Maximum width of external nares | – | 33 | – |
| Greatest width across zygomatic processes of squamosals | 322e | 220 | 186e |
| Greatest width of premaxillae | – | 83 | – |
| Greatest parietal width within temporal fossae | 154 | 135 | 100 |
| Vertical external height of braincase from midline of basisphenoid to summit of supraoccipital, but not including external occipital crest | 135 | 112 | – |
| Greatest length of left temporal fossa, measured to external margin of raised suture | – | 99 | – |
| Greatest width of left temporal fossa at right angles to greatest length | – | 51 | – |
| Major diameter of left temporal fossa proper | – | 111 | – |
| Minor diameter of left temporal fossa proper | 59 | 45 | – |
| Distance from foremost end of junction between nasals to hindmost point of margin of supraoccipital crest | – | 143e | – |
| Length of orbit –from ventral apex of preorbital process of frontal to apex of postorbital process | – | 55 | 40+ |
| Length of antorbital process of lacrimal | – | 18 | 12 |
| Greatest length of left pterygoid | 132 | 79 | – |
| Maximum width across occipital condyles | 92 | 78 | – |
| Height of foramen magnum | 33 | 35 | – |
| Width of foramen magnum | 39 | 32 | – |
| Cranial length –antorbital notch to condyles | – | 211 | 165+ |
| Greatest length of left mandibular ramus (as preserved) | – | 251+ | – |
| Greatest length of right mandibular ramus (as preserved) | – | 244+ | – |
| Maximum height at mandibular condyle | – | 54 | – |

**Notes.**
Abbreviations:  e,  estimate; + = measurement on incomplete element.

**Frontal**—Only the posteriormost portions of the frontals are preserved, but they are eroded (Fig. 1). Dorsally, the interfrontal suture seems to have been completely fused, and it posteriorly formed a broad V-shaped contact with the parietals, which continues as a vertical contact along the temporal surface (Fig. 3).

**Parietal**—As in other simocetids, the parietals are broadly exposed dorsally, and the interparietal is either absent or fused early in ontogeny (c.135[0], 136[1]; Fig. 1). The parietals do not extend anterolaterally, resembling *Simocetus rayi*, and differing from *Olympicetus* where the parietals extend into the base of the supraorbital processes. The

**Table 2 Dimensions of simocetid vertebrae.** Measurements (in mm) of cervical vertebrae of Simocetidae gen. et sp. A (LACM 124104) and *Olympicetus thalassodon* sp. nov. (LACM 158720).

| | LACM 124105 | LACM 158720 |
|---|---|---|
| **Atlas** | | |
| Maximum height | – | 70 |
| Maximum length | 32 | 27 |
| Width across anterior articular facets | 80+ | – |
| Width across posterior articular facets | 94 | 74 |
| Maximum width (across transverse processes) | – | 108 |
| Mid-dorsal length | – | 24 |
| Mid-ventral length (including odontoid process) | 37 | 22 |
| Neural canal height | – | 44 |
| Neural canal width | 45 | 38 |
| **Axis** | | |
| Maximum height of centrum | 46 | 33 |
| Maximum width of centrum | 47 | – |
| Maximum length of centrum | 44 | 30 |
| Width across anterior articular facets | 92e | 77 |
| Maximum width (across transverse processes) | 144e | 97 |
| Width of neural canal | 46 | 33 |
| **Cervical 3** | | |
| Height of centrum | 49 | 34 |
| Width of centrum | 53 | 34 |
| Length of centrum | 20 | 12 |
| Maximum width (across transverse processes) | 164e | 96e |
| Width of neural canal | 38e | – |
| **Cervical 4** | | |
| Height of centrum | – | 34 |
| Width of centrum | – | 35 |
| Length of centrum | – | 12 |
| **Cervical 5** | | |
| Height of centrum | – | 31 |
| Width of centrum | – | 32 |
| Length of centrum | – | 12 |
| **Cervical 6** | | |
| Height of centrum | – | 27+ |
| Length of centrum | – | 10+ |

**Notes.**
Abbreviations: e, estimate; + = measurement on incomplete element..

parietal exposure in the intertemporal region is anteroposteriorly short and broad in dorsal view, with an ovoid cross section (c.137[1]). Posterodorsally, the parietal-supraoccipital contact is transversely broad and anteriorly convex, while along the temporal surface, the parietal forms a vertical contact with the frontal (c.134[0];Fig. 1), and seems to have formed part of the posterior edge of the optic infundibulum; abaft to this point the parietal become laterally convex towards the contact with the squamosal (Figs. 3–4). Anteroventrally, on

the temporal surface, the parietal descends to contact the orbitosphenoid, a portion of the dorsal lamina of the pterygoid, the alisphenoid, and the squamosal, with which it forms part of the subtemporal crest (Fig. 4). Its contact with the squamosal on the temporal surface becomes an interdigitated, dorsally arched suture posterior to this point. In ventral view the parietal contacts the squamosal medially, partially constricting the cranial hiatus (c.184[2]; Figs. 2C–2D, 4).

**Supraoccipital**—The anterior half of the supraoccipital is not preserved, but based on the corresponding sutural marks in the parietal, it anterior edge formed a gentle semicircular arch that reached anteriorly to a level in line with the anterior half of the squamosal fossa (c.140[0], 153[1]; Fig. 1), resembling the condition observed in *Olympicetus* spp. The preserved portion of the supraoccipital forms a gently concave surface that seems to have lacked an external occipital crest (c.156[?0], 311[0]; Figs. 1, 2A–2B) observed in other simocetids. The nuchal crest is oriented dorsolaterally (c.154[1], c.155[0]), and seems to have been gently sinuous, descending posterolaterally to meet the supramastoid crest (Fig. 1, 2A–2B, 3).

**Exoccipital**—The occipital condyles are semilunar in outline, with well-defined edges, and bounded dorsally by shallow, transversely oval supracondylar fossae (c.157[1]; Figs. 2A–2B) as in *Simocetus rayi* and *Olympicetus avitus*. The foramen magnum has an oval outline, being slightly wider than high. The paroccipital processes are transversely broad and directed posteroventrally, reaching posteriorly to a level approximating the posterior edge of the condyles (c.198[1]; Fig. 2). The ventral edge of the paroccipital processes is anteroposteriorly broad, becoming thinner medially towards the broad jugular notch (c.197[0]). The hypoglossal foramen is rounded (~4 mm in diameter), located ventrolateral to the corresponding occipital condyle and well separated from the jugular notch (c.196[0]; Fig. 2).

**Basioccipital**—The basioccipital crests are short, transversely thin, oriented ventrolaterally, and diverging posteroventrally at an angle between 58−60° (c.192[0], 195[2]; Fig. 2). Each crest contacts the corresponding posterior lamina of the pterygoid along a posteroventrally oriented suture. The ventral surface between the crests is flat, with no distinct rectus capitus anticus fossa (c.193[0]). Anteriorly the contact with the basisphenoid is obscured by the vomer (Figs. 2C–2D).

**Squamosal**—The squamosal plate is flat to gently convex, contacting the parietal along a dorsally arched suture that descends anteroventrally along a sinuous path to form the posteromedial edge of the subtemporal crest (Figs. 1 and 3). Only the right zygomatic process is preserved, although incompletely, missing its anterolateral corner. The process is long, oriented anteriorly, robust and somewhat inflated when viewed dorsally, constricting the squamosal fossa (c.143[0], 189[3]; Figs. 1, 2C–2D, 3–4). The squamosal fossa is relatively deep, with a moderately sigmoidal outline of its ventral surface and gently sloping anteriorly (c.147[2], 148[1], 149[1]; Fig. 1). When viewed laterally, the dorsal edge of the zygomatic process is flat to gently convex (c.144[0]), while its ventral edge is concave (c.151[0]; Figs. 3–4). The supramastoid crest is more prominent proximally, continuing posteromedially to join the nuchal crest (c.150[0]). The sternomastoid muscle fossa on the posterior edge of the zyogomatic process is a large, shallow oval depression, broadly visible in posterior or

lateral view (c.145[1]; Figs. 2A–2B, 3). The squamosal exposure lateral to the paroccipital processes is moderate in posterior view (c.146[1]; Figs. 2A–2B). Ventrally, the postglenoid process is incompletely preserved, but seems to have been anteroposteriorly broad as in other simocetids. Posterior to the base of the postglenoid process, the external auditory meatus seems to have been broad (c.190[?0]; the posttympanic process is not preserved). The glenoid fossa is shallowly concave with nearly indistinct borders. Medial to the glenoid fossa is a shallow, oval tympanosquamosal recess (c.179[2]; Figs. 2C–2D). The falciform process is anteroposteriorly long (c.177[0]; Figs. 2C–2D, 3–4). The periotic fossa is partially obscured by a fragment of periotic; the anterior part of the fossa contains a small foramen spinosum close to the medial suture with the parietal (c.187[1]; Figs. 2C–2D), resembling the condition observed in *Olympicetus avitus*. Anteromedially, the squamosal contacts the alisphenoid along an anterolaterally oriented suture that follows the anterodorsal edge of the groove for the mandibular branch of the trigeminal nerve (c.181[1]); the groove wraps around the posterior end of the pterygoid sinus fossa, opening anteriorly (c.182[1]; Figs. 2C–2D, 4).

**Pterygoid**—The pterygoids are incompletely preserved, missing the hamular processes (Figs. 2C–2D). As in other simocetids, the pterygoids are ventromedially separated by a diamond-shaped palatal exposure of the vomer (Figs. 2C–2D). The pterygoid sinus fossa is anteroposteriorly long (99 mm) and dorsoventrally deep (at least 63 mm on the left side), transversely narrower anteriorly (25 mm) and becoming broader posteriorly (46 mm) (Figs. 2C–2D, 4). The anterior edge of the pterygoid sinus fossa is at the level of the pterygo-palatine suture, extending posteriorly to the anterior edge of the foramen ovale (c.164[2]; Figs. 2C–2D). The dorsal lamina contacts the orbitosphenoid anterodorsally, the frontal and the alisphenoid posterodorsally, along an irregularly sinuous contact, and forms the roof of the pterygoid sinus (c.166[0]; Fig. 4). The lateral lamina is transversely thin and is slightly deflected ventromedially, where, if complete, it would have met the medial lamina to enclose the pterygoid sinus fossa (c.165[?0]; Figs. 2C–2D, 3–4). The medial lamina is incompletely preserved, but medially contacts the lateral flange of the horizontal plate of the vomer to form the lateral wall of the choana, while laterally it forms the medial wall of the pterygoid sinus fossa (Figs. 2C–2D, 3–4).

**Alisphenoid**—Only a small portion of the alisphenoid can be observed on the temporal wall, where its exposure is small, wedged in between the squamosal, frontal and lateral lamina of the pterygoid (c.142[1]; Figs. 3–4). Its more anteromedial portions are covered by sediment.

**Orbitosphenoid/Optic Infundibulum**—The orbitosphenoid is exposed within the optic infundibulum where it is in contact with the parietal dorsally and palatine ventrally, and forms the dorsal, medial and ventral walls of the optic canal. A sulcus along the ventrolateral portion of the orbitosphenoid, close to its suture with the palatine, is likely the groove for the maxillary nerve (V2). Anteromedially, the bones are eroded, while more posteriorly they are obscured by sediment; therefore additional features of the optic infundibulum cannot be properly interpreted.

**Mandible**—The mandible is missing for the most part, with the exception of the left coronoid process (Fig. 1). The process has a subtriangular outline, as preserved being

about as long as high, with the dorsal edge slightly recurved medially. The general outline resembles the coronoid process of *Olympicetus avitus* (*Vélez-Juarbe, 2017*).

**Dentition**—Only a double-rooted upper right molariform tooth is preserved in association with the specimen (Figs. 5A–5C). The mesial root is mostly missing, but seems to have been buccolingually broader than the distal root, which is more cylindrical and slightly recurved buccally. The crown (mesiodistal length = 10 mm; height = seven mm; maximum buccolingual width = 8 mm) is worn, is longer than tall, and is buccolingually broader on its anterior half due to the presence of a lingual bulge, somewhat resembling tooth 'mo3' of *Olympicetus avitus* (Fig. S1E; *Vélez-Juarbe, 2017*), but differing by lacking a well-defined secondary carina with denticles. The crown has three denticles, with the apical one slightly larger than the two on the distal carina, but there are no denticles on the blunter, mesial carina (Figs. 5A–5C). There is no buccal cingulum, and only a nearly inconspicuous cingulum occurs on the distolingual corner of the base of the crown. The outline of the crown, as well as the presence of a buccolingually broad mesial root, or alternatively a third, lingual root, is similar to the condition observed in the P4 of *Simocetus rayi*, and is tentatively assigned to that position (*Fordyce, 2002*).

**Cervical vertebrae**—Only the first three cervical vertebrae are preserved, and they are unfused (c.279[0], 280[?0]; Figs. 5D–5I). The dorsal arch of the atlas is missing, as is the distal end of the transverse processes. The anterior articular facets have a semilunar outline and are shallowly concave, with relatively poorly defined ventrolateral and medial edges. The posterior facets for articulation with the axis have a suboval outline, with gently convex articular surfaces and sharp, well-defined edges. The posterior facets gently merge ventromedially with the articular facet for the odontoid (Fig. 5E). The ventral arch has a more prominent hypapophysis than that observed in *Olympicetus* spp. (Fig. 5E). The base of the transverse processes flares posterolaterally.

The axis is missing most of the apex and left half of the dorsal arch, as well as the left transverse process (Figs. 5F–5G). The pedicle is anteroposteriorly broad and flattened transversely. The postzygapophysis is oriented posterolateroventrally, forming a flat, smooth surface (Fig. 5G). The anterior articular surface is broad, with a suboval outline, and raised edges; the surface is shallowly concave, merging ventromedially with the ventral surface of the odontoid process (Fig. 5F). The odontoid process is short, broad and blunt, with a mid-dorsal ridge that extends along the dorsal surface of the centrum, reaching the distal end (Fig. 5F). Posteriorly, the centrum has a cardiform outline. The epiphysis is fused, and its surface is concave, with a mid-ventral cleft that slightly bifurcates towards its posteroventral end. The ventral surface of the centrum has a mid-ventral keel that becomes broader and more prominent towards the posterior end of the centrum. The transverse process is anteroposteriorly flat and oriented mainly laterally. There are no transverse foramina (Figs. 5F–5G).

The third cervical preserves only a portion of the right side of the neural arch; the pedicle is anteroposteriorly flattened and transversely broad. Both anterior and posterior epiphyses are fused (Figs. 5H–5I). The prezygapophysis consists of a rounded, flat surface that is oriented anterodorsomedially, complementing its counterpart in the axis. The transverse foramen is large, being slightly broader than tall (16 mm × 11 mm). The transverse process

is mainly oriented laterally; its posterior surface forms a low keel that extends from the base to the apex, and its anteroventral edge is flared (Fig. 5I). The centrum is rounded, anteroposteriorly short, with shallowly concave proximal and distal articular surfaces. Low midline keels are present along the ventral and dorsal surfaces of the centrum. A pair of small (~4 mm) nutrient foramina occur on each side of the mid-dorsal keel.

**Remarks**—LACM 124104 represents the largest known simocetid, with an estimated bizygomatic width of 322 mm, in comparison with that of *Simocetus rayi* (238 mm), which (using equation "i" from from *Pyenson & Sponberg, 2011*) results in estimated body lengths of about 3 m and 2.3 m, respectively, both of which are larger than those estimated for *Olympicetus* spp. (see below). This large simocetid shows a unique combination of characters, some of which are shared with *Olympicetus* spp. such as the more retracted position of the supraoccipital (c.140[0]), the dorsolateral orientation of the nuchal crest (c.154[1]), a shallow tympanosquamosal recess (c.179[1,2]), and an alisphenoid/squamosal suture that courses along the groove for the mandibular branch of the trigeminal nerve (c.181[1]). At the same time, some of the preserved characters seem to be unique to this taxon amongst simocetids, such as a deep squamosal fossa (c.147[2]) and the path of the groove for the mandibular branch of the trigeminal nerve which wraps around the posterior end of the pterygoid sinus fossa (c.182[1]). This specimen does preserve a remarkable amount of details of the size and morphology of the pterygoid sinus fossa, which together with other simocetids, suggest that they had well developed, large fossae, particularly when compared to those of other early diverging odontocetes, such as *Archaeodelphis patrius*, which seems to have much shorter fossae (LACM 149261, cast of type). LACM 124104 resembles, and may be congeneric with an odontocete skull from the early Oligocene Lincoln Creek Formation of Washington State, briefly described by *Barnes, Goedert & Furusawa (2001)*, sharing many characters of its morphology, including its large size (bizygomatic width = 265 mm) and the pachyostotic appearance of some of the cranial bones; this will be addressed in more detail in a follow-up study.

OLYMPICETUS *Vélez-Juarbe, 2017*

**Type species**—*Olympicetus avitus Vélez-Juarbe, 2017*.
**Included species**—*Olympicetus avitus*; *Olympicetus thalassodon* sp. nov., *Olympicetus* sp. 1.
**Temporal and geographic range**—Oligocene (late Rupelian–early Chattian; 33.7–26.5 Ma) of Washington State, USA.
**Emended Diagnosis**—Small odontocetes, with bizygomatic width ranging from 145–220 mm (c.335[0,1]), with symmetric skulls and heterodont dentition, resembling *Simocetus rayi* (*Fordyce, 2002*). Differs from *Simocetus*, other simocetids, and other stem odontocetes by the following combination of characters: having a concave posterior end of the palatal surface of the rostrum (c.19[0]; shared with Xenorophidae); posterior buccal teeth closely spaced (c.26[0]; shared with *Ashleycetus planicapitis*, *Patriocetus kazakhstanicus*, *Agorophius pygmaeus* and *Ankylorhiza tiedemani*), differing from the widely-spaced teeth of *S. rayi*; buccal teeth with ecto- and entocingula (c.32[1], 33[0]; shared with *Xenorophus sloani Kellogg, 1923a*, *Echovenator sandersi*, *Cotylocara macei* and *P. kazakhstanicus*), and unlike *S. rayi* where these features are absent; lacrimal and jugal separated (c.54[0]; shared

with CCNHM 1000, Xenorophidae, *P. kazakhstanicus*, *Ag. pygmaeus* and *An. tiedemani*); presence of a short maxillary infraorbital plate (c.60[1]; shared with CCNHM 1000 and *Archaeodelphis patrius*; = infraorbital process *sensu* *Mead & Fordyce, 2009*); infratemporal crest of the frontal forming a well-defined ridge along the posterior edge of the sulcus for the optic nerve (c.63[0]; shared with Xenorophidae); posteriormost end of the nasal process of the premaxilla in line with the anterior half of the supraorbital process of the frontal (c.75[2]), differing from the longer process of *S. rayi*; posteriormost end of the ascending process of the maxilla in line with the posterior half of the supraorbital process of the frontal (c.78[2]; shared with *Ashleycetus planicapitis* and *Archaeodelphis patrius*); lack of a premaxillary cleft (c.110[0]; present in *S. rayi*); anteriormost point of the supraoccipital in line with the floor of the squamosal fossa (c.140[0]), differing from the more anterior position in *S. rayi*; having a relatively shallow squamosal fossa (c.147[1]; shared with *Ar. patrius* and *P. kazakhstanicus*), thus differing from the deeper fossae of *Simocetus rayi* and Simocetidae gen. et sp. A; involucrum of the tympanic bulla lacking a transverse groove (c.272[1]; shared with *C. macei*); dorsal process of atlas larger than ventral process (c.278[2]); presence of three mesial and three to four distal denticles on main upper molars (c.328[3], 329[3,4]); and, presence of four distal denticles on main lower molars (c.331[4]). Potential autapomorphies of this clade include: absence of a posterior dorsal infraorbital foramen ( = maxillary foramen; c.76[0]), differing from *S. rayi* which has two foramina on each side located medial to the orbit; presence of a transverse cleft on the apex of the zygomatic process of the squamosal (c.337[1]); arched palate, and, saddle-like profile of the skull roof (when viewed laterally).

### *OLYMPICETUS THALASSODON*, sp. nov.
(Figs. 6–13; Tables 1–5)

**Holotype**—LACM 158720, partial skull with articulated mandibles, including 18 teeth, periotics and tympanic bullae, cervical vertebrae 1–6, and hyoids; missing distal end of rostrum/mandible. Collected by J. L. Goedert and G. H. Goedert, July 30, 1983.
**Type locality and horizon**—LACM Loc. 8093, Murdock Creek, Clallam Co., Washington State, USA. (48°09′27″N, 123°52′17″W = locality JLG-75). The specimen was found as a large concretion about 130 m northwest of LACM Loc. 5123.
**Formation and age**—Pysht Formation, between 30.5–26.5 Ma (Oligocene: late Rupelian-early Chattian; *Prothero, Streig & Burns, 2001a*; *Vélez-Juarbe, 2017*).
**Temporal and geographic range**—Oligocene of Washington State, USA.
**Differential diagnosis**—Species of relatively small bodied odontocete with bizygomatic width of about 220 mm (c.335[1]), differing from *Olympicetus avitus* and *Olympicetus* sp. 1 by the following combination of characters: dorsolateral edge of ventral infraorbital foramen formed by lacrimal (c.58[2]), differing from *Olympicetus* sp. 1 where it is formed by the maxilla, and *O. avitus* where it is formed by the maxilla and lacrimal; intertemporal region with ovoid cross section with the presence of a low sagittal crest (c.137[0]); lack of a well-defined sternomastoid fossa on the posterior edge of the zygomatic process of the squamosal (c.145[0]); tympanic bulla proportionately narrow and long (c.252[0].
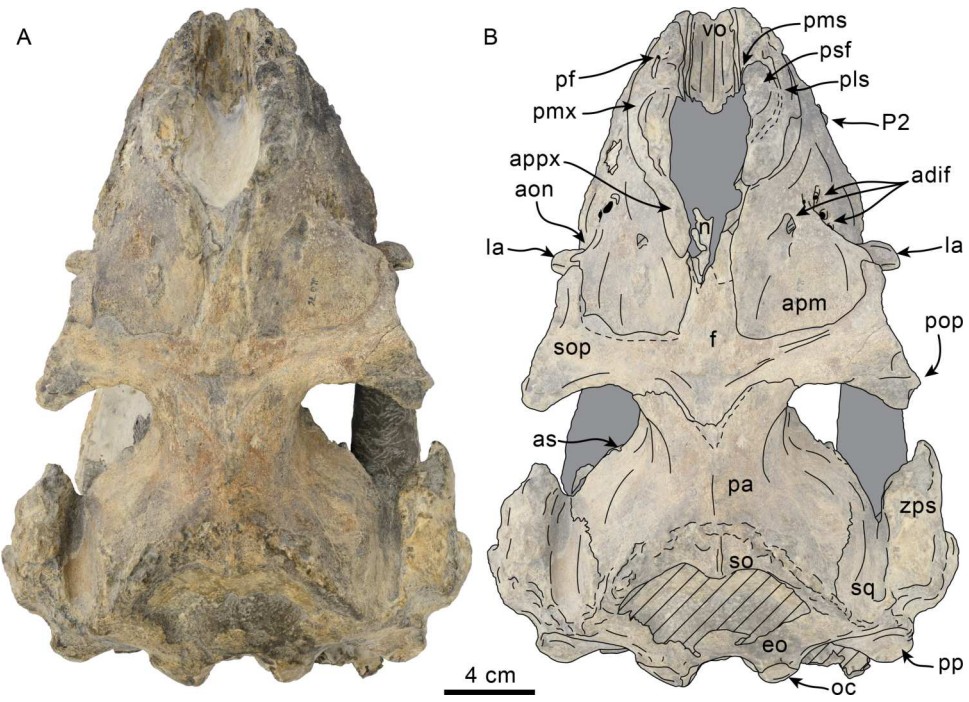

**Figure 6** **Dorsal view of skull of *Olympicetus thalassodon* sp. nov. (LACM 158720).** (A) Unlabeled and (B) labeled skull in dorsal view. Diagonal lines denote broken surfaces, gray shaded areas are obscured by sediment. Abbreviations: adif, anterior dorsal infraorbital foramina; aon, antorbital notch; ascending process of maxilla; appx, ascending process of premaxilla; as, alisphenoid; eo, exoccipital; f, frontal; la, lacrimal; n, nasal; oc, occipital condyle; P2, second upper premolar; pa, parietal; pf, premaxillary foramen; pls, posterolateral sulcus; pms, posteromedial sulcus; pmx, premaxilla; pop, postorbital process; pp, paroccipital process of exoccipital; psf, premaxillary sac fossa; so, supraoccipital; sop, supraorbital process of frontal; sq, squamosal; vo, vomer; zps, zygomatic process of squamosal.

Further differing from *O. avitus* by: posterior wall of the antorbital notch formed by the lacrimal (c.16[1]); interprominential notch of the tympanic bulla divided by a transverse ridge (c.268[0]); upper molars with four denticles on the distal carinae (c.329[4]); lower molars with a single mesial denticle (c.330[1]), and parietals not forming part of the supraorbital processes, differing from *O. avitus* where they extend into the posteromedial part of the process; and from *Olympicetus* sp. 1 by: dorsal edge of orbit higher, relative to the lateral edge of rostrum (c.48[2]); and, temporal crest along the posterior edge of the supraorbital process of the frontal (c.132[0]). *Olympicetus thalassodon* sp. nov. can be further differentiated from other simocetids by the following characters: mandible with a relatively straight profile in lateral view (c.39[0]), differing from the more strongly arched mandible of *S. rayi*; mandibular condyle positioned at about the same level as the alveolar row (c.46[1]); lack of a well-defined dorsal condyloid fossa (c.157[0]; otherwise present on other simocetids); posterior process of the periotic exposed on the outside of the skull (c.250[0]); moderately large bizygomatic width (c.335[2]; shared with *S. rayi*), differing from the smaller size of *O. avitus* and *Olympicetus* sp. 1, or the relatively larger Simocetidae gen. et sp. A; nasals contacting the maxillae along their posterolateral corners; longer

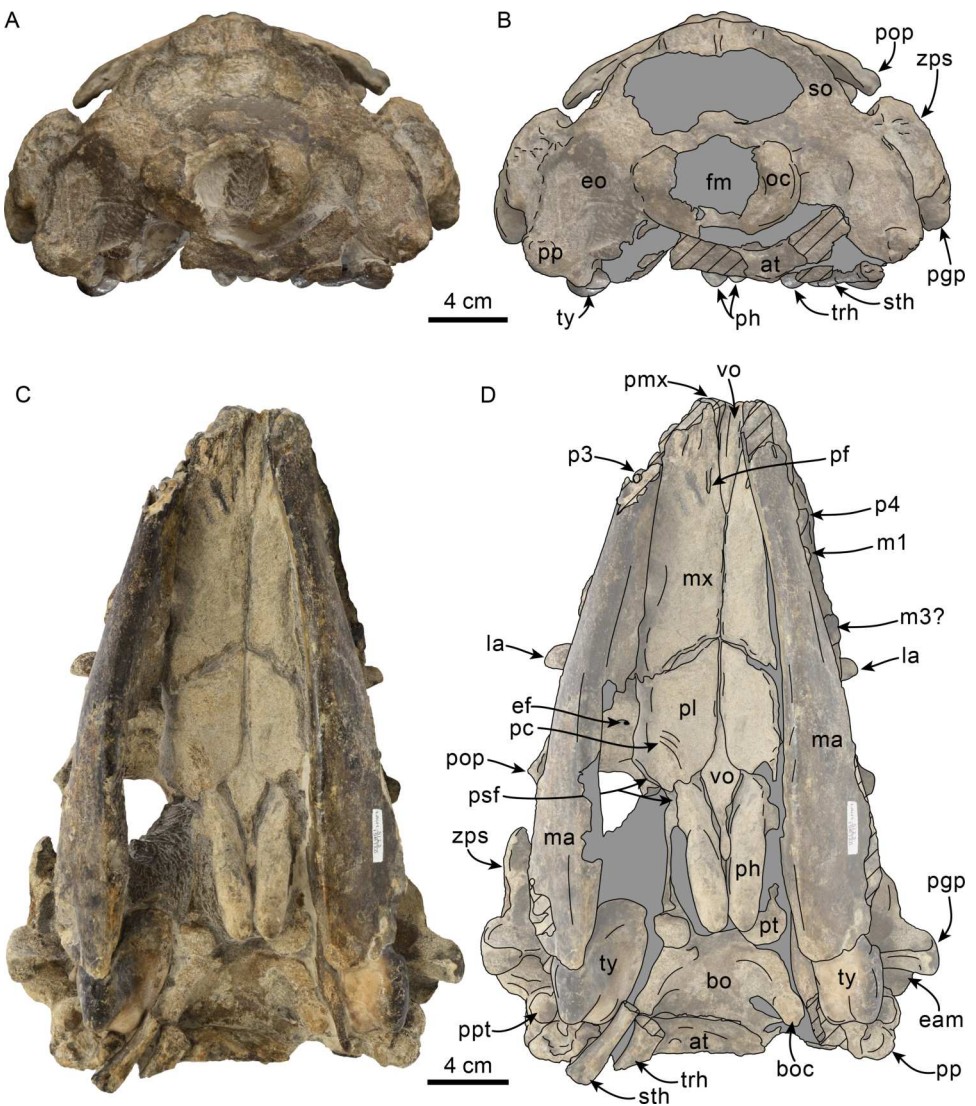

**Figure 7 Posterior and ventral views of skull of *Olympicetus thalassodon* sp. nov. (LACM 158720).**
Unlabeled (A) and labeled (B) skull in posterior view; (C) unlabeled and labeled skull in ventral view. Diagonal lines denote broken surfaces, gray shaded areas are obscured by sediment. Abbreviations: at, atlas; bo, basioccipital; boc, basioccipital crest; eam, external auditory meatus; ef, ethmoid foramen; la, lacrimal; m1, first lower molar; m3?, third lower molar; ma, mandible; mx, maxilla; p3–4, third and fourth lower premolars; pc, palatal crest; pf, major palatine foramen; pgp, postglenoid process; ph, pterygoid hamulus; pl, palatine; pmx, premaxilla; pop, postorbital process; pp, paroccipital process; ppt, posterior process of tympanic; psf, pterygoid sinus fossa; pt, pterygoid; sth, stylohyal; trh, thyrohyal; ty, tympanic; vo, vomer; zps, zygomatic process of squamosal.

paroccipital and postglenoid processes; and, thyrohyals tubular and not fused to basihyal (c.338[0]).

**Etymology**—Combination of *thalasso-* from the Greek word 'thalassa' meaning 'sea' and *-odon* from the Greek word 'odon' meaning 'tooth', in reference to the marine habitat of the species and its particular dental morphology.

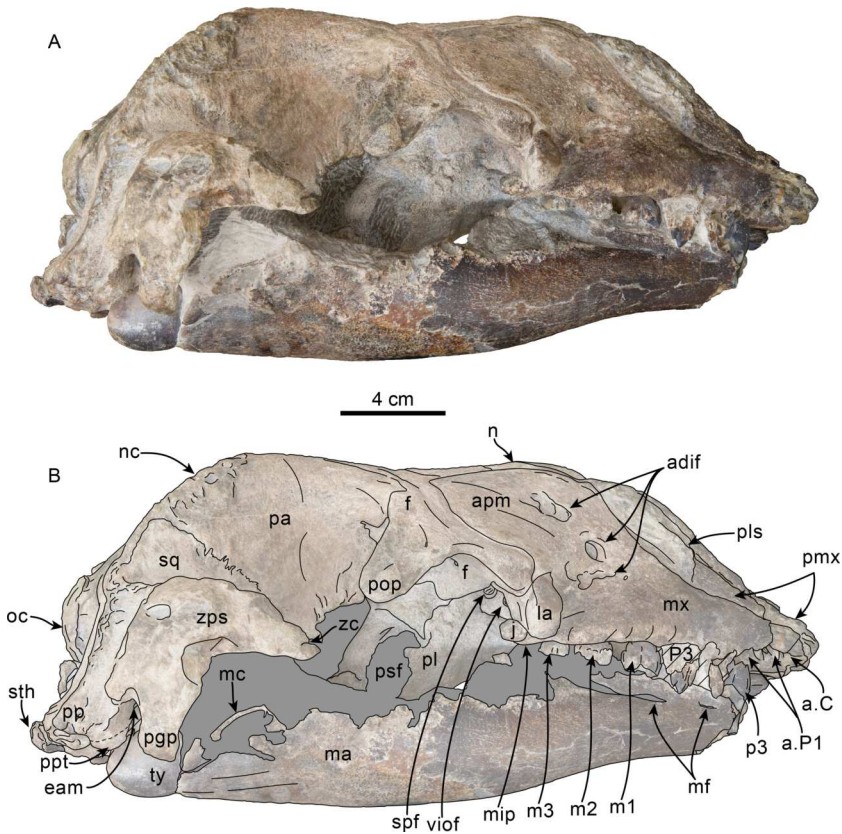

**Figure 8** **Lateral view of skull of *Olympicetus thalassodon* sp. nov. (LACM 158720).** Unlabeled (A) and labeled (B) skull in right lateral view. Diagonal lines denote broken surfaces, gray shaded areas are obscured by sediment. Abbreviations: a.C, alveolus for upper canine; a.P1, alveoli for first upper premolar; adif, anterior dorsal infraorbital foramina; apm, ascending process of maxilla; eam, external auditory meatus; f, frontal; j, jugal; la, lacrimal; m1–3, lower molars 1, 2 and 3; ma, mandible; mc, mandibular condyle; mip, maxillary infraorbital plate; mf, mental foramina; mx, maxilla; n, nasal; nc, nuchal crest; oc, occipital condyle; p3, lower third premolar; P4, upper fourth premolar; pa, parietal; pgp, postglenoid process; pl, palatine; pls, posterolateral sulcus; pop, postorbital process; pp, paroccipital process; psf, pterygoid sinus fossa; ptp, posttympanic process; spf, sphenopalatine foramen; sq, squamosal; sth, stylohyoid; ty, tympanic; viof, ventral infraorbital foramen; zc, zygomatic cleft; zps, zygomatic process of squamosal.

## Description

Description is based on the holotype (LACM 158720; Figs. 6–13). Some of the preserved mandibular and maxillary teeth are *in situ*, allowing for determination of associated, loose teeth. The estimated body length is ∼2.15 m, based on equation ''i'' for stem Odontoceti in *Pyenson & Sponberg (2011)*. The terminology used herein follows *Mead & Fordyce (2009)*. Based on the closed or tightly sutured contacts between the cranial bones, LACM 158720 is considered to represent an adult individual.

**Premaxilla**—The part of the premaxillae anterior to the premaxillary foramen is not preserved. Each premaxilla preserves a single, small (diam. = 3 mm) foramen located far anterior to the antorbital notch (c.70[1], 71[0], 72[0]; Fig. 6). The ascending process adjacent to the external nares is divided by a long posterolateral sulcus (c.73[2]) and a short,

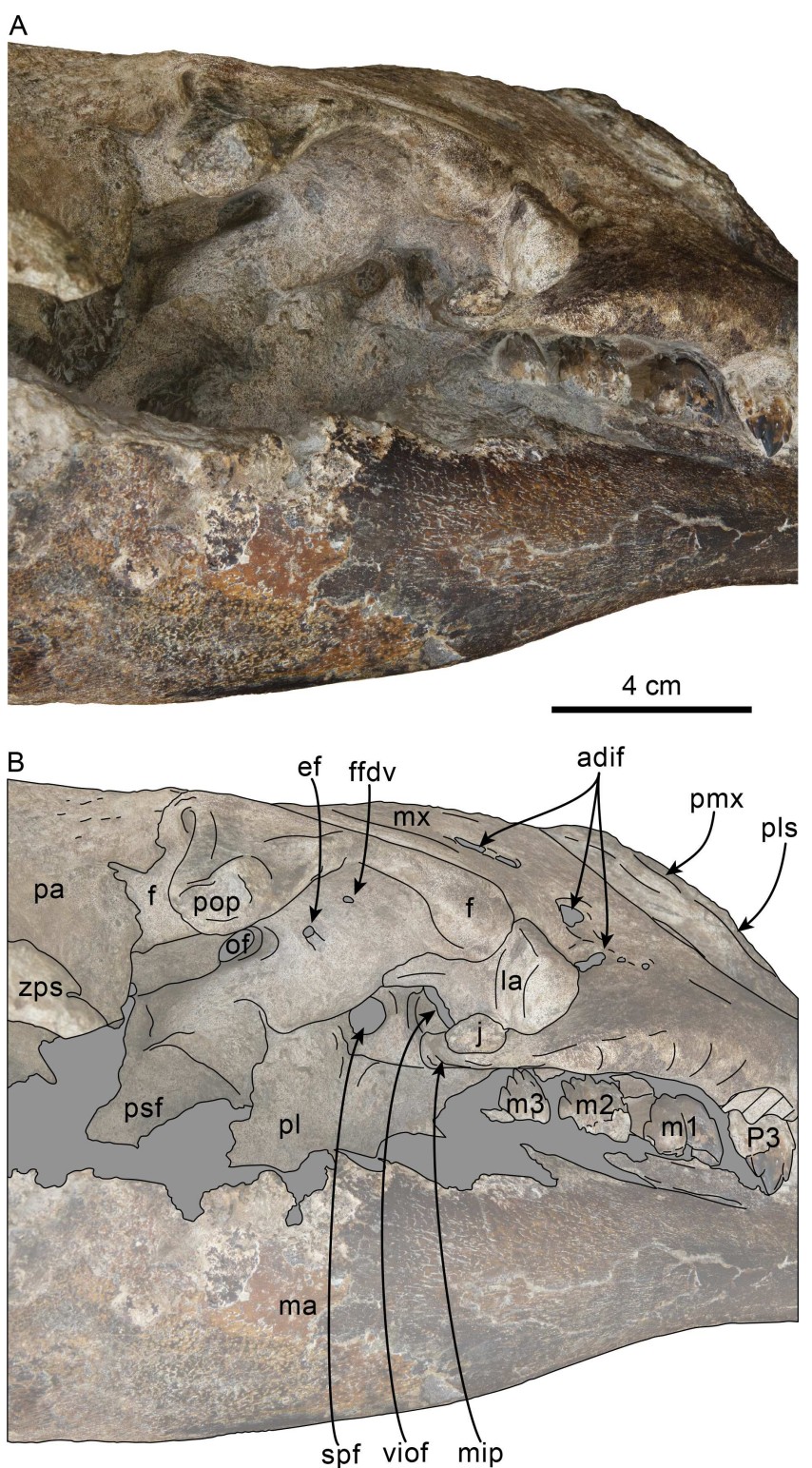

**Figure 9** **Orbital region of skull of *Olympicetus thalassodon* sp. nov. (LACM 158720).** Unlabeled (A) and labeled (B) orbital region in right lateral view. (continued on next page...)
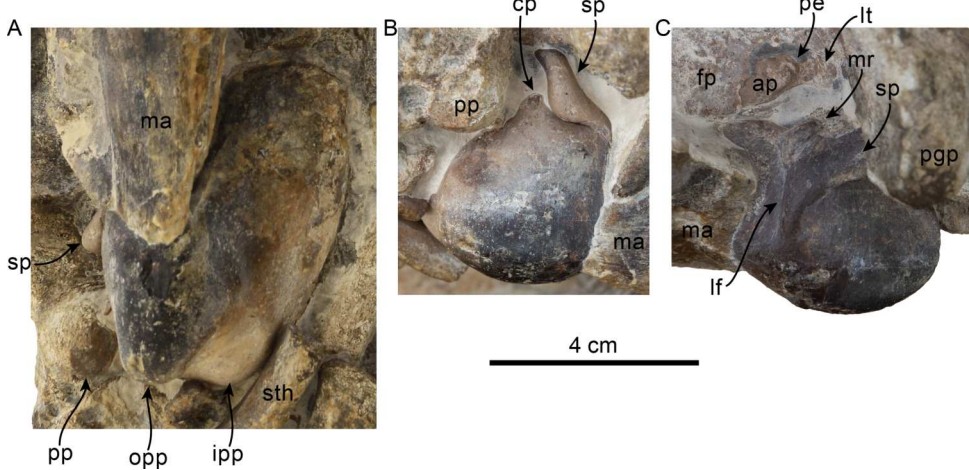

**Figure 10 Tympanic bullae and periotic of *Olympicetus thalassodon* sp. nov. (LACM 158720).** Articulated right tympanic bulla in ventral (A) and lateral (B) views; articulated left tympanic bulla and periotic in anterolateral (C) view. The bullae and periotic have been highlighted to differentiate them from the surrounding bones which obscure some parts. Abbreviations: ap, anterior process; cp, conical process; fp, falciform process; ipp, inner posterior prominence; lf, lateral furrow; lt, lateral tuberosity; ma, mandible; mr, mallear ridge; opp, outer posterior prominence; pe, periotic; pgp, postglenoid process; pp, posterior process; sp, sigmoid process; sth, stylohyal.

incipient, posteromedial sulcus (c.319[1]), both of which extend from the premaxillary foramen, forming the lateral and anteromedial limits of the premaxillary sac fossa (Fig. 6). The premaxillary sac fossae are anteroposteriorly flat to shallowly concave, transversely narrow, and anteroposteriorly long (c.69[0]; 320[0], 324[1]), resembling the condition observed in *O. avitus*. The premaxillae form the lateral edges of the external nares and mesorostral canal (c.74[0]). Posterior to the premaxillary sac fossa, the ascending process of the premaxilla extends posteriorly as a transversely thin flange, reaching a level just beyond the preorbital process of the frontal (c.75[2]), leaving a narrow gap where the maxilla contacts the nasal. In contrast, in *O. avitus* the ascending process extends farther posteriorly, to a point closer to the middle of the supraorbital processes, separating the nasals from the maxillae (*Vélez-Juarbe, 2017*).

**Maxilla**—As preserved, the palatal surface is anteroposteriorly concave and transversely convex to flat (c.17[0]). Anteriorly the vomer is exposed ventrally through an elongated window between the maxillae as in *Simocetus rayi*. Similarly, a pair of major palatine foramina are located on each side at the proximal end of this opening (c.18[0]; Figs. 7C–7D). Posteriorly, the maxillae contacts the palatines along an anteriorly-bowed contact

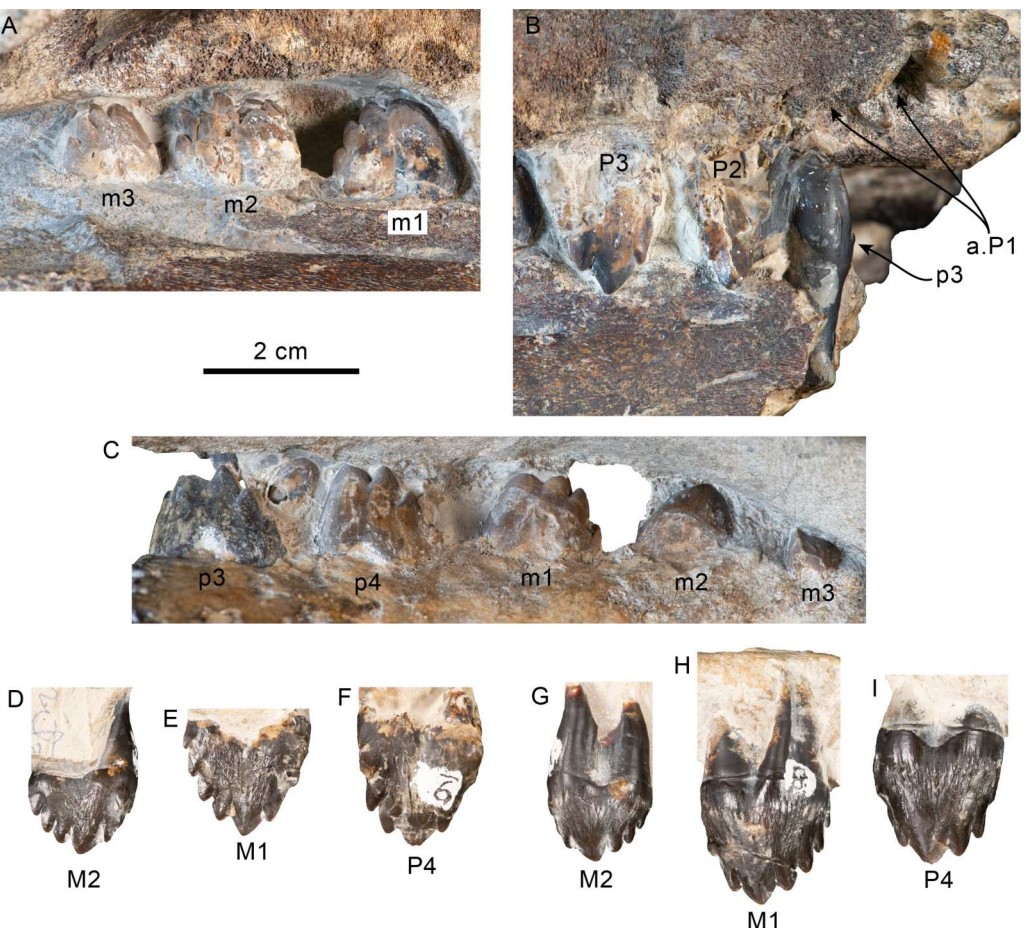

**Figure 11** **Upper and lower right dentition of *Olympicetus thalassodon* sp. nov. (LACM 158720).** Upper and lower right postcanine teeth in buccal (A–B) views; lower right postcanine teeth (p3-m3) in lingual (C) view; upper right P4-M2 in buccal (D–F) and lingual (G–I) views. Abbreviations: a.P1, alveoli for first upper premolar; M1-2, first and second upper molars; m1-3, first through third lower molars; P2-4, second through fourth upper premolars; p3-4, third and fourth lower premolars.

(c.20[0], 21[0]). The alveolar row diverges posteriorly (c.23[0]); it is incompletely preserved anteriorly, but based on the preserved dentition and visible alveoli, there were at least seven closely-spaced maxillary teeth, with the most posterior six representing double-rooted P1-4, M1-2, with the most anterior of the preserved alveoli representing an anteroventrally-oriented single rooted ?canine (c.24[4], 26[0]; Fig. 8). Posteriorly, the maxillary tooth row extends beyond the antorbital notch, forming a short infraorbital plate that underlies the jugal (c.60[1]; Fig. 9). The ventral infraorbital foramen has an oval outline (15 mm wide by 9 mm high) and is bounded laterally and dorsally by the lacrimal and ventrally and medially by the maxilla (c.58[2], 59[0]; Fig. 9).

Proximally, the rostrum is wide relative to the width of the skull across the orbits (c.7[1]), and the lateral edges of the maxillae are bowed out, giving the antorbital notch a 'V'-shaped outline (c.12[1]; Fig. 6). The surface of the maxillae anterior and anteromedial

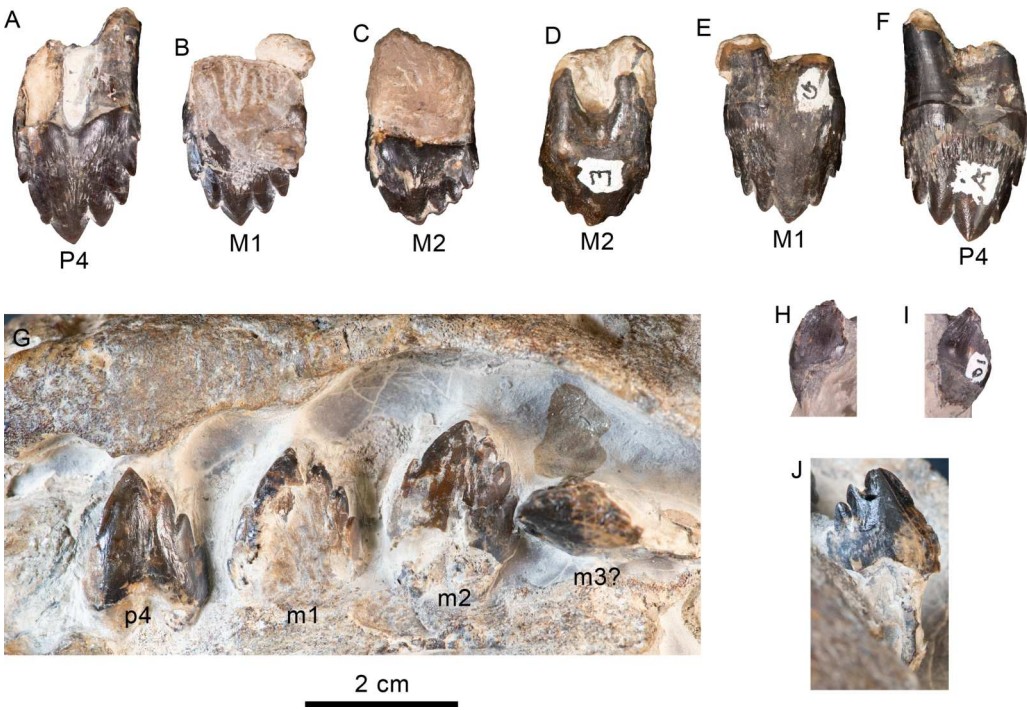

**Figure 12  Upper and lower left dentition of *Olympicetus thalassodon* sp. nov. (LACM 158720).** Upper left P4-M2 in buccal (A–C) and lingual (D–F) views; lower left postcanine teeth (p4-m2) in buccal (G) view; canine or incisor in buccal (H) and mesial (I) views; postcanine tooth, likely the left m3, in lingual (J) view. Abbreviations: M1-2, first and second upper molars; m1-2, first and second lower molars; P4/p4, upper and lower fourth premolars.

to the orbits is flat to shallowly convex (c.66[0]), lacking the rostral basin observed in some xenorophids (*e.g.*, *Cotylocara macei*; *Geisler, Colbert & Carew, 2014*). As in *O. avitus*, this surface has a cluster of three to four anterior dorsal infraorbital foramina with diameters ranging between 4–6 mm, with the posteriormost foramen located dorsomedial to the antorbital notch (c.65[3]). However, in contrast to *O. avitus* the maxilla does not extend anterolaterally to form the posterior wall of the antorbital notch (c.16[1]; Figs. 6, 8), thus more closely resembling the condition observed in *Simocetus rayi*. Posteromedial to the antorbital notch, the maxilla extends over the supraorbital process, covering a little more than the anterior half of the process and laterally to within 12 mm of the edge of the orbit, while medially it contacts the ascending process of the premaxilla and the nasal, forming a gently sloping dorsolaterally-facing surface (c.49[0], 77[1], 78[], 79[0], 80[0], 130[0], 308[1]; Figs. 6, 8).

**Vomer**—Dorsally the vomer forms the ventral and lateral surfaces of the mesorostral canal, which seems to have been dorsally open, at least for the length of the rostrum that is preserved. Th vomer has a V- to U-shaped cross section, with a more acute ventral edge anteriorly (c.5[0]; Fig. 6). Anteriorly, along the palatal surface of the rostrum, the vomer is exposed through a narrow elongate window mostly between the maxillae and the premaxillae distally, resembling the condition in *S. rayi* and, possibly, *Olympicetus avitus*

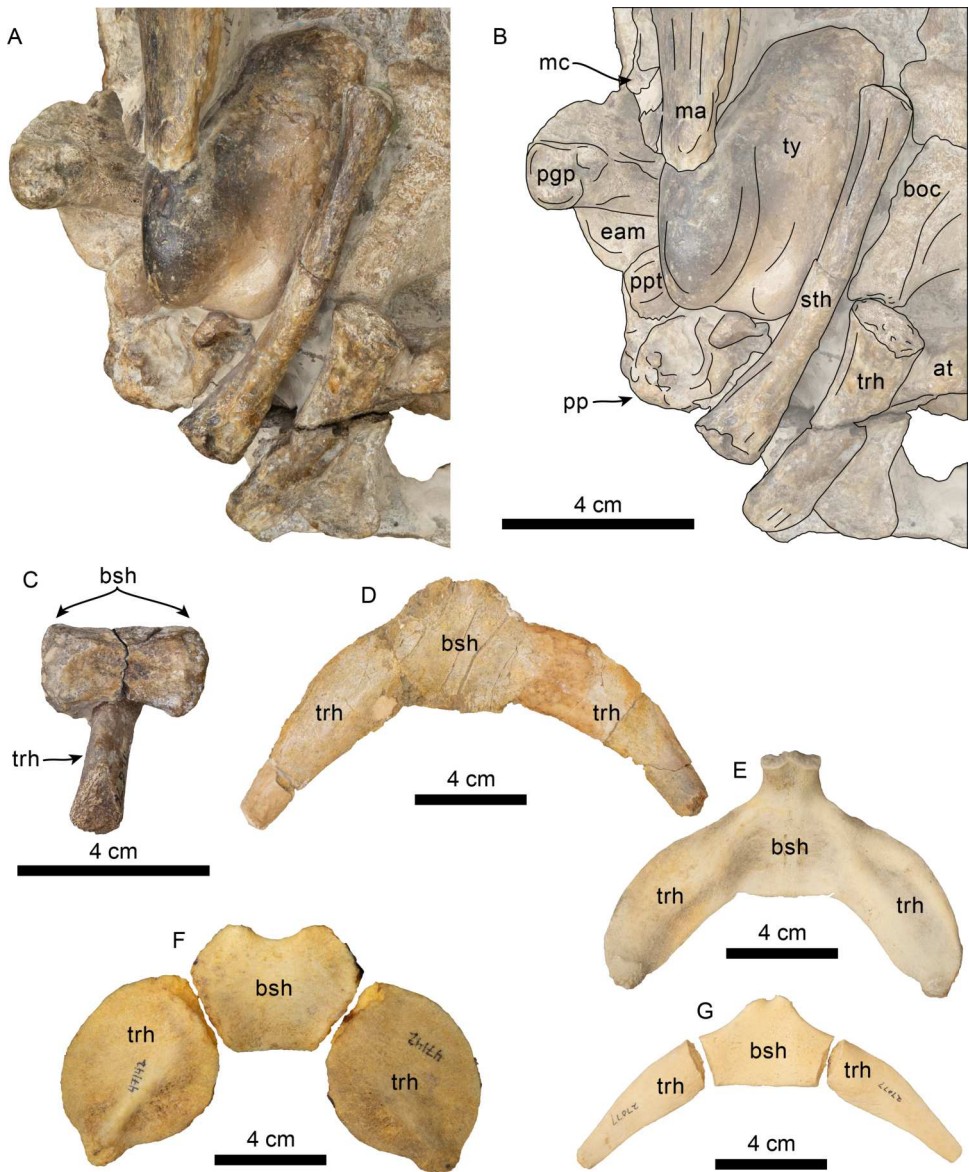

**Figure 13 Hyoid elements of *Olympicetus thalassodon*. sp. nov. (LACM 158720) and other odontocetes.** (A) Unlabeled and (B) labeled closeup of the right side of the basicranium of Olympicetus thalassodon in ventral view. Dorsal views of basihyal and thyrohyals of: (C) Olympicetus thalassodon (LACM 158720); (D) Albireo whistleri (UCMP 314589); (E) Phocoenoides dalli (LACM 43473); (F) Kogia sima (LACM 47142); and, (G), Sagmatias obliquidens (LACM 27077). Abbreviations: at, atlas; boc, basioccipital crest; bsh, basihyal; eam, external auditory meatus; ma, mandible; mc, mandibular condyle; pgp, postglenoid process; pp, paroccipital process; ppt, posterior process of the tympanic; sth, stylohyal; trh, thyrohyal; ty, tympanic.

(Figs. 7C–7D; *Fordyce, 2002*; *Vélez-Juarbe, 2017*). The vomer is exposed again towards the posterior end of the palate along a diamond-shaped window between the palatines and the pterygoids, resembling *S. rayi* (Figs. 7C–7D; *Fordyce, 2002*) Similarly, the vomer seems to have been exposed posteriorly in *O. avitus*, although the window may have been

**Table 3 Dimensions of simocetid tympanic bullae.** Measurements (in mm) of tympanic bullae of *Olympicetus thalassodon* sp. nov. (LACM 158720), *Olympicetus avitus* (LACM 126010), and *Olympicetus* sp. A (LACM 124105) (modified from *Kasuya, 1973*; *Geisler, Colbert & Carew, 2014*).

|  | LACM 158720 | LACM 126010 | LACM 124105 |
|---|---|---|---|
| Maximum length (without posterior process) | 65 | 50 | 49 |
| Maximum length (including posterior process) | 74 | 54 | – |
| Distance from anterior tip to inner posterior prominence | 61 | 50 | 48 |
| Maximum width at level of the sigmoid process | 40 | 35 | 34 |
| Height at sigmoid process | 46 | 37 | 36 |
| Maximum width of sigmoid process | – | 15 | 15 |
| Maximum length of posterior process | 16+ | 18 | – |

**Notes.**
Abbreviations: +, measurement on incomplete or obscured element..

**Table 4 Dimensions of simocetid teeth.** Measurements (in mm) of left (l) and right (r) teeth of *Olympicetus thalassodon* sp. nov. (LACM 158720).

| Designation | Length of crown | Width of crown | Height of crown |
|---|---|---|---|
| ?Canine | 7.4 | 7.2 | 7.7 |
| P2 (r) | – | – | 15.6 |
| P3 (r) | 15.7 | – | 17.5 |
| P4 (r) | 16.5 | 9.7 | 17.5 |
| P4 (l) | 17.9 | 9.3 | 18.3 |
| M1 (r) | 16.4 | 9.4 | 17.9 |
| M1 (l) | 16.5 | 9.4 | 16.7 |
| M2 (r) | 14.1 | 8.1 | 11.9 |
| M2 (l) | 14.6 | 8.4 | 11.7 |
| p3 (r) | 17.1 | 7.4 | 14.4+ |
| p4 (r) | 15.2 | – | 13.6+ |
| p4 (l) | 16.7 | – | 18.6 |
| m1 (r) | 17.8 | 6.4 | 13.9+ |
| m1 (l) | 17.6 | – | 18.3 |
| m2 (r) | 16.5 | – | 13.5+ |
| m2 (l) | 17.4 | – | 17.3 |
| m3 (r) | 13.4 | – | 11.6 |
| ?m3 (l) | 15.4 | 9.0 | 13.5 |

**Notes.**
Abbreviations: +, measurement on incomplete element.

comparably smaller. The choanae are filled with sediment, thus making it impossible to determine the posterodorsal extension of the vomer (c.191[?]).

**Palatine**—As in *Simocetus* and *Olympicetus avitus*, the anterior edge of the horizontal plate of the palatine extends to about 10 mm anterior to the level of the antorbital notches, forming the shallowly concave proximal surface of the palate (Figs. 7C–7D). The posterior edges of the right and left palatines are separated in the midline by the

**Table 5  Dimensions of simocetid hyoid elements.** Measurements (in mm) of hyoid elements of *Olympicetus thalassodon* sp. nov. (LACM 158720) (modified after *Johnston & Berta, 2011*).

| | |
|---|---|
| **Stylohyal (right)** | |
| Maximum length | 85 |
| Maximum width of distal articular surface | 11 |
| Anteroposterior thickness at mid length | 10 |
| Transverse width at mid length | 6 |
| Maximum width of proximal articular surface | 16 |
| Anteroposterior thickness of proximal articular surface | 8 |
| **Basihyal** | |
| Maximum length along the midline | 14 |
| Maximum depth along the midline | 10 |
| Maximum transverse width | 33 |
| Length of articular surface | 20 |
| Height of articular surface | 14 |
| **Thyrohyal (right)** | |
| Maximum length | 59 |
| Maximum width of distal articular surface | 11 |
| Maximum height of distal articular surface | 16 |
| Dorsoventral thickness at mid length | 7 |
| Transverse width at mid length | 11 |
| Maximum width of proximal articular surface | 18 |
| Maximum height of proximal articular surface | 13 |

vomer, even more than in *Simocetus* (Figs. 7C–7D; *Fordyce, 2002*). Posterolaterally an elevated palatal crest originates at the contact with the pterygoid hamulus and extends anterodorsally along the lateral surface of the palatine, approximating, but not reaching, the infundibulum for the sphenopalatine and infraorbital foramina. It instead become a shallow groove that reaches the sphenopalatine foramen as in *O. avitus* (Figs. 7C–7D, 8). The lateral surface of the palatine contacts the frontal dorsally to form the posteroventral edge of the sphenopalatine foramen, and the maxilla anteriorly, and forms the ventral edge of the infundibulum for the sphenopalatine and infraorbital foramina (Figs. 8–9). In posterolateral view, the infundibulum has an oval outline, measuring 28 × 15 mm, while the rounded sphenopalatine foramen has a diameter of about 8 mm. Ventrally and laterally, each palatine has a nearly transverse contact with the corresponding pterygoid (c.163[1]; Figs. 7C–7D, 8), resembling the condition observed in *O. avitus*, *Simocetus rayi* and *Archaeodelphis patrius*.

**Nasal**—The nasals are poorly preserved and seem to have formed the highest point of the vertex (c.114[?0], 124[0], 125[0], 312[0]; Figs. 6, 8) as in *Olympicetus avitus* and *Simocetus*. Anteriorly, the nasals reach to about 24 mm beyond the antorbital notches, while posteriorly they are in line with the preorbital process of the frontals (c.81[3], 123[1]; Fig. 6). The nasals are anteroposteriorly elongated, face dorsally, form a low transversely convex arch, are dorsoventrally thin (<3 mm) and are separated posteriorly along the midline by the narial processes of the frontal (c.116[0], 118[0], 120[1], 121[2], 122[1], 312[0], 321[0]).

Each nasal seems to contact the ascending process of the premaxilla for most of its length with only its posterolateral corner contacting the maxilla, differing from *Olympicetus avitus* where the premaxilla extends beyond the posterior edge of the nasal (*Vélez-Juarbe, 2017*).
**Frontal**—Dorsally along the midline, the frontals are wedged between the maxillae and posterior edge of the nasals, forming a large semi-rectangular surface (c.126[1]; Fig. 6). Posterior to this surface, the frontals are shallowly depressed towards their contact with the parietals, forming a saddle-like outline of the skull roof in lateral view, resembling the condition observed in *O. avitus* (Fig. 8). The interfrontal suture is completely fused; dorsally the frontals form a broad, V-shaped contact with the parietals, whereas their contact along the temporal surface is nearly vertical. The supraorbital processes gently slope ventrolaterally from the midline (c.47[0]), and only their anterior half is covered by the ascending process of the maxillae (Figs. 6, 8). The preorbital processes are rounded and only partially covered by the maxillae and are thus exposed dorsally; anteriorly they contact the maxillae and anteroventrally the lacrimals. The postorbital process is blunt, long, and oriented posterolaterally and ventrally to a level nearly in line with the lacrimal when viewed laterally (c.62[0]; Fig. 8). The orientation of the postorbital process gives the orbit a slight anterolateral orientation in dorsal view, and in lateral view the orbit is highly arched and positioned high relative to the rostral maxillary edge as in *O. avitus* (c.48[2]; Figs. 6, 8). The posterior edge of the supraorbital process is defined by a relatively sharp orbitotemporal crest that becomes blunter towards its contact with the orbital process of the parietal.

Ventrally, in the orbital region, the frontal contacts the lacrimal anterolaterally to form the anterior edge of the orbit (Figs. 8–9). More medially the frontal contacts the maxilla and palatine, forming the posterodorsal border of the infundibulum for the sphenopalatine and infraorbital foramina (Figs. 8–9). Medially, the optic foramen has an oval outline ($\sim$10 $\times$ 5 mm) and is oriented anterolaterally; the posterior edge of the optic foramen and infundibulum is defined by a low infratemporal crest (c.63[0]; Fig. 9). As in *Simocetus rayi* and *O. avitus*, a small ($\sim$3 mm diameter) ethmoid foramen (sensu *Fordyce, 2002*) is located anterolateral to the optic foramen, while a series of additional, smaller foramina (1–2 mm) for frontal diploic veins are located more laterally.
**Lacrimal + Jugal**—Only a small, cylindrical portion of the proximal end of the jugal is preserved; it is set in a close-fitting socket formed by the lacrimal anterodorsally, and the maxilla anteriorly and ventrally (c.54[0], 55[0]; Figs. 8–9). As preserved, the jugal is visible only in lateral or ventral views, because dorsally it is covered by the lacrimal and thus resembles the condition observed in CCNHM 1000 by *Racicot et al. (2019)*. The lacrimal is enlarged and shaped like a thick rod that covers the anterior surface of the preorbital process of the frontal; a lacrimal foramen or canal is absent (c.51[1], 52[0], 53[1]; Fig. 6, 8–9). The lacrimals are broadly visible in dorsal view as they are not covered by the maxillae as in *Olympicetus avitus*, thus resembling the condition observed in *Simocetus rayi*; ventrally their exposure is anteroposteriorly short relative to the length of the supraorbital process of the frontal (c.56[0]), but are elongated mediolaterally, forming the dorsolateral and dorsal edges of the ventral infraorbital foramen (c.58[2]), differing from *O. avitus* where they are formed by the maxilla and lacrimal.

**Parietal**—The parietals are broadly exposed in dorsal view, with no clear indication of the presence of an interparietal (c.135[0], 136[1]; Fig. 6), although it is visible in some ontogenetically young specimens that can be referred to *Olympicetus* sp. (*i.e.,* CCNHM 1000, *Racicot et al., 2019*; see discussion). In dorsal view, the anterior ends of the parietals meet the frontals along a broad V-shaped suture, with their anterolateral corners extending for a short distance along the base of the postorbital processes of the frontals, although not as far as in *Olympicetus avitus*. Posterior to the frontal-parietal suture, a low incipient sagittal crest gives the intertemporal region an ovoid cross section (c.137[0]), similar to the condition in *O. avitus* and *Simocetus rayi*. As in *O. avitus*, the parietals contact the supraoccipital along an anteriorly convex suture when viewed dorsally. The temporal surface of the parietal is flat to shallowly concave anteriorly, with a near vertical suture with the frontal (c.134[0]; Fig. 9) as it descends to form the posterior wall of the optic infundibulum; the temporal surface of the parietal becomes more inflated posteriorly and posteroventrally, where it contacts the squamosal and alisphenoid (Figs. 6, 8). The anteroventral edge of the parietal forms a semilunar notch that likely contacted part of the alisphenoid and the dorsal lamina of the pterygoid, then continuing posteriorly to form part of the subtemporal crest.

**Supraoccipital**—The anterior edge of the supraoccipital forms a semicircular arch when viewed posteriorly and dorsally, extending nearly as far anteriorly as the anterior edge of the squamosal fossa (c.140[0], 153[1]) as in *Olympicetus avitus* and *Simocetus rayi* (Figs. 6, 7A–7B). The posterior surface is incompletely preserved, but seems to have had a low external occipital crest (c.156[?1], 311[?0]). The nuchal crest is oriented dorsolaterally (c.154[1]), curving posteriorly and ventrally to meet the supramastoid crest of the squamosals (Figs. 6, 7A–7B, 8).

**Exoccipital**—The occipital condyles have a semilunar outline and are transversely and dorsoventrally convex, with sharp dorsal and lateral edges. Although the bone is poorly preserved, there is no indication for the presence of well-defined dorsal condyloid fossae (c.157[0]), differing from the condition in *Olympicetus avitus* (Figs. 7A–7B). The surfaces lateral to the condyles are shallowly convex transversely, and the paroccipital processes are broad, oriented posteroventrally to a point nearly, but not reaching the posterior edge of the condyles (c.198[2]; Fig. 6).

**Basioccipital**—The basioccipital is partially covered by part of the atlas posteriorly and hyoids posteroventrally (Fig. 7). The basioccipital crests are oriented ventrolaterally, diverging posteriorly at about an angle between 60−70° . Sediment covering the lateral surface of the crests makes it hard to determine their transverse thickness, but they seem to have been transversely narrow (c.192[0]); 195[2]), with their posteroventralmost end forming a small flange as in *Simocetus rayi* (c.194[2]; Figs. 7C–7D). No well-developed rectus capitus anticus fossa is discernible on the ventral surface (c.193[0]).

**Squamosal**—The zygomatic processes are partially eroded, more so on the left side; however, its general morphology is conserved on the right side. The processes are oriented anteriorly (c.143[0]) and seem to have been relatively long (c.189[?3]). In lateral view the dorsal edge of the zygomatic process is greatly convex dorsally (c.144[0]), whereas ventrally it is strongly concave (c.151[0]) (Fig. 8). The apex of the zygomatic process

has a transverse cleft (best preserved on the right side; c.337[1]; Fig. 8), which occurs in the type of *Olympicetus avitus* (LACM 149156) as well as in *Olympicetus* sp. (CCNHM 1000), and may be a unique feature of the genus (*Racicot et al., 2019*). Posteriorly the sternomastoid fossa is nearly absent (c.145[0]), contrasting with the deeper fossa observed in *O. avitus* and *Olympicetus* sp. 1 (see below). In dorsal view, the zygomatic process is mediolaterally broad, forming a transversely narrow and relatively shallow squamosal fossa as in *O. avitus* (c.147[1]; Fig. 6). The floor of the squamosal fossa is slightly sigmoidal, sloping gently anteroventrally towards its anterior end (c.148[1], 149[0]), and is bounded laterally and posteriorly by a fairly continuous supramastoid crest (c.150[0]), which extends medially to join the nuchal crest (Fig. 6). Medially, the squamosal plate is flat, with an interdigitated suture with the parietal that slopes anteroventrally at about 45° towards the anterior edge of the squamosal fossa and subtemporal crest and contacts the alisphenoid. Posteroventrally, the postglenoid process is long, moreso than in *Simocetus rayi* and *O. avitus*, and anteroposteriorly broad, with near parallel anterior and posterior borders that end in a squared-off ventral end (c.152[2]; Figs. 7C–7D, 8). Abaft the postglenoid process, the external auditory meatus is deep and anteroposteriorly broad (c.190[0]), bounded anteriorly by a low anterior meatal crest, that, as in *O. avitus*, seems to have formed the posterior edge of a fossa for the reception of the sigmoid process of the squamosal. The posttympanic process does not extend as far ventrally as the postglenoid process; its ventral surface is tightly sutured to the posterior process of the tympanic bulla (Figs. 7C–7D, 8). In ventral view, the glenoid fossa is poorly defined, although a very shallow, nearly indistinguishable tympanosquamosal recess occurs medially (c.179[?1,2]), as in *O. avitus* and *S. rayi*. Anteromedially the falciform process is anteroposteriorly broad with a nearly square outline (about 15 mm by 15 mm; c.177[0]), medially contacting the distal half of the anterior process of the periotic (Fig. 10C), resembling the condition observed in *Simocetus rayi*, *Archaeodelphis patrius* and basilosaurids (*Allen, 1921*; *Luo & Gingerich, 1999*; *Fordyce, 2002*; *Uhen, 2004*). In posterior view, the squamosal has a relatively narrow exposure lateral to the exoccipitals (c.146[1]; Figs. 7A–7B).

**Pterygoid**—In ventral view, the pterygoids form robust, cylindrical hamular processes that are not excavated by the pterygoid sinuses (c.173[1], 174[0]) and are separated anteriorly along the midline by a diamond-shaped exposure of the vomer, resembling the condition observed in *Simocetus rayi* (Fig. 7; *Fordyce, 2002*:fig: 4). The hamuli are long, extending posteriorly as far as the level of the middle of the zygomatic processes (c.175[3]). The dorsal lamina extends dorsally, reaching the frontal, and, judging from the preserved sutures, posteriorly, to join the parietal and alisphenoid, forming the roof of the sinus fossa as in *Olympicetus avitus* (c.166[0]; Figs. 8–9). As in *Simocetus rayi*, the ventralmost point of the pterygoid sinus fossa is at the base of the hamuli just anterior to the Eustachian notch, suggesting that the nasal passages were underlain by the sinus fossa (Figs. 7C–7D). The medial lamina forms the deep Eustachian notch, and bulges laterally at this point; posteriorly, it extends to contact the basioccipital crest. The pterygoid sinus fossa is dorsoventrally high (~45 mm) and somewhat compressed mediolaterally (~23 mm wide), extending forwards to the level of the posterior edge of the supraorbital process of the frontal (c.164[2]; Figs. 7C–7D, 8–9).

**Alisphenoid**—Only small portions of the alisphenoid can be observed on both sides. In lateral view, only a small portion of the alisphenoid is exposed on the temporal fossa, where it forms the posteromedial part of the subtemporal crest (c.142[1], 166[0]) as in other *Olympicetus* (*Vélez-Juarbe, 2017*; see below).

**Orbitosphenoid/Optic Infundibulum**—The orbitosphenoid is fused with surrounding bones, unlike the ontogenetically younger specimen of *Olympicetus avitus*. Within the optic infundibulum, the foramen rotundum and orbital fissure seem to have a similar diameter, both being transversely broader (~10 mm) than high (~6 mm) (Fig. 9), with the first located in a slightly more posteromedial position, resembling the condition in *O. avitus* (Fig. 9). However, no distinct groove for the ophthalmic artery is preserved in *Olympicetus thalassodon*, differing from *Simocetus rayi*, *O. avitus* and *Olympicetus* sp. 1 (*Fordyce, 2002*:fig.13; Figs. 8–9). The foramen rotundum opens ventrolateral to the orbital fissure, with the path for the maxillary nerve (V2) being bound ventrally by the pterygoid and palatine (Fig. 9).

**Periotic**—Only a small portion is visible on the right side. The anterior process contacts the falciform process anteriorly for about half its length. Posterior to this contact, a portion of the anterior process is visible, as is the epitympanic hiatus, which is bounded posteriorly by a prominent ventrolateral tuberosity (Fig. 10C).

**Tympanic Bulla**—Both bullae are still articulated with the cranium and mainly visible in ventral view (Fig. 10). The tympanic bullae are transversely narrow and elongated (c.252[0]), differing from the proportionately broader bullae of *Olympicetus avitus* and *O.* sp. A (see below). In ventral view, the lateral surface is more convex and the straighter medial side is gently convex anteriorly, with no indication of a spine (c.251[0]). The posterior surface of the bulla is bilobed, being divided by a broad interprominential notch (c.267[1]) that is divided by a transverse ridge (c.268[0]), differing from the bulla of *Olympicetus avitus*, but resembling that of *Olympicetus* sp. A. Both posterior prominences are level with each other (c.270[0]), the ventromedial keel forms a smooth curve posteriorly (c.253[0]), while more anteriorly it is poorly defined as the surface is nearly flat (c.274[2], 275[?0]).

A vertical, broad lateral furrow can be observed in lateral view (c.257[0], 258[0]), while more dorsally the sigmoid process curves posteriorly at its base, and is nearly vertical and perpendicular to the long axis of the bulla (c.259[0], 260[0]; Figs. 10B–10C). Although not entirely visible, the dorsal edge of the sigmoid process likely contacted the sigmoid fossa of the squamosal (c.261[?0]). The posterior process is partially visible at its contact with the posttympanic process in lateral view (c.250[0]; Figs. 7C–7D, 8, 10A–10B) and seems to have had more or less the same thickness throughout its length (c.266[0]).

**Mandible**—Left and right mandibular rami are nearly in articulation with the skull and are only missing coronoid processes and their distal ends, including the symphyseal region (Figs. 7C–7D, 8). As preserved, the mandibles are nearly straight, with their ventral border gently arching dorsally at about mid length (c.39[0], 43[1]; Figs. 7C–7D, 8), differing from the highly arched mandible of *Simocetus rayi* (*Fordyce, 2002*). Proximally, the pan bone region is transversely thin and likely formed an enlarged mandibular fossa (c.44[1]). Posterodorsally on the right side, the lateral edge of the condyle can be observed,

suggesting that its dorsal surface sits at the level of, or below, the alveolar row (c.46[1]; Fig. 8). Anteriorly, the right ramus preserves five double-rooted teeth *in-situ*, which are interpreted as representing p3-4 and m1-3, whereas the left ramus preserves three teeth that are interpreted as m1-2 and p4 (Figs. 8–9, 11–12). Multiple mental foramina are longitudinally arranged along the rami below the alveolar row; most are oval, ranging in size from 2 to 4 mm in height and up to 10 mm long, with the more posterior ones connected by a fissure as in *Olympicetus avitus* (Fig. 8; *Vélez-Juarbe, 2017*:fig. 7A).

**Dentition**—Taking a conservative approach to the tooth count, this specimen is interpreted as non-polydont as in *Simocetus rayi* (*Fordyce, 2002*), although incipient polydonty cannot be entirely ruled out, as it seems to be present on other simocetids from the eastern North Pacific (*e.g.*, LACM 140702; *Barnes, Goedert & Furusawa, 2001*). Between the teeth and alveoli, the preserved upper and lower dentition is interpreted to represent C, P1-4, M1-2 and p3-4, m1-3 (Figs. 8–9, 11–12). No conspicuous signs of tooth wear are observed in either upper or lower teeth, similar to the condition observed in *Olympicetus avitus*, and differing from that in *Simocetus rayi*, which shows signs of apical wear (*Fordyce, 2002*). The postcanine teeth are proportionately large, multicusped, transversely flattened, and nearly as high as long (c.31[1], 314[0]), resembling the condition observed in postcanine teeth of *Olympicetus avitus*, *Olympicetus* sp. 1, and *Simocetus rayi* (Figs. 8–9, 11–12). As in *Olympicetus avitus* and *Simocetus rayi*, the crowns of postcanine teeth of *O. thalassodon* have a mesiodistally concave buccal surface and are more convex lingually, with the apex of the crowns slightly recurved lingually. The bases of the crowns are ornamented with vertical striae extending apically from ecto- and entocingula, particularly on the posteriormost upper teeth (c.27[1], 32[1], 33[0]; Figs. 11–12). The crowns consist of a main apical denticle and smaller accessory denticles along the mesial and distal carinae; both apical and accessory denticles are more triangular than the more lanceolate ones observed in *O. avitus* (c.34[0]; 35[0]; Figs. 11–12; *Vélez-Juarbe, 2017*). In double-rooted teeth, the roots become fused proximally, with broad grooves on both buccal and lingual sides that extend to the base of the crown, giving it an 8-shaped cross section as in *Simocetus rayi* (*Fordyce, 2002*). In P4 and M1 the mesial root is cylindrical, tapering distally, whereas the distal root is buccolingually broader and oblong in cross section. In M2 this condition is reversed, with the mesial root being transversely broader; mesial and distal roots of the lower teeth seem to be subequal in size, both being cylindrical and tapering distally.

The anteriormost end of the right maxilla has a single alveolus (diameter = 6 mm) that curves posterodorsally and is interpreted as that of a canine, which is separated by a short interalveolar septum from two adjoining alveoli (each with a diameter ~7 mm) for a double-rooted P1 (Fig. 8, 11B). The second (P2) and third (P3) upper premolars are missing on the left side and incompletely preserved on the right; they are slightly higher than long, consisting of a main denticle with at least two accessory denticles on the mesial and distal edges, resembling teeth 'ap1' and ap2' of *O. avitus* (Fig. S1; *Vélez-Juarbe, 2017*:fig. 7D-E, Q-R). Three closely associated teeth that became disarticulated from the maxilla are still joined by matrix, and along with three other loose teeth represent left and right P4, M1-2; these have more equilateral crowns, being nearly as long as wide, with stronger lingual and labial cingula and ornamentation along the base of the crowns; the

crowns of P4 and M1 consist of a main apical denticle, with four distal and three mesial accessory denticles that diminish in size towards the base (c.328[1], 329[2]; Figs. 11E–11H, 12A–12B, 12E–12F). Their overall morphology resembles that of teeth 'mo1' and 'mo2' of *Olympicetus avitus* (Fig. S1; *Vélez-Juarbe, 2017*; fig.7M-N, Z-Aa). The second molar (M2) is the smallest of the series, and the crown is longer than tall. Its crown consists of a main apical denticle, four distal and two mesial accessory denticles, with the apices of all denticles slightly slanted distally (Figs. 11D, 11I, 12C–12D). As in *Simocetus rayi* and *Xenorophus sloanii*, the mesial and distal carinae on the upper posterior postcanines trend towards the buccal side of the teeth so that in occlusal view, the apical and accessory denticles are arranged in an arch (*Fordyce, 2002*; *Uhen, 2008*). These characteristics and other features discussed below allow for the reassignment of some of the teeth of *Olympicetus avitus*, with teeth 'mo1' and 'mo2' representing right and left M2, respectively, whereas 'ap1' and 'ap2' represent left upper premolars (Fig. S1; *Vélez-Juarbe, 2017*:fig.7). An isolated single-rooted tooth is interpreted as an upper canine or incisor (Figs. 12H, 12I). The crown is conical, with vertical striation along its lingual surface and a buccal cingulum; mesial and distal carinae seem to be present, with larger denticles along the distal carina.

The preserved lower dentition includes p3-4, m1-3, and p4, m1-2 on the right and left mandibles, respectively (Fig. 8, 11A–11C, 12C). As with the upper premolars, p3-4, m1-3 have a triangular outline of the crown in buccal or lingual views; in occlusal view the mesial and distal carinae do not trend buccally as opposed to the upper molars. Furthermore, in p3-4 and m1-2 the mesial carina has two accessory denticles (c.330[2]) that are much smaller than the apical denticle, whereas three to four accessory denticles occur along the distal carina (c.331[4]), with the apical ones being nearly as large as the apical denticle, and then diminishing in size towards the base of the crown (Fig. 8, 11A–11C, 12C). The buccal sides of the lower premolars and molars are unornamented, with only a few inconspicuous vertical striae but no prominent cingulum, while lingually striae are more prevalent, and a cingulum is present (Figs. 11A–11C, 12G). As in the upper toothrow, the last tooth, in this case m3, is the smallest in the series, seemingly lacking accessory denticles on the mesial carina and having three subequal denticles along the distal carina. As with the preceding teeth, ornamentation is nearly absent on the buccal side (Fig. 11A). An isolated tooth adjacent to the posterior end of the left maxilla and mandible may represent the left m3 (Fig. 12J). This tooth resembles the right m3, but its mesial carina is partially damaged, so it is unclear if any accessory denticles were present; its distal carina contains three denticles that diminish in size basally. The lower postcanine dentition of *Olympicetus thalassodon* appears to be characterized by having less conspicuous ornamentation on the buccal side, and more vertically aligned carinae. Based on these characteristics the lower dentition of *Olympicetus avitus* is reinterpreted as follows: teeth 'pp1-4' represent left p3-m2, while 'pp5', 'pp7', and 'pp6' represent p3, p4, and m1 from the right side (Fig. S1; see also *Vélez-Juarbe, 2017*: fig.7F-G, J, L, S-T, W, Y).

**Hyoid**—Most of the hyoid elements are preserved in LACM 158720, including the basihyal, stylohyals and thyrohyals (Figs. 13A–13C). The basihyal has a rectangular, blocky outline, with both lateral ends expanded, forming broad, quadrangular rugose surfaces for the articulation of the paired elements (stylo- and thyrohyals). The mid portion is subtriangular

in cross-section, and the dorsal surface is shallowly concave transversely. The partial left thyrohyal obscures the posteroventral surface of the bone. The partial left and the complete right thyrohyals and stylohyals are preserved (Figs. 13A–13C). The thyrohyals are not fused to the basihyal and are fairly straight, with a transversely oval cross section at mid-length; overall they are shorter but more robust than the stylohyals, and not flattened, wing-like as in extant mysticetes and odontocetes (c.338[0]; Fig. 13). The proximal articular surface has a rectangular outline, and the surface is rugose and shallowly convex. Distally, the shaft is twisted, so that the distal articular surface is nearly perpendicular to the long axis of the proximal surface. The distal articular surface has a more oval outline that is rugose and shallowly convex. The stylohyals are long and slender, and the right stylohyal is nearly in articulation with the paroccipital process (Figs. 13A–13B). Along the long axis they are bowed laterally, with the shaft having a more flattened, oval cross-section along its length, with both, proximal and distal ends expanded, being overall, nearly identical to the stylohyoid of *Olympicetus avitus* (*Vélez-Juarbe, 2017*). The proximal end is transversely expanded with a nearly flat, rugose articular surface. Distally, the shaft becomes twisted, so that the distal end is offset at about 45° from the proximal articular surface. The lack of fusion between the thyrohyal and basihyal, and the cylindrical shape of the thyrohyal resembles the condition observed in basilosaurids (*e.g.*, *Dorudon atrox* (*Andrews, 1906*), and *Cynthiacetus peruvianus Martínez-Cáceres & de Muizon, 2011*; *Uhen, 2004*; *Martínez-Cáceres, Lambert & de Muizon, 2017*) and some stem mysticetes (*e.g.*, *Mammalodon colliveri Pritchard, 1939*, *Fucaia buelli Marx, Tsai & Fordyce, 2015*, and *Mystacodon selenensis Lambert et al., 2017*; *Fitzgerald, 2010*; *Muizon et al., 2019*), whereas in more derived odontocetes (*e.g.*, *Brygmophyseter shigensis* (*Hirota & Barnes, 1995*), *Kogia breviceps* (*Blainville, 1838*), *Albireo whistleri Barnes, 1984*, *Kentriodon nakajimai Kimura & Hasegawa, 2019*, and *Tursiops truncatus* (*Montagu, 1821*); Figs. 13D–13G) these bones are partially or completely fused, and the thyrohyals tend to be more flattened and plate- or wing-like (*Reidenberg & Laitman, 1994*; *Hirota & Barnes, 1995*; *Barnes, 2008*; *Johnston & Berta, 2011*; *Kimura & Hasegawa, 2019*).

**Cervical Vertebrae**—The atlas, axis and C3-7 are partially preserved and unfused (c.279[0], 280[0]; Fig. 14; Table 2). The dorsal arch of the atlas has a low, blunt mid-dorsal ridge that extends nearly the whole length of the arch. The vertebral foramen is broken, although it seems to have occupied the same position as that of *Olympicetus avitus* (*Vélez-Juarbe, 2017*). The anterior articular facets are obscured because the atlas is still attached to the skull, while the posterior facets have a reniform outline and form a dorsoventrally elongate, smooth, flat surface that extends dorsal to the articulation for the odontoid process (Fig. 14A). On the ventral arch, the hypapophysis that would have articulated with the odontoid process is short as in *O. avitus* and unlike the longer, more robust process of Simocetidae gen. et sp. A, and *Echovenator sandersi* (*Churchill et al., 2016*). The transverse processes are oriented slightly posterolaterally and are divided by a broad, rounded notch into a larger, more robust dorsal process and a smaller, knob-like ventral process (c.278[2]; Fig. 14A). The neural canal has an oval outline.

The axis is missing the dorsal arch. The odontoid process is short and blunt. The anterior articular surface has a subtriangular outline and is flat to shallowly concave, extending

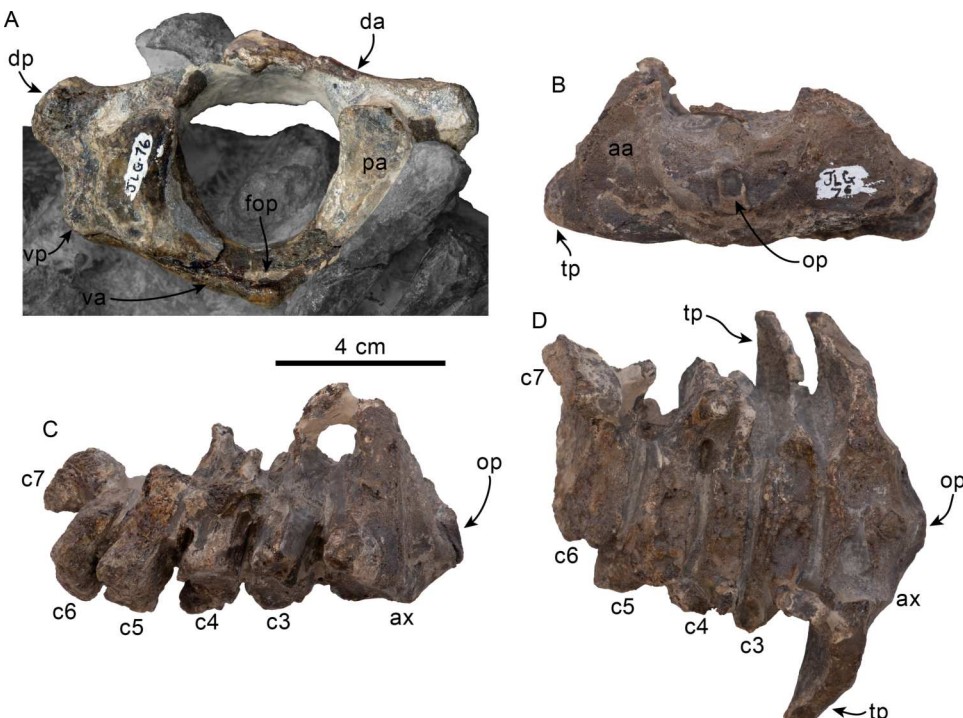

**Figure 14** **Cervical vertebrae of *Olympicetus thalassodon* sp. nov. (LACM 158720).** (A) atlas in posterior view; (B) axis in anterior view; (C) axis and third through seventh cervicals in right lateral view; (D) axis and third through seventh cervicals in dorsal view. Abbreviations: aa, anterior articular surface; ax, axis; c3-7, third through seventh cervical vertebrae; da, dorsal arch; dp, dorsal process; fop, facet for odontoid process; op, odontoid process; pa, posterior articular surface; tp, transverse process; vp, ventral process.

anteroventrally and being continuous with the ventral surface of the odontoid process (Fig. 14B). The transverse processes are oriented posterolaterally, with a triangular outline when viewed anteriorly. Their ventral surface is anteroposteriorly broad, forming a flat surface that faces ventrally and slightly posteriorly, with a sharp anterior edge (Figs. 14B–14D). Dorsomedially, the posterior surface of the transverse process forms a relatively deep, concave surface. Cervicals 3–6 are missing their dorsal arches and transverse processes for the most part, while only a small portion of C7 is preserved. The centra are anteroposteriorly flat and slightly wider than high; the epiphyses are unfused (Figs. 14C–14D). The right transverse process of C3 is partially preserved, and its morphology is similar to that of the axis.

**Remarks**—*Olympicetus thalassodon* represents an adult individual, in contrast with the other specimens of *Olympicetus* thus far described, which represent neonatal (LACM 126010, CCNHM 1000) and subadult (LACM 149156, LACM 124105) individuals (*Vélez-Juarbe, 2017*; *Racicot et al., 2019*). This could potentially raise the question whether *O. thalassodon* represents an adult individual of *O. avitus* or *Olympicetus* sp. 1 (described in detail below). However, *O. thalassodon* differs from *O. avitus* and *Olympicetus* sp. 1 by characters that do not seem to be the result of differences between individuals of the same

species or ontogenetic stage. For example, *O. thalassodon* differs from other *Olympicetus* by having a larger, more elongate tympanic bulla (Table 3). Nevertheless, ontogenetic variation can be ruled out to explain this difference because odontocetes show precocial development of the tympanic bullae (*Buffrénil, Dabin & Zylberberg, 2004*; *Lancaster et al., 2015*). Other characteristics, such as the number of denticles in the carinae of upper and lower molars, can also be ruled out as resulting from ontogenetic or intraspecific variation. These taxa can further be differentiated from each other by morphological characters of the orbital region, such as the arrangement of the bones that form the dorsolateral edge of the ventral infraorbital foramen, the height of the orbit relative to the lateral edge of the rostrum, and the composition of the posterior wall of the antorbital notch.

*OLYMPICETUS* sp. 1
(Figs. 15–20; Tables 1, 3 and 6)

**Material**—LACM 124105, partial skull, including two partial teeth, left tympanic bulla and right periotic; missing distal end of rostrum, zygomatic arches, parts of the neurocranium and mandible. Collected by J. L. Goedert December 17, 1983.
**Locality and horizon**—LACM Loc. 5123, Murdock Creek, Clallam Co., Washington State, USA (48°09′25″N, 123°52′10″W). See above for additional information from this locality.
**Formation and age**—Pysht Formation, between 30.5–26.5 Ma (Oligocene: late Rupelian-early Chattian; *Prothero, Streig & Burns, 2001a*; *Vélez-Juarbe, 2017*).
**Temporal and geographic range**—Oligocene of Washington, USA.

## Description

The description is based solely on LACM 124105 and will focus on morphological characters that differentiate it from *Olympicetus avitus* and *O. thalassodon*. As with the type of *Olympicetus avitus*, LACM 124105 seems to represent a subadult individual, showing some partially open sutures, such as the basisphenoid-presphenoid suture. Multiple areas of the skulls show evidence of erosion (*e.g.*, rostrum, skull roof), likely as a result of wave action, because specimens from this locality are usually recovered as concretions along the beach.
**Premaxillae**—Only part of the left ascending process of the premaxilla is preserved (Fig. 15). The ascending process borders the external nares as it ascends towards the vertex (c.74[0]); however, its incomplete preservation posterior to the nasals does not permit identification of its posteriormost extent. A relatively deep sulcus extends along its anterior border, which is consistent with the placement and morphology of the posterior extent of the posterolateral sulcus in *Olympicetus avitus* (c.73[2]; Figs. 15 and 17; *Vélez-Juarbe, 2017*).
**Maxilla**—Only part of the rostral portion of the maxilla is preserved (Figs. 15–18). Ventrally, the palatal surface is incompletely preserved along the midline and along the alveolar rows; however, the parts that are preserved indicate that it was transversely convex, with the alveolar rows slightly more elevated dorsally (Fig. 17). Posteriorly, the contact between the maxillae and palatines seems to have been triangular to anteriorly bowed (c.20[?0], 21[1]; Fig. 16) as in other *Olympicetus*. The alveolar rows, although incompletely

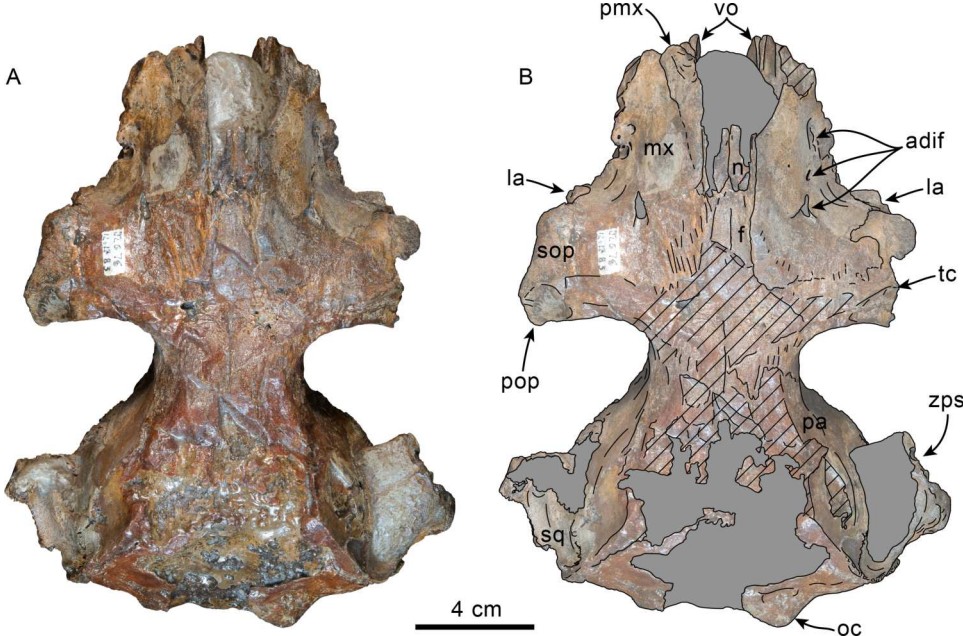

**Figure 15 Dorsal view of skull of *Olympicetus* sp. 1 (LACM 124105).** Unlabeled (A) and labeled (B) skull in dorsal view. Diagonal lines denote broken surfaces, gray shaded areas are obscured by sediment. Abbreviations: adif, anterior dorsal infraorbital foramina; f, frontal; la, lacrimal; mx, maxilla; n, nasal; oc, occipital condyle; pa, parietal; pmx, premaxilla; pop, postorbital process; sop, supraorbital process of frontal; sq, squamosal; tc, temporal crest; vo, vomer; zps, zygomatic process of squamosal.

preserved, diverged posteriorly and had at least three pairs of closely-spaced, double-rooted postcanine teeth (c.23[0], 26[0]). Based on the preserved posterior border of the alveolar row, it seems that at least a short maxillary infraorbital plate was present (c.60[1]; Fig. 17). In posteroventral view, the ventral infraorbital foramen has an oval outline (~12 mm wide by nine mm high); its dorsolateral, ventral, and ventromedial edges are defined by the maxilla, and its dorsomedial edge is defined by the frontal (c.58[0], [59[0]).

In dorsal view, the rostrum seems to have been fairly wide (c.7[1]; Fig. 15). Dorsally, at the base of the rostrum, the maxilla faces dorsolaterally and is shallowly convex to flat as it ascends over the supraorbital processes of the frontal; thus as in other species of *Olympicetus*, it lacks a rostral basin (c.66[0]; Fig. 15). At the base of the rostrum, at least three anterior dorsal infraorbital foramina range in diameter between 2–5 mm, with a fourth, more posterior foramen, dorsomedial to the antorbital notch (c.65[3]; Figs. 16–18). The maxillae are eroded at the level of the antorbital notches, so it is uncertain if these formed part of the posterior wall of the notch as in *Olympicetus avitus*. The ascending process of the maxilla partially covers the supraorbital process of the frontal, extending posteriorly and posteromedially beyond the anterior half of the process, coming into contact with the nasal process of the frontal near the midline and forming a gently sloping surface towards the edge of the orbit, but not reaching its lateral border (c.49[0], 77[1], 78[2], 79[0], 80[0], 130[0], 308[1]; Fig. 15).

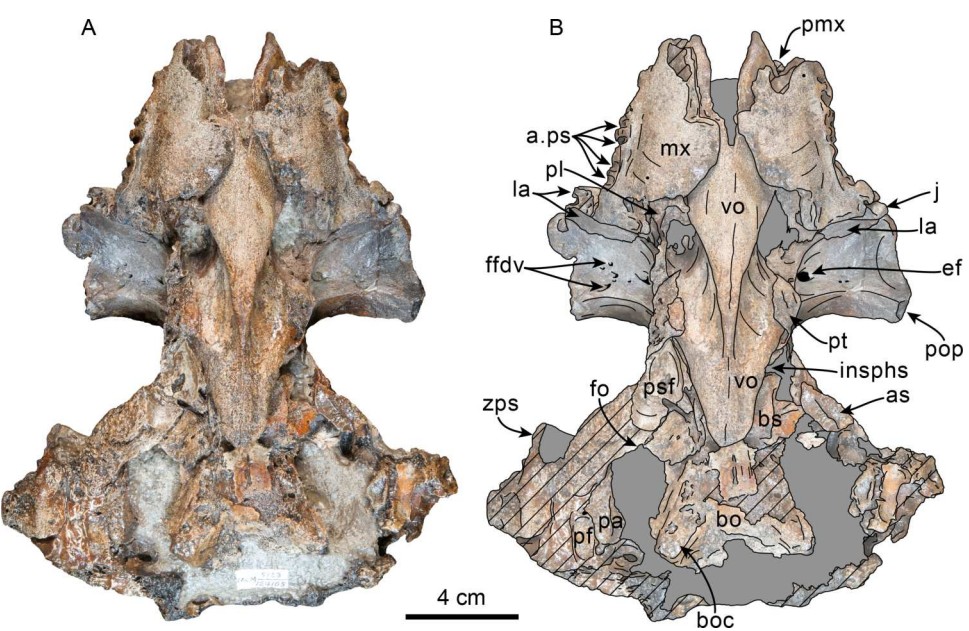

**Figure 16  Ventral view of skull of *Olympicetus*. sp. 1 (LACM 124105).** Unlabeled (A) and labeled (B) skull in ventral view. Diagonal lines denote broken surfaces, gray shaded areas are obscured by sediment. Abbreviations: a.ps, alveoli for postcanine teeth; as, alisphenoid; bo, basioccipital; boc, basioccipital crest; bs, basisphenoid; ef, ethmoid foramen; ffdv, foramina for frontal diploic veins; insphs, intersphenoidal synchondrosis; j, jugal; la, lacrimal; mx, maxilla; pa, parietal; pf, periotic fossa; pl, palatine; pmx, premaxilla, pop, postorbital process; psf, pterygoid sinus fossa; pt, pterygoid; vo, vomer; zps, zygomatic process of squamosal.

**Vomer**—The vomer is mostly missing anterior to the antorbital notches and eroded anteroventrally; nevertheless, it is evident that it formed the lateral and ventral surfaces of the mesorostral canal. Ventrally, the vomer likely was exposed through a diamond-shaped window towards the posterior end of the palate as in other simocetids (Fig. 16). Dorsal and posterodorsal to this point the vomer forms the nasal septum, forming the medial walls of the choanae. From the posterior palatal exposure, the vomer gently slopes posterodorsally to form a triangular, horizontal plate extending over the still open, basisphenoid-presphenoid suture, but not reaching as far posterior as the fused basisphenoid/basioccipital contact (c.191[0]; Fig. 16). The horizontal plate of the vomer contacts the dorsal laminae of the pterygoids along its anterolateral ends (Figs. 16–18).

**Palatine**—Only some very small fragments of the right palatine are preserved. Posterodorsally, a fragment of lateral surface of the palatine reaches the frontal, forming part of the infundibulum for the sphenopalatine and infraorbital foramina as well as the posterior border of a round (~5 mm diameter) sphenopalatine foramen (Fig. 18). The infundibulum has an oval outline, being broader than high (20 mm × 10 mm), and is bounded dorsally by the frontal and lacrimal, and the maxilla ventrally and ventrolaterally (Fig. 18).

**Nasal**—Although incompletely preserved, the nasals seem to have been the highest point of the vertex, were longer than wide and dorsoventrally thin, as in other simocetids (c.114[0],

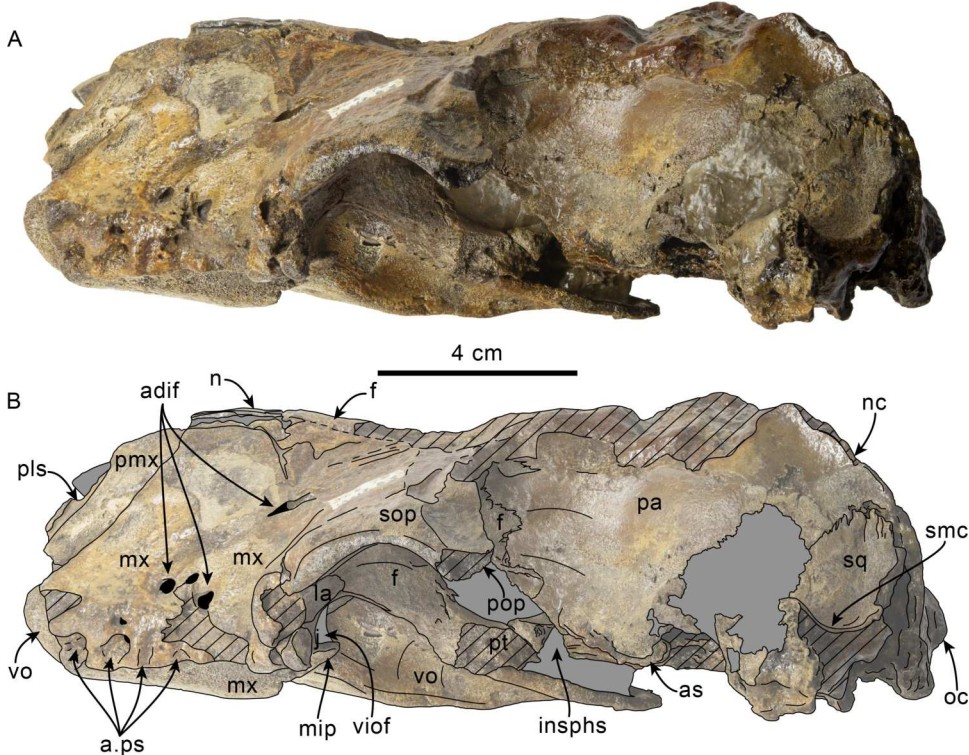

**Figure 17 Left lateral view of skull of *Olympicetus.* sp. 1 (LACM 124105).** Unlabeled (A) and labeled (B) skull in left lateral view. Diagonal lines denote broken surfaces, gray shaded areas are obscured by sediment. Abbreviations: a.ps, alveoli for postcanine teeth; adif, anterior dorsal infraorbital foramina; as, alisphenoid; f, frontal; j, jugal; la, lacrimal; mip, maxillary infraorbital plate; mx, maxilla; n, nasal; nc, nuchal crest; oc, occipital condyle; pa, parietal; pmx, premaxilla; pop, postorbital process; pt, pterygoid; smc, supramastoid crest; sop, supraorbital process; sq, squamosal; viof, ventral infraorbital foramen; vo, vomer.

116[0], 118[?0], 124[0], 125[0], 312[0]; Figs. 15 and 17). Along their posterior borders, the nasals are separated by the narrow, narial processes of the frontals (Fig. 15). The anterior edges of the nasals are incompletely preserved, but extended far forward of the anterior edge of the supraorbital processes, whereas posteriorly it seems that they reach a level in line with the anterior edge of the supraorbital processes (c.81[3], 123[0]; Fig. 15).

**Frontal**—As in other *Olympicetus*, a wedge-shaped exposure of the frontals occurs along the midline, surrounded by the maxillae laterally and nasals anteriorly, although poor preservation of the surrounding bones does not allow precise determination of the size of this exposure relative to the nasals (Fig. 15). Along the midline, the bone is poorly preserved, although it does seem that the frontals are lower than the nasals, preserving the saddle-like profile (in lateral view) seen in other species of *Olympicetus*. Posteriorly, the frontal-parietal suture seems to have been broadly V-shaped dorsally, and sinusoidal in the temporal region, with no extension of the parietals into the supraorbital processes. Laterally, the supraorbital processes slope very gently ventrolaterally (c.47[?0]; Fig. 17). Dorsally, the maxillae only partially cover the supraorbital processes, leaving the preorbital and postorbital processes broadly exposed dorsally (Fig. 15). Anteroventrally, the preorbital

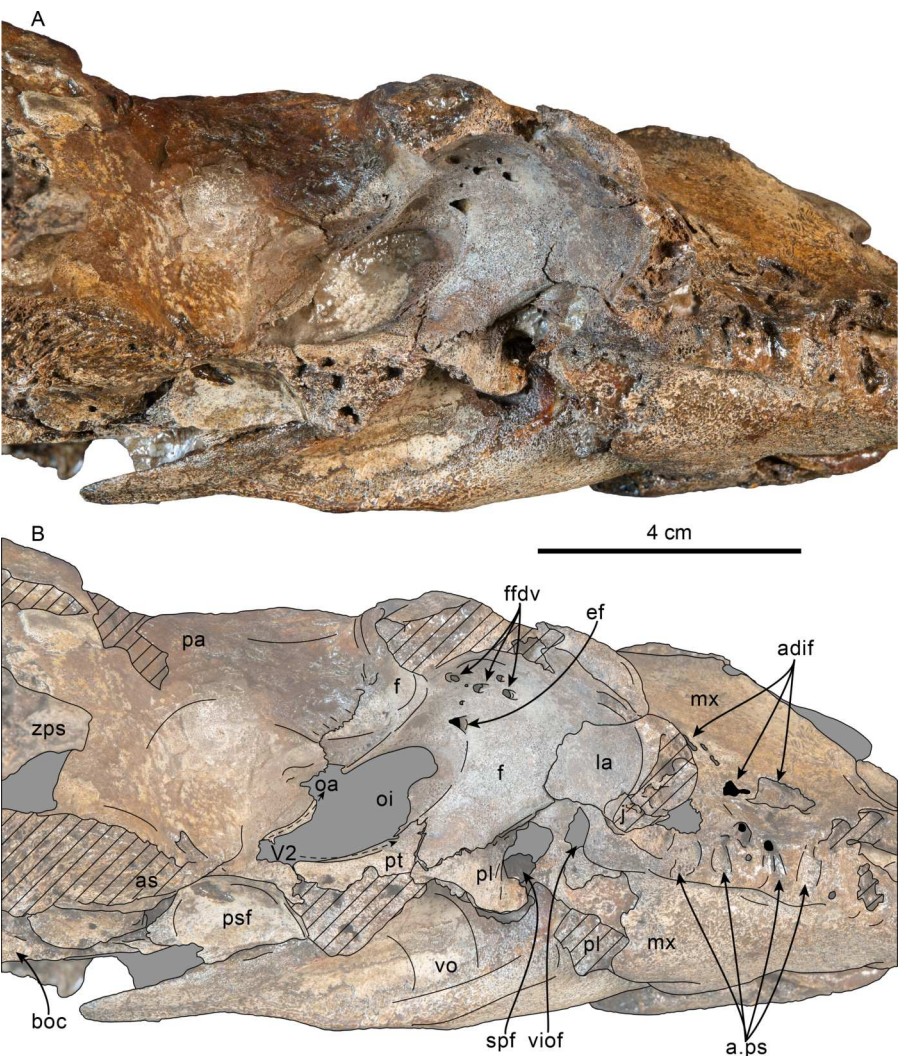

**Figure 18  Ventrolateral view of skull of *Olympicetus* sp. 1 (LACM 124105).** Unlabeled (A) and labeled (B) skull in right ventrolateral view focusing on the features of the orbital region. Diagonal lines denote broken surfaces, gray shaded areas are obscured by sediment. Abbreviations: a.ps, alveoli for postcanine teeth; adif, anterior dorsal infraorbital foramina; as, alisphenoid; boc, basioccipital crest; ef, ethmoid foramen; ffdv, foramina for frontal diploic veins; f, frontal; j, jugal; la, lacrimal; mx, maxilla; oa, path for ophthalmic artery; oi, optic infundibulum; pa, parietal; pl, palatine; psf, pterygoid sinus fossa; pt, pterygoid; spf, sphenopalatine foramen; viof, ventral infraorbital foramen; V2, path for maxillary nerve; vo, vomer; zps, zygomatic process of squamosal.

process contacts the lacrimal. The postorbital processes are incompletely preserved, but seem to have been relatively short, robust, and oriented posteroventrolaterally (Figs. 15 and 17). In lateral view the dorsal edge of the orbit is highly arched but positioned at a lower position (c.48[1]; Fig. 17) relative to the lateral edge of the rostrum than is observed in *Olympicetus avitus* or *O. thalassodon*. A low and sharp temporal crest extends anterolaterally from near the frontal/parietal suture and into the posterodorsal and dorsal surface of the supraorbital process (c.132[2]; Fig. 15), differing from the condition in other *Olympicetus*.

Ventrally, the frontal contacts the lacrimal anteroventrally and the maxilla and/or palatine more medially, resulting in the frontal forming part of the posterodorsal edge of the infundibulum for the ventral infraorbital and sphenopalatine foramina (Figs. 16 and 18). The optic foramen is partially covered by sediment; its general orientation seems to be anterolateral, with its posterior border being defined by a low, but sharp infratemporal crest (c.63[0]). Similar to other simocetids, a small (~3 mm diameter) ethmoid foramen is anterolateral to the optic foramen and is accompanied by four to five smaller (1–2 mm) foramina located along the dorsolateral roof of the orbit (Figs. 16 and 18).

**Lacrimal + Jugal**—Only a small portion of the jugal is preserved, but it is evident that it was not fused with the lacrimal (c.54[0], 55[0]; Figs. 17–18). The portion of the jugal that is preserved is stout and cylindrical, tapering medially and wedged between the lacrimal and maxilla, which excludes it from forming part of the ventral infraorbital foramen (Figs. 17–18). The lacrimal is large, and rod-like, broadly visible in dorsal and lateral views, but with a proportionately small ventral exposure (c.51[1], 56[0]). It contacts the preorbital process of the frontal anteroventrally, tapering medially, and seems to have been exposed anteriorly, forming part of the posterior wall of the antorbital notch but not extending dorsally onto the supraorbital process (c.52[0]; Fig. 15, 17–18).

**Parietal**—The parietals are exposed dorsally but badly eroded (c.135[0], 136[?]; Fig. 15). The parietals contact the frontals along a broad, V-shaped suture, but differ from the condition seen in other species of *Olympicetus* in that they do not extend into the base of the supraorbital processes. In cross section through the intertemporal region, the parietals seem to have an ovoid outline (c.137[?1]), resembling the condition in *Olympicetus avitus*. Along the temporal surface the parietal becomes more inflated posteriorly towards its contact with the squamosal and alisphenoid (Figs. 17–18). Ventrally, the parietal has an internal projection that contacts the squamosal medial to the periotic fossa, constricting the cranial hiatus as in other simocetids (c.184[2]; Fig. 16).

**Supraoccipital**—The supraoccipital is only partially preserved, with the exception of its dorsolateral borders. The nuchal crests are sharp, directed dorsolaterally, and only slightly overhanging the temporal fossae (c.154[1]; Fig. 15), and curving posteroventrally to join the supramastoid crests of the squamosals.

**Exoccipital**—The exoccipital is poorly preserved. Dorsal to the remaining parts of the right occipital condyle is what seems to be a shallow dorsal condyloid fossa (c.157[?1]). The surface lateral to the condyles is flat to shallowly convex.

**Basioccipital**—As preserved, the basioccipital crests seem to have been relatively thick transversely (c.192[?1]) and oriented posterolaterally, at about an angle of 45 degrees (c.195[3]; Fig. 16). The rest of the ventral surface is incompletely preserved.

**Squamosal**—The zygomatic processes are incompletely preserved. Posteromedially, the sternomastoid fossa forms a distinct emargination that is overhung dorsally by the supramastoid crest, much more than in *Olympicetus avitus* (c.145[1]; Fig. 15). The supramastoid crest seems to have been continuous with the nuchal crest (c.150[0]; Fig. 17). The squamosal plate contacts the parietal along an anteroventrally sloping interdigitated suture, meeting the alisphenoid to form part of the subtemporal crest (Fig. 17). Ventrally,

the squamosal is heavily eroded and only a small portion of the periotic fossa is preserved, where it contacts the medial extension of the parietal (Fig. 16).

**Pterygoid**—Most of the pterygoid is missing on both sides of the skull. A portion of the dorsal lamina extends posterodorsally towards the parietal and contributes to the posteroventral edge of the optic infundibulum as in *Olympicetus avitus* (Figs. 17–18). As preserved, the pterygoid sinus fossa is anteroposteriorly longer than wide and is located entirely anterior to the foramen ovale (c.164[2], 169[0]; Figs. 16 and 18).

**Alisphenoid**—As seen in *Olympicetus avitus*, the alisphenoid forms the posterodorsal surface of the pterygoid sinus fossa (Figs. 16 and 18). The medial and posterior ends of the bone are incompletely preserved or eroded on both sides, making it difficult to determine the position of the alisphenoid-squamosal suture or the path of the mandibular nerve (V3). On the temporal wall, the exposure of the alisphenoid is limited to a small sliver, because it is mostly overlapped by the parietal and the squamosal (c.142[1]; Figs. 17–18).

**Basisphenoid**—Posteriorly the basisphenoid is fused with the basioccipital, and anteriorly its suture to the presphenoid (sphenoidal synchondrosis) is still open, resembling the growth stage of the type of *Olympicetus avitus* (*Vélez-Juarbe, 2017*). The ventral surface is flat and covered by the horizontal plate of the vomer (Fig. 16).

**Optic Infundibulum**—The optic infundibulum is a slightly sinusoidal opening bounded by the frontal anteriorly and dorsally, parietal posteriorly, pterygoid ventrally and anteroventrally (Fig. 18). The optic foramen, orbital fissure and foramen rotundum are still partly covered by sediment. The frontal forms most of the borders of the optic foramen anterodorsally, whereas posteroventrally the foramen rotundum was bounded laterally by the parietal and floored by the pterygoid. The anteroventral edge of the parietal that forms part of the infundibulum has a narrow groove that trends anterodorsally and would have carried the ophthalmic artery, resembling the condition in *Simocetus rayi* and *Olympicetus avitus* (Fig. 18; *Fordyce, 2002*; *Vélez-Juarbe, 2017*). Along the ventral edge of the infundibulum, the pterygoid has a distinct but shallow groove that would have presumably carried the maxillary nerve (V2), extending along its dorsolateral surface and diverging slightly over its lateral surface anteriorly (Fig. 18).

**Malleus**—The left malleus is still attached with the corresponding tympanic (Fig. 19). The head has a semicircular outline, with paired facets for articulation with the incus that are oriented at about 90 degrees to each other; the more anterior facet is about twice as large as the posterior one, as in *Olympicetus avitus* (Fig. 19; *Vélez-Juarbe, 2017*). The tubercule is relatively large, nearly as long as the head (c.199[0]; Fig. 19). The manubrium is prominent, with its apex forming a slightly recurved muscular process (Fig. 19). The anterior process is fused laterally to the tympanic, dorsally forming a continuous surface with the mallear ridge. Meanwhile, the ventral edge of the anterior process is shelf-like and together with the mallear ridge forms a deep, narrow sulcus for the chorda tympani (Figs. 19A, 19C, 19E).

**Tympanic Bulla**—Only the left tympanic bulla is preserved (Fig. 19) but missing its posterior process. Overall it closely resembles in size and morphology that of *Olympicetus avitus* (*Vélez-Juarbe, 2017*). In dorsal or ventral view, the bulla has a heart-shaped outline, being relatively short and wide (c.252[1]), unlike the larger and transversely narrower bulla of *Olympicetus thalassodon* (Figs. 10, 19). The lateral surface of the tympanic bulla is broadly

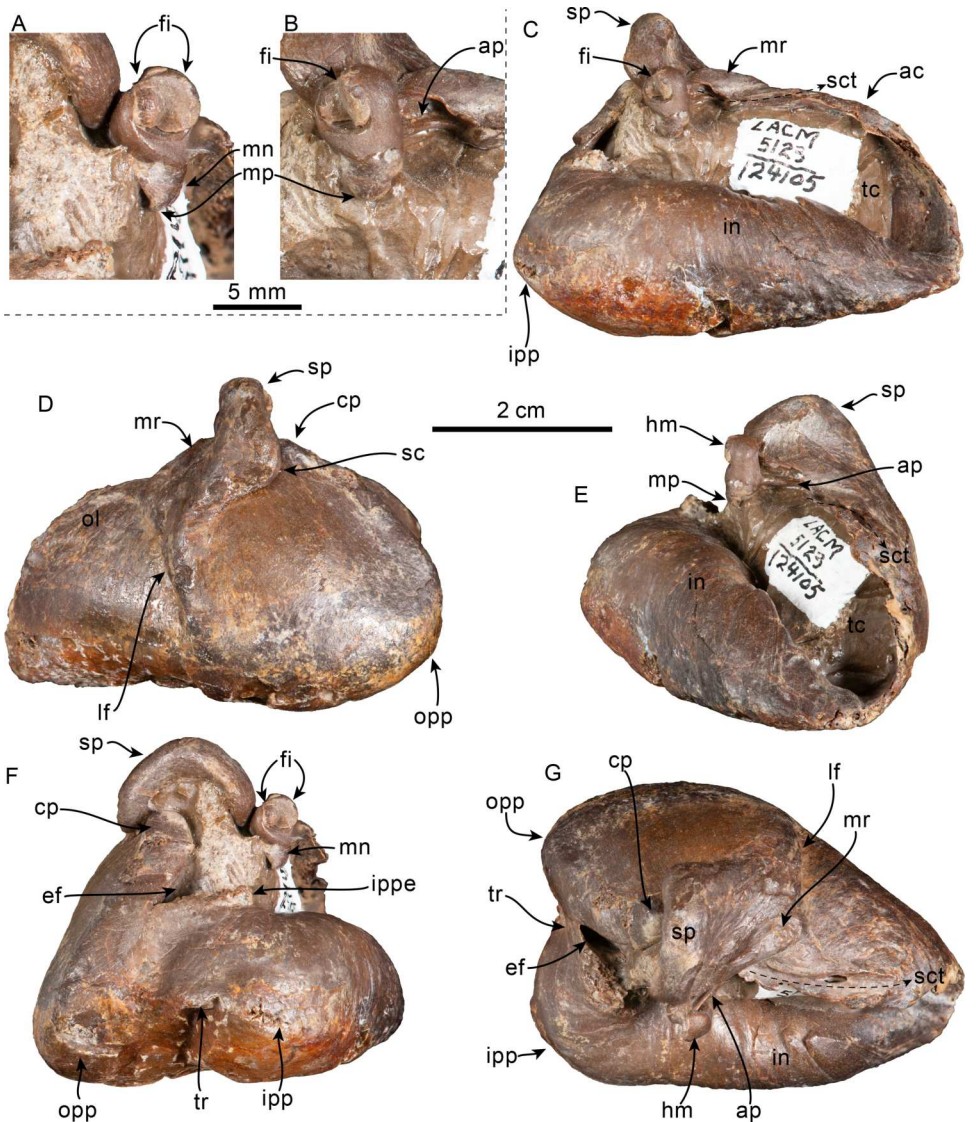

**Figure 19 Malleus and tympanic bulla of *Olympicetus* sp. 1 (LACM 124105).** Left malleus in posterior (A) and medial (B) views. Left malleus and tympanic bulla in medial (C), lateral (D), anterior (E), posterior (F), and dorsal (G) views. Abbreviations: ac, anterodorsal crest; ap, anterior process; cp, conical process; ef, elliptical foramen; fi, facet for incus; hm, head of malleus; in, involucrum; ipp, inner posterior prominence; ippe, inner posterior pedicle; lf, lateral furrow; mn, manubrium; mp, muscular process; mr, mallear ridge; ol, outer lip; opp, outer posterior prominence; sc, sigmoid cleft; sct, sulcus for chorda tympani; sp, sigmoid process; tc, tympanic cavity; tr, transverse ridge.

convex, whereas the medial surface is straight; the posterior prominences give the bulla a bilobed outline posteriorly, but anteriorly, the lateral surface converges medially more steeply than the medial surface along a smooth curve. There is no indication of the presence of an anterior spine (c.251[0]). Posteriorly, a broad interprominential notch extends from the level below the elliptical foramen, continuing along the ventral surface of the bulla as a short, shallow median furrow for only about a third of its length (c.267[0]). The

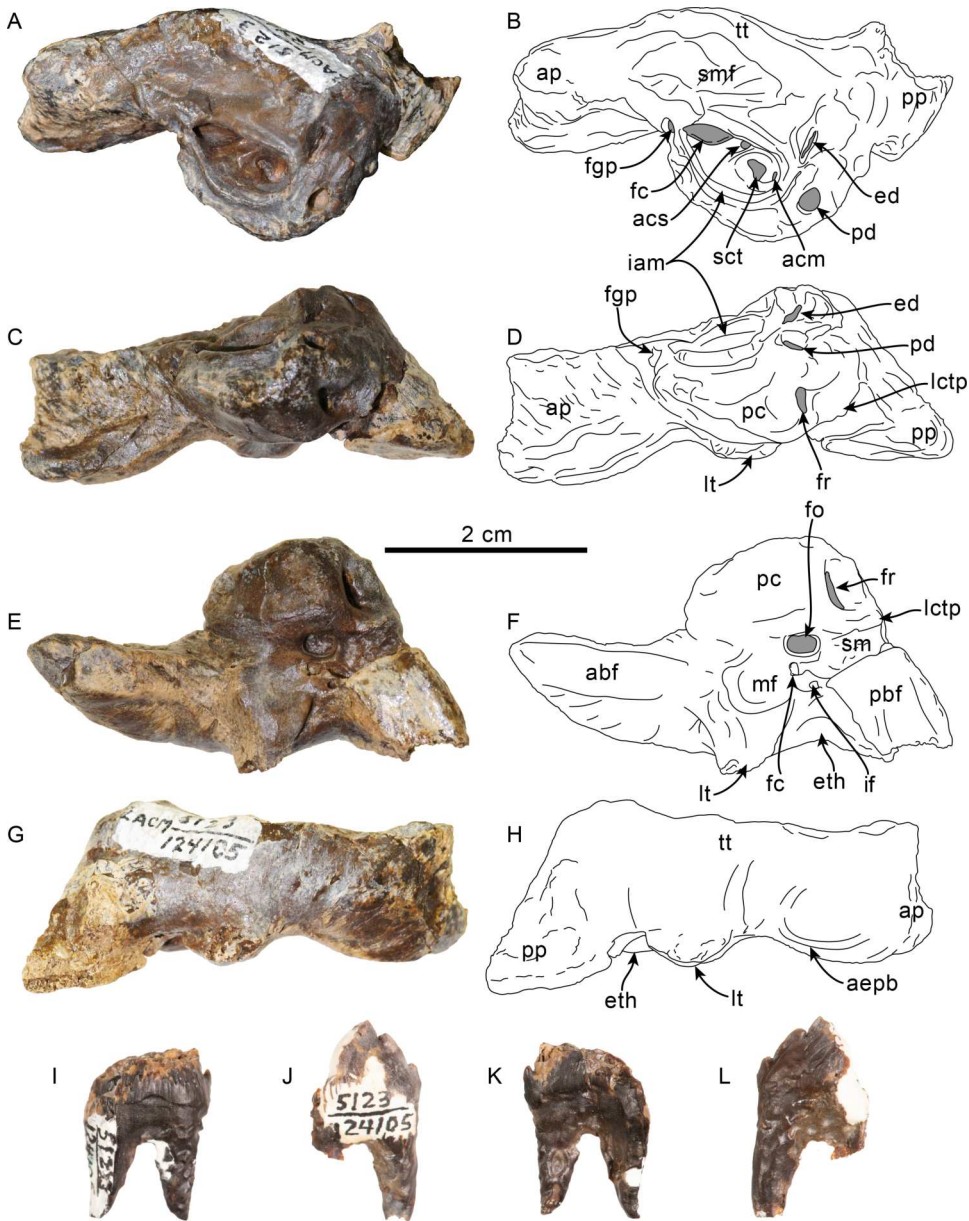

**Figure 20** **Periotic and teeth of *Olympicetus* sp. 1 (LACM 124105).** Unlabeled and labeled right periotic in dorsal (A–B), medial (C–D), ventral (E–F), and lateral (G–H) views. Postcanine teeth in buccal (I–J) and lingual (K–L) views. Abbreviations: abf, anterior bullar facet; acm, area cribrosa media; acs, area cribrosa superior; aepb, anteroexternal+parabullary sulcus; ap, anterior process; ed, aperture for endolymphatic duct; eth, epitympanic hiatus; fc, facial canal; fgp, foramen for greater petrosal nerve; fo, fenestra ovalis; fr, foramen rotundum; iam, internal acoustic meatus; if, incudal fossa; lctp, lateral caudal tympanic process; pbf, posterior bullar facet; pc, pars cochlearis; pd, aperture for perilymphatic duct; lt, lateral tuberosity; mf, mallear fossa; pp, posterior process; sct, spiral cribriform tract; sm, stapedial muscle fossa; smf, suprameatal fossa; tt, tegmen tympani.

interprominential notch is divided by a transverse ridge (c.268[0]; Fig. 19D), resembling the condition observed in *Olympicetus thalassodon*, and differing from that of *O. avitus*, which does not have an interprominential ridge. The inner and outer prominences extend posteriorly to nearly the same level (c.270[0]). The ventromedial keel is poorly defined, forming a smooth curve around the posterior part of the involucrum, its posteromedial surface just slightly bulging farther medially than the rest of the involucrum (c.253[0], 274[2], 275[0], 276[0]). The elliptical foramen seems to have been narrow, and nearly vertical (c.262[0]).

In lateral view, the ventral edge of the bulla is nearly flat (c.269[0]), differing from the more broadly concave ventral margin observed in some xenorophids, like *Albertocetus meffordorum* (*Uhen, 2008*). The lateral furrow is nearly vertical, forming a relatively broad sulcus (c.257[0], 258[0]; Fig. 19B). Dorsally, the sigmoid process is vertical and perpendicular to the long axis of the bulla (c.259[0]), with its posterior edge curving anteriorly along a smooth curve (c.260[0]). The mallear ridge extends obliquely from the anteromedial base of the sigmoid process towards the dorsalmost extension of the lateral furrow. A narrow, dorsally open sulcus for the chorda tympani extends anteriorly for a length of 17 mm along the dorsomedial edge of the outer lip, originating at the junction between the anterior process of the malleus and the mallear ridge (Figs. 19A, 19C, 19E). The anterodorsal crest descends steeply towards the anterior edge of the bulla.

In medial view the dorsal and ventral edges of the involucrum gradually converge towards the anterior end of the bulla (c.271[0]; Fig. 19A). The involucrum has numerous, faint vertical ridges (c.272[1]), differing from the deeper grooves observed in xenorophids, like *Albertocetus meffordorum* (*Uhen, 2008*).

**Periotic**—Only the right periotic is preserved (Figs. 20A–20H) and is overall very similar to that of *Olympicetus* sp. (CCNHM 1000) described by *Racicot et al. (2019)*. The anterior process is oriented anteriorly and short relative to the length of the pars cochlearis, with its anteroventral and anterodorsal ends being bluntly pointed and together giving it a nearly squared-off outline in medial or lateral view (c.201[0], 202[0], 204[2]; Figs. 20C–20D). In medial or lateral view, the anterior process is deflected ventrally to a point below the ventral edge of the pars cochlearis (c.203[1]; Figs. 20C–20D). The anteroventral surface of the anterior process forms a slightly convex to flat ventral surface (c.205[0]; Figs. 20C–20D). In lateral view, at the base of the anterior process is a shallow, C-shaped sulcus that begins near the anteroventral edge, curves posteroventrally towards the lateral tuberosity, then curves anterodorsally; it is interpreted as a combined anteroexternal+parabullary sulcus (sensu *Tanaka & Fordyce, 2014*; Figs. 20G–20H). This condition resembles that of other early odontocetes such as *Waipatia maerewhenua Fordyce, 1994*, and *Notocetus vanbenedeni Moreno, 1892*, but differs from others like *Otekaikea marplesi* (*Dickson, 1964*) where these sulci are separate, and from the much deeper sulcus in *Papahu taitapu Aguirre-Fernández & Fordyce, 2014* (*Tanaka & Fordyce, 2014*; *Viglino et al., 2022*). In cross-section, the anterior process is ovoid, being dorsoventrally taller (~14 mm) than mediolaterally wide (~9 mm) (c.209[1]). The anterior part of the ventral surface of the anterior process has as well-defined anterior bullar facet (c.210[3]; Figs. 20E–20F). Posterior to the anterior bullar facet, the fovea epitubaria forms a smooth curve that is interrupted by a prominent lateral

(ventrolateral) tuberosity (c.212[1]). The lateral tuberosity has a triangular outline in ventral view but does not extend as far laterally as in other stem odontocetes such as *Cotylocara macei* (*Geisler, Colbert & Carew, 2014*), being instead barely visible in dorsal view. A broadly arched epitympanic hiatus lies posterior to the lateral tuberosity and anterior to the base of the posterior process (c.213[1]). Posteromedial to the epitympanic hiatus, is a small (diameter: ~2 mm) rounded fossa incudis, while anterior to it and medial to the lateral tuberosity is a broad (diameter: ~6 mm), circular mallear fossa (c.214[1], 215[0]; Figs. 20E–20F). The lateral surface of the periotic is generally smooth with the exception of the posterior process, whose lateral surface is rugose (c.217[2]; Figs. 20G–20H). Medially, the anterior process is separated from the cochlea by a well-defined groove (anterior incisure, *sensu Mead & Fordyce, 2009*) that extends anterodorsally, and marks the origin for the tensor tympani muscle (c.218[1]).

In dorsal view, a low crest delimits laterally the dorsal surface of the periotic; it extends from the low pyramidal process towards the anterodorsal spine of the anterior process (Figs. 20A–20B). Medial to this crest is an elongated depression, the suprameatal fossa, which is about 13.5 mm long by 7 mm wide, and around 1.5 mm deep (Figs. 20A–20B). The fundus of the internal acoustic meatus is funnel-shaped, with an oval outline, delimited by a low ridge (c.235[0]; 236[0]). The area cribrosa media (*sensu Mead & Fordyce, 2009*; *Orliac et al., 2020*; = inferior vestibular area of *Ichishima, Kawabe & Sawamura, 2021*) and the spiral cribiform tract are separated by a very low ridge, these two are in turn separated from the area cribrosa superior (previously called the foramen singulare, *Orliac et al., 2020*; = superior vestibular area of *Ichishima, Kawabe & Sawamura, 2021*) by a low transverse crest that lies about 3 mm below the upraised rim of the internal acoustic meatus, while it is separated from the dorsal opening of the facial canal by a ridge that is slightly lower (~4 mm from the edge of the rim) (c.237[2]; Figs. 20A–20B). The proximal opening of the facial canal has an oval outline and is located anterolateral to the spiral cribriform tract (c.238[0], 239[1]). Anterodorsally it is bridged, forming a "second" foramen, which is smaller and rounded (Figs. 20A–20D), resembling the condition observed in other early odontocetes such as *Waipatia maerewhenua*, and similarly, is interpreted as the foramen for the greater petrosal nerve (*Fordyce, 1994*). The aperture for the endolymphatic duct (vestibular aqueduct) is slit-like (~4 mm long by 1 mm wide) and located posterolateral to the internal acoustic meatus, just below the more vertical posterior surface of the pyramidal process and separated from the fenestra rotunda by a very wide distance (c.230[3]; Figs. 20A–20D). In contrast, the aperture for the perilymphatic duct (cochlear aqueduct) is rounded (diameter = 3 mm) and located posteromedial to the internal acoustic meatus and medial to the aperture for the endolymphatic duct, and broadly separated from the fenestra rotunda (c.228[1], 229[2]). A small, curved depression posteroventral to the aperture for the endolymphatic duct is interpreted as a shallow stylomastoid fossa (c.225[1]). The dorsomedial surface of the cochlear portion has a shallow depression that accentuates the raised medial rim of the internal acoustic meatus. In medial view, the cochlea is dorsoventrally thin (maximum height ~11 mm), its ventromedial surface is anteroposteriorly convex, and a low, faint ridge extends along its ventrolateral end (c.221[0]; Figs. 20C–20F). In ventral view, the cochlear portion has a subrectangular

outline (c.219[1], 220[1], 222[1]). Posteriorly, the fenestra rotunda is located towards the lower half of the posterior surface, and it is wider than high (4 × 2 mm), with a kidney-shaped outline (c.223[0]). Posterolateral to the fenestra rotunda, the lateral caudal tympanic process projects farther posteriorly than the rest of the posterior surface of the cochlea, although it is not as prominent as that of other simocetids (*i.e.,* CCNHM 1000; *Racicot et al., 2019*). Its ventral and posterior borders intersect along a curved edge (c.226[1]; Figs. 20C–20F). Ventrally, the fenestra ovalis is longer than wide (4 × 3 mm) and located towards the posterior half of the cochlea. The ventral opening of the facial canal (~2 mm in diameter) is lateral to the fenestra ovalis and is separated by a sharp crest. The facial canal opens posteroventrally and continues as a groove that merges with the stapedial muscle fossa at the base of the posterior process; the fossa is deep and rounded, with its posterodorsal edge nearly in line with the fenestra rotunda (c.224[0]).

The posterior process is short and robust, with its long axis oriented posterolaterally (c.246[1], 247[1], 249[0]; Figs. 20A–20B, 20E–20F). Proximally, the lateral surface of the posterior process is rough, with an irregular, near vertical ridge interpreted here as a poorly-developed articular rim (c.240[1]), resembling the condition in other simocetids (*i.e.,* CCNHM 1000) and early odontocetes like *Notocetus vanbenedeni*, and differing from the more prominent articular rim observed in platanistids (*Muizon, 1987*; *Racicot et al., 2019*; *Viglino et al., 2022*; Figs. 20A–20B). The dorsal edge of the posterior process has a linear profile (c.248[0]). The posterior bullar facet has a kite-shaped outline; its surface is smooth and shallowly concave transversely (c.242[0], 243[0]); the edges of the facet are sharp, with the exception of the posteromedial edge which is rounder (c.244[0]).

**Dentition**—Only two incompletely preserved teeth are associated with LACM 124105 (Figs. 20I–20L). Both are postcanine teeth, with striated enamel, and ecto- and entocingula and at least two denticles along the mesial carina (c.27[1], 32[1] 33[0], 35[?1]). On both teeth, one of the surfaces is concave, which resembles the condition observed on the buccal side of upper postcanine teeth of other simocetids (*e.g., Olympicetus thalassodon*). The roots are long and conical, becoming fused proximally. Tooth PCa (Fig. 20I, 20K) measures 12 mm long (mesiodistally) by 6 mm wide (buccolingually), and tooth PCb (Fig. 20J, 20L) measures 9 mm high and 6 mm wide (buccolingually).

**Remarks**—LACM 124105 shares multiple diagnostic features with the other named species of *Olympicetus*, such as having a temporal fossa that is broadly open dorsally, unfused lacrimal/jugal (c.54[0]), lacking a maxillary foramen (c.76[0]; = posterior dorsal infraorbital foramen), and maxilla covering only about the anterior half of the supraorbital process of the frontal (c.77[1]). However, it does differ by having a more sharply defined infratemporal crest, the orbit at a lower position relative to the edge of the rostrum (c.48[1]; Fig. 17), the dorsolateral edge of the ventral infraorbital foramen formed by the maxilla (c.58[0]), and more notably, the lateral end of the temporal crest extending along the posterodorsal surface of the supraorbital process of the frontal (c.132[2]; Fig. 15). These differences are considered to be species-related, and not the result of ontogenetic change as this specimen shows a similar growth stage as the type of *Olympicetus avitus* (LACM 149156; *Vélez-Juarbe, 2017*). Nevertheless, because of its incomplete preservation, it is preferably left in open nomenclature until better material belonging to this taxon is identified.

**Table 6  Dimensions of simocetid periotic.** Measurements (in mm) of periotic of *Olympicetus* sp. 1 (LACM 124105) (modified from *Kasuya, 1973*; *Racicot et al., 2019*).

| | |
|---|---|
| Maximum length | 43 |
| Proximal dorsoventral thickness of anterior process | 12 |
| Length of anterior process | 16 |
| Transverse width of anterior process at mid-length | 9 |
| Dorsoventral height of anterior process at mid-length | 13 |
| Maximum width of periotic | 22 |
| Least distance between fundus of internal auditory meatus and aperture for endolymphatic foramen | 2 |
| Least distance between fundus of internal auditory meatus and aperture for perilymphatic foramen | 3 |
| Least distance between fenestra rotunda and endolymphatic foramen | 7 |
| Least distance between fenestra rotunda and perilymphatic foramen | 3 |
| Length of posterior bullar facet | 11 |
| Width of posterior bullar facet | 8 |
| Transverse width of cochlear portion | 10 |
| Anteroposterior length of cochlear portion | 15 |

## Results of the phylogenetic analysis

The phylogenetic analysis resulted in four most parsimonious trees, 3,691 steps long, with retention index (RI) = 0.518 and consistency index (0.181). Other statistical values are shown in the strict consensus tree (Fig. 21, Fig. S2). Based on these results, Simocetidae now seems to form a monophyletic group that consists of *Simocetus rayi*, CCNHM 1000 (*Olympicetus* sp.), *Olympicetus* sp. 1, *Olympicetus avitus*, *O. thalassodon*, and Simocetidae gen. et sp. A (LACM 124104) (Fig. 21, Fig. S2).

## DISCUSSION

Although particular attention has been paid to Oligocene mysticetes from the North Pacific over the last few decades (*e.g.*, *Barnes et al., 1995*; *Okazaki, 2012*; *Marx, Tsai & Fordyce, 2015*; *Peredo & Pyenson, 2018*; *Solis-Añorve, Gozález-Barba & Hernández-Rivera, 2019*; *Hernández Cisneros, 2022*; *Hernández Cisneros & Nava-Sánchez, 2022*), the same cannot be said with regards to the odontocetes. Oligocene odontocetes from around the North Pacific are not entirely missing from the scientific literature and have been mentioned multiple times, often identified informally as "non-squalodontid odontocetes", "agorophiid" or "*Agorophius*-like" (see *Whitmore Jr & Sanders, 1977*; *Goedert, Squires & Barnes, 1995*; *Barnes, 1998*; *Barnes, Goedert & Furusawa, 2001*; *Fordyce, 2002*; *Hernández Cisneros, González Barba & Fordyce, 2017*). However, given their importance, most of these have yet to be properly described, and our understanding of species richness and relationships between Oligocene odontocetes from the North Pacific is not fully understood. More importantly, these early odontocetes can potentially advance our understanding of

the origins and early diversification of odontocetes, as well as acquisition of some of their distinguishing features, such as echolocation.

The first of these taxa to be described was *Simocetus rayi* from the early Oligocene (33.7–30.6 Ma) Alsea Fm. of Oregon, which was placed in its own family, Simocetidae, and is currently one of the geologically oldest named odontocetes (*Prothero et al., 2001b*; *Fordyce, 2002*). Since then, only two other North Pacific Oligocene odontocetes have been named, specifically, the platanistoid *Arktocara yakataga* from the Oligocene Poul Creek Fm. in Alaska, which may be amongst the earliest crown odontocetes, and the stem odontocete *Olympicetus avitus* from the Pysht Fm. in Washington (*Boersma & Pyenson, 2016*; *Vélez-Juarbe, 2017*). More recently, *Racicot et al. (2019)* described a neonatal skull (CCNHM 1000) from the Pysht Fm. in Washington, which closely resembles *Olympicetus avitus* but did not group with *Simocetus rayi* nor with *O. avitus*. Instead, all three taxa occupied different positions outside of crown odontocetes (*Racicot et al., 2019*). Other potential Oligocene odontocetes include the squaloziphiid *Yaquinacetus meadi Lambert, Godfrey & Fitzgerald, 2018*, and the platanistoid *Perditicetus yaconensis Nelson & Uhen, 2020*, both from the latest Oligocene to early Miocene Nye Mudstone, but more precise chronostratigraphic resolution would be needed to determine their precise age.

Herein, the description of three additional specimens from the mid-Oligocene Pysht Formation in Washington have potentially clarified the relationship between stem odontocetes from the North Pacific. The results (Fig. 21, Fig. S2) show a more inclusive Simocetidae, differing from earlier analyses (*e.g.*, *Vélez-Juarbe, 2017*; *Racicot et al., 2019*) where *Simocetus* and *Olympicetus* occupied different positions within stem odontocetes. Furthermore, the phylogenetic analysis recovered CCNHM 1000 as part of the Simocetidae, differing from the analysis of *Racicot et al. (2019)*, where it was recovered at the base of a clade including all odontocetes, with the exception of Xenorophidae. As discussed by *Racicot et al. (2019)*, CCNHM 1000 does resemble *Olympicetus avitus*; more specifically, based on the new specimens described here, it shares with *Olympicetus* spp. closely-spaced posterior buccal teeth (c.26[0]), buccal teeth with ecto- and entocingula (c.32[1], 33[0]), presence of a small maxillary infraorbital plate (c.60[1]), and the presence of a transverse cleft on the apex of the zygomatic process (c.337[1]), amongst others. However, CCNHM 1000, does show some dental characteristics that set it apart from *O. avitus* as discussed by *Racicot et al. (2019)*, and others that differentiate it from other specimens of *Olympicetus*, such as presence of an interparietal (c.136[0]), a more anterior position of the apex of the supraoccipital (c.140[1]), and a very low nuchal crest (c.154[2]). Some of these characters, such as the position of the apex of the supraoccipital and the morphology of the nuchal crest are also observed in the neonate skull (LACM 126010) referred to *O. avitus*, suggesting that these characters change ontogenetically, with neonatal individuals displaying more plesiomorphic conditions. Along these same lines, the presence of a distinct interparietal in CCNHM 1000, most likely another ontogenetic feature, is interpreted in the present phylogenetic analysis as a plesiomorphic character, which when combined with the other ontogenetic characteristics mentioned previously, may account for the more basal position of CCNHM 1000 in the phylogenetic analysis (Fig. 21). Besides this, it seems clear that CCNHM 1000 should be regarded as a neonate of *Olympicetus* sp.

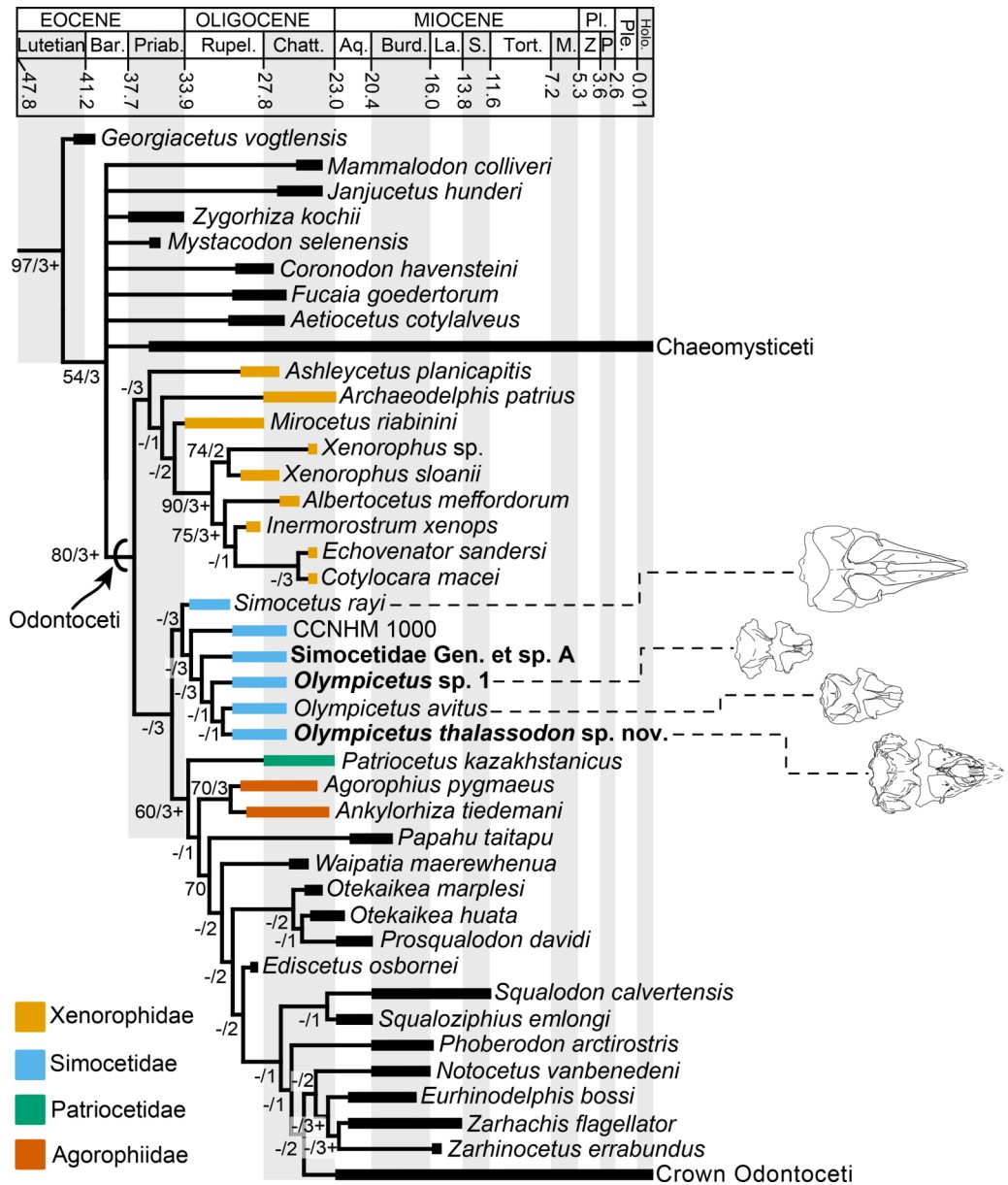

**Figure 21** **Time calibrated phylogeny of Cetacea.** Phylogenetic tree showing relationship between Simocetidae with other odontocetes; Chaeomysticeti and crown Odontoceti clades are pruned. Strict consensus tree based on four most parsimonious trees of length = 3691, with retention index (RI) = 0.518, and consistency index (CI) = 0.181. Temporal ranges for taxa follow *Lloyd & Slater (2021)* and *Sander et al. (2021)*. The numbers at the nodes indicate bootstrap/decay index values. Abbreviations: Aq., Aquitanian; Bar., Bartonian; Burd., Burdigalian; Chatt., Chattian; Holo., Holocene; La., Langhian; M., Messinian; P, Piacenzian; P., Pliocene; Ple., Pleistocene; Priab., Priabonian; Rupel., Rupelian; S., Serravalian; Tort., Tortonian; Z, Zanclean. Time scale based on *Cohen et al. (2013)*.

The inclusion of CCNHM 1000 has some interesting implications for Simocetidae. *Racicot et al. (2019)* described the inner ear morphology of CCNHM 1000, showing that it does not have the capability of ultrasonic hearing, which is suggestive that other taxa within

this clade are also non-echolocating odontocetes, at least as neonates. Future studies on the inner ear morphology of the periotics of other simocetids of more advanced ontogenetic stages, such as specimens of *Simocetus rayi*, *Olympicetus thalassodon*, *Olympicetus* sp. (LACM 124105), as well as those of other simocetids that will be described in future works, such as USNM 244226 (*Olympicetus* sp.), USNM 205491 (Simocetidae gen. et sp. nov.), and LACM 140702 (Simocetidae gen. et sp. nov.), will likely provide more information to this regard.

## Stem Odontocetes from the North Pacific

The early odontocete clade Simocetidae now includes six OTUs: *Simocetus rayi*, *Olympicetus avitus*, *Olympicetus* sp. (LACM 124105), *O. thalassodon* (LACM 158720), Simocetidae gen. et sp. A (LACM 124104) and CCNHM 1000 (Fig. 21). All specimens, with the exception of *S. rayi*, are from the Pysht Fm., with four of them, LACM 124104, LACM 124105, LACM 158720 and CCNHM 1000, coming from the same general area (LACM Locs. 5123 and 8093). The results of the phylogenetic analysis resemble those of an earlier, preliminary study that also recovered a monophyletic Simocetidae composed of most of the OTUs used here as well as a few others undescribed specimens from the eastern North Pacific, but that also recovered *Ashleycetus planicapitis*, from the early Oligocene of South Carolina, as part of that clade (*Vélez-Juarbe, 2015*). In contrast, the results of the present work suggest that Simocetidae represents an endemic radiation of North Pacific stem odontocetes, that parallels that of the Aetiocetidae in the same region (*Hernández Cisneros & Vélez-Juarbe, 2021*), and the Xenorophidae (here considered to include Ashleycetidae and Mirocetidae; Fig. 21) in the North Atlantic and Parathethys (*Marx, Lambert & Uhen, 2016a*). Interestingly, simocetids and xenorophids overlap temporally with some platanistoids such as *Arktocara yakataga* and *Waipatia* spp. (*Fordyce, 1994*; *Tanaka & Fordyce, 2015*; *Boersma & Pyenson, 2016*; *Tanaka & Fordyce, 2017*; *Gaetan, Buono & Gaetano, 2019*; *Viglino et al., 2021*; but see *Viglino et al., 2022* with regards to *W. maerewhenua*). This suggests that crown odontocetes appeared at least by the late Oligocene, pending a more precise assessment of the age or *A. yakataga*, and that the initial diversification of odontocetes may have occurred during the latest Eocene to early Oligocene. This is further supported by the early Rupelian (33.7–30.6 Ma; *Prothero et al., 2001b*) age of the Alsea Fm., where *Simocetus rayi* was found, which places Simocetidae amongst, if not the earliest, diverging odontocete clade (pending a better age assessment for *Mirocetus riabinini*; *Sanders & Geisler, 2015*). The discovery and description of additional odontocetes from the Makah, Pysht, and Lincoln Creek formations in Washington State, and Alsea and Yaquina formations in Oregon, would likely provide new insights with regards to early odontocete diversification. This highlights the importance of the fossil record of the North Pacific towards further understanding the early history and radiation of odontocetes.

At present, there are no published accounts of simocetids from the western North Pacific, although these are expected to be present based on the occurrence of closely-related marine tetrapods in Oligocene deposits on both sides of the basin (*e.g.*, plotopterids, desmostylians, aetiocetids; *Olson, 1980*; *Domning, Ray & McKenna, 1986*; *Ray, Domning & McKenna, 1994*; *Olson & Hasegawa, 1996*; *Inuzuka, 2000*; *Barnes & Goedert, 2001*; *Sakurai,*

*Kimura & Katoh, 2008*; *Ohaski & Hasegawa, 2020*; *Mayr & Goedert, 2016*; *Mayr & Goedert, 2022*; *Mori & Miyata, 2021*; *Hernández Cisneros & Vélez-Juarbe, 2021*), which makes this apparent absence an interesting question. However, some records from Japan bear close resemblance to simocetids and should be analyzed further. These include a mandible with two cheek teeth (KMNH VP 000011) and an isolated tooth (KMNH VP 000012) referred by *Okazaki (1988)* to *Squalodon* sp. from the Oligocene Waita Formation of the Ashiya Group. The general morphology of the mandible (KMNH VP 000011) resembles *Olympicetus thalassodon* and other basal odontocetes with multi-cusped cheek teeth, such as *Prosqualodon davidis Flynn, 1947*, and *Waipatia maerewhenua*. In these taxa the dorsal surface of the mandibular condyle is at about the same level as the horizontal ramus and the ventral border is relatively straight (*Flynn, 1947*; *Fordyce, 1994*). Furthermore, the two cheek teeth preserved with KMNH VP 000011 are much more like those of *Olympicetus*, with the more anterior tooth (B3 in *Okazaki, 1988*) having only a small accessory denticle along the base of the mesial carina, while three larger denticles are observed distally, that increase in size apically, greatly resembling the premolars of *O. thalassodon* (Figs. 11A, 11C, 12G). Meanwhile, the second tooth (B7 in *Okazaki, 1988*) resembles the m3 of *Olympicetus thalassodon*, by being smaller than the more anterior teeth, and having three accessory denticles along the distal carina that diminish in size towards the base of the crown, lacking accessory denticles along the mesial carina, and little to no ornamentation on the buccal side. The isolated tooth (KMNH VP 000012) resembles cheek tooth 'pp4' of *Olympicetus avitus* (reinterpreted above as the left m2), as they are relatively low and long, with multiple accessory denticles along the mesial and distal carinae, as well as having lingual and buccal cingula (*Okazaki, 1988*; *Vélez-Juarbe, 2017*). One distinguishing character is that the accessory denticles of *Olympicetus* spp. and the Waita Fm. odontocetes are closer in size to the main cusp than those of other basal odontocetes with multi-cusped cheek teeth. For example, lower cheek teeth of *Squalodon calvertensis Kellogg, 1923b*, *Prosqualodon davidis*, *P. australis Lydekker, 1894*, *Phoberodon arctirostris Cabrera, 1926*, and *Waipatia* spp. do have accessory denticles along their distal edges, but those are much smaller than the main cusp (*Flynn, 1947*; *Fordyce, 1994*; *Tanaka & Fordyce, 2015*; *Gaetan, Buono & Gaetano, 2019*; *Viglino et al., 2019*). The combination of these morphological features suggests that the specimens described by *Okazaki (1988)* could be considered as aff. *Olympicetus* sp., although this requires confirmation by direct observation of the specimens. Other cetaceans from the Ashiya Group include the toothed mysticete *Metasqualodon symmetricus Okazaki, 1982*, from the Waita Fm., considered to represent an aetiocetid or a more basal mysticete outside Aetiocetidae, and the eomysticetid *Yamatocetus caniliculatus Okazaki, 2012*, from the Jinnobaru Fm. (*Okazaki, 1987*; *Okazaki, 1994*; *Fitzgerald, 2010*; *Geisler et al., 2017*).

Similarly, other potential records of simocetids are found in the late Oligocene El Cien Formation of Baja California Sur. *Hernández Cisneros, González Barba & Fordyce (2017)* briefly discussed two skulls from the El Cien Fm., comparing one with *Simocetus rayi* and the other with an undescribed skull (USNM 205491) from the Alsea Fm.; they may represent other undescribed simocetids. These odontocetes from El Cien Fm. are currently under study (A. E. Hernández-Cisneros, pers. comm., 2021), and other described taxa from this formation include kekenodontids, aetiocetids, eomysticetids, and other

stem mysticetes (*Hernández Cisneros & Tsai, 2016*; *Hernández Cisneros, González Barba & Fordyce, 2017*; *Solis-Añorve, Gozález-Barba & Hernández-Rivera, 2019*; *Hernández Cisneros, 2022*; *Hernández Cisneros & Nava-Sánchez, 2022*). These records from the Jinnobaru Fm. and El Cien Fm., resemble the odontocete assemblage of the Pysht Fm., which includes simocetids, aetiocetids and other early mysticetes, and it is therefore likely that simocetids would be present in these units as well (*Barnes et al., 1995*; *Peredo & Uhen, 2016*; *Vélez-Juarbe, 2017*; *Shipps, Peredo & Pyenson, 2019*; *Hernández Cisneros & Vélez-Juarbe, 2021*; this work).

## Dentition and feeding in simocetids

As in most other groups of stem odontocetes (*e.g.*, xenorophids, agorophiids), simocetids have an heterodont dentition, but do seem to have a more conservative tooth count, closer to that of basilosaurids such as *Cynthiacetus peruvianus* (*Martínez-Cáceres & de Muizon, 2011*), which consists of three incisors, one canine, four premolars, two upper and three lower molars, a pattern that is also observed in early mysticetes like *Janjucetus hunderi* Fitzgerald, 2006, and *Mystacodon selenensis* (*Fitzgerald, 2010*; *Lambert et al., 2017*). While the tooth count of some simocetids is hard to interpret (*e.g.*, *Olympicetus avitus*; *Vélez-Juarbe, 2017*), others such as *Simocetus rayi* and *Olympicetus thalassodon* offer more definite clues with regards to their dentition. In the case of *Simocetus rayi*, its tooth count seems to be secondarily reduced from the plesiomorphic condition through the loss of the upper incisors, while the lower ones are retained (*Fordyce, 2002*). Although most are not preserved in the holotype, the teeth of *S. rayi* were widely separated and small (when compared to those of *Olympicetus*). In contrast, the teeth of *Olympicetus thalassodon* are closely spaced, and based on the preserved teeth and alveoli, the dental formula of the latter is tentatively interpreted as ?I3, C, P4, M2/?i3, c, p4, m3. The presence of three incisors is based in part on LACM 140702, although there is also the possibility that *O. thalassodon* had no incisors, resembling the condition of *S. rayi*. Nevertheless, if these interpretations are correct, then the dentition of simocetids is the most plesiomorphic amongst odontocetes, paralleling that of early mysticetes. This would contrast with xenorophids, which seem to have a polydont dentition; for example, *Xenorophus sloanii* and *Echovenator sandersi* both have a significantly higher count of postcanine teeth (*Sanders & Geisler, 2015*; *Churchill et al., 2016*). However, the dentition of many xenorophids is still unknown, including key taxa, such as *Archaeodelphis patrius*, which may offer additional insight into early odontocete dental evolution.

Although different simocetids seem to share similar conservative tooth counts and generalized features of their teeth, there are some interesting differences between some of the species. One conspicuous difference between the dentition of *Olympicetus avitus* and *O. thalassodon* is the presence of a "carnassial"-like tooth in the former (Fig. S1; tooth 'mo3' in *Vélez-Juarbe, 2017*:fig.7O,Bb). This tooth is distinguished from all other postcanine teeth by having a lingual lobe with a secondary carina with accessory denticles that descends lingually from the apex (Fig. 13E), while its root is expanded lingually, giving the impression of the presence of three roots (mesial, distal and lingual), rather than two (mesial and distal) as in the other postcanine teeth. Meanwhile, a third, lingual root seems to be present in

the P4 of *Simocetus rayi* (*Fordyce, 2002*), in an unnamed *Simocetus*-like taxon from the Lincoln Creek Fm. (*Barnes, Goedert & Furusawa, 2001*) and in LACM 124104 (described above), and could be a character that is shared among some simocetids, although better preserved specimens are needed to corroborate this. The presence of a third, lingual root and a lingual lobe is otherwise unknown in other odontocetes, toothed mysticetes, and basilosaurids (*Uhen, 2004*; *Martínez-Cáceres, Lambert & de Muizon, 2017*), but present in more basal forms (*e.g.*, protocetids and kekenodontids; *Kellogg, 1936*; *Kassegne et al., 2021*; *Corrie & Fordyce, 2022*). A somewhat similar crown morphology is observed in protocetids such as *Indocetus ramani Sahni & Mishra, 1975*, *Aegyptocetus tarfa Bianucci & Gingerich, 2011*, and *Togocetus traversei Gingerich & Cappetta, 2014*, as well as in *Kekenodon onamata Hector, 1881*, all of which have a protocone lobe supported by a lingual root in the more posterior upper premolars and molars (*Bajpai & Thewissen, 2014*; *Kassegne et al., 2021*; *Corrie & Fordyce, 2022*). However, the lobe on the lingual side of the teeth of protocetids and *K. onomata* is located distolingually, differing from the condition observed in *O. avitus* and LACM 124104, in which the lobe is located mesiolingually, and may thus not be homologous. Interestingly, tooth B7 (*sensu Sanders & Geisler, 2015*) of *Xenorophus sloani* seems to present a more inconspicuous version of the "carnassial" tooth of simocetids this tooth occupies a position similar to that of P4 in *Simocetus rayi*, and this character should be explored further as more specimens become available.

Some of the morphological characters observed in described simocetids, such as the arched palate, short and broad rostrum, smaller and widely-spaced teeth, as in *Simocetus rayi*, were interpreted as features of a bottom suction feeder (*Fordyce, 2002*; *Werth, 2006*; *Johnston & Berta, 2011*). *Olympicetus* shares some of these features, such as the arched palate. However, *O. thalassodon*, has closely spaced, larger teeth, as well as a relatively gracile, unfused hyoid apparatus (Figs. 11–13A–13C; *Johnston & Berta, 2011*; *Viglino et al., 2021*; *Werth & Beatty, 2023*), which suggest that this taxon was instead a raptorial or combined feeder (Fig. 22). Taking this into account, it is likely that simocetids employed different methods of prey acquisition, likely akin to the amount of variation observed in other contemporaneous groups, such as xenorophids, which include taxa with long narrow rostra (*e.g.*, *Cotylocara macei*; *Geisler, Colbert & Carew, 2014*) that can be interpreted as raptorial feeders, as well as a brevirostrine suction feeding taxon (*i.e.*, *Inermorostrum xenops*; *Boessenecker et al., 2017*). Thus it seems that several methods of prey acquisition evolved iteratively across different groups of odontocetes soon after their initial radiation (*Hocking et al., 2017*; *Kienle et al., 2017*).

# CONCLUSIONS

Three new specimens of odontocetes from the early to late Oligocene Pysht Formation were described herein, further increasing our understanding of richness and diversity of early odontocetes, specially for the North Pacific region. Inclusion of this new material in a phylogenetic analysis showed that Simocetidae is a much more inclusive clade, which besides *Simocetus rayi*, now includes *Olympicetus avitus*, *O. thalassodon* sp. nov., *Olympicetus* sp. 1, and a large unnamed taxon. Of these, *Olympicetus thalassodon* is one

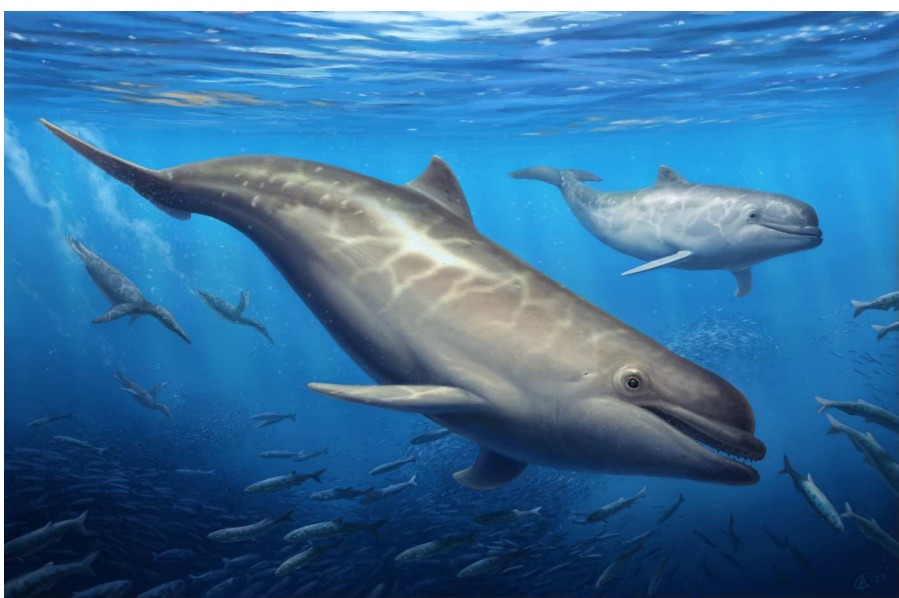

**Figure 22 Reconstruction of *Olympicetus thalassodon* sp. nov.** Life reconstruction of *Olympicetus thalassodon* pursuing a school of fishes alongside plotopterid birds (background) somewhere in the eastern North Pacific Ocean. Art by Cullen Townsend.

of the most completely known simocetids, offering new information on the cranial and dental anatomy of early odontocetes, while the inclusion of CCNHM 1000 within this clade suggest that simocetids may not have had the capabilities for echolocation at least during their earlier ontogenetic stages. This shows that some morphological features that have been correlated with the capacity to echolocate, such as an enlarged attachment area for the maxillonasolabialis muscle, and presence of a premaxillary sac fossae (*Fordyce, 2002*; *Geisler, Colbert & Carew, 2014*), may have appeared before the acquisition of ultrasonic hearing. Furthermore, the dentition of simocetids, as interpreted here, seems to be the most plesiomorphic amongst odontocetes, while other craniodental features within members of this clade suggests various forms of prey acquisition techniques, including raptorial or combined in *Olympicetus* spp., and suction feeding in *Simocetus* (as suggested by *Fordyce, 2002*). Meanwhile, body size estimates for simocetids show that small to moderately large taxa are present in the group, the largest taxon being represented by LACM 124104, with an estimated body length of 3 m. This length places it amongst the largest Oligocene odontocetes, only surpassed in bizygomatic width (and therefore estimated body length) by *Mirocetus riabinini* and *Ankylorhiza tiedemani* (*Riabinin, 1938*; *Boessenecker et al., 2020*; *Sander et al., 2021*). Finally, the new specimens described here add to a growing list of Oligocene marine tetrapods from the North Pacific, further facilitating faunistic comparisons with other contemporaneous and younger assemblages in the region, such as those in Mexico (*e.g.*, El Cien Fm.) and Japan (*e.g.*, Waita Fm.), thus improving our understanding of the evolution of marine faunas in the region.

**Abbreviations**

| | |
|---|---|
| **c** | character state as described and numbered by Sanders and Geisler (2015) and subsequent works, *e.g.*, (c.15[0]) refers to state 0 of character 15 |
| **LACM** | Vertebrate Paleontology Collection, Natural History Museum of Los Angeles County, Los Angeles, CA, USA |
| **KMNH VP** | Kitakyushu Museum of Natural History, Kitakyushu City, Japan |
| **USNM** | Department of Paleobiology, National Museum of Natural History, Smithsonian Institution, Washington, D.C., USA. |

## ACKNOWLEDGEMENTS

I wish to extend my gratitude to E. M. G. Fitzgerald (MV), N. D. Pyenson (USNM), R. E. Fordyce (UO) and M. Viglino (CONICET-CENPAT) for discussions about early odontocete morphology, to J. G. M. Thewissen (NEOMED) for providing cast of the upper teeth of *Indocetus ramani*, to E. M. G. Fitzgerald (MV), N. D. Pyenson (USNM) and D. J. Bohaska (USNM) for access to collections under their care, and also, to James L. Goedert and the late Gail H. Goedert, for collecting and donating the specimens described in this work to the Natural History Museum of Los Angeles County. This manuscript benefited greatly and was improved by the careful and thoughtful reviews of O. Lambert (IRSNB), J. Geisler (NYIT) and an anonymous reviewer. Finally, many thanks to Academic Editors N. D. Pyenson and A. Farke for their help and handling of the manuscript, and to Cullen Townsend for his fantastic reconstruction of *Olympicetus thalassodon* in its environment.

### Funding

The authors received no funding for this work.

### Competing Interests

The authors declare there are no competing interests.

### Author Contributions

- Jorge Velez-Juarbe conceived and designed the experiments, performed the experiments, analyzed the data, prepared figures and/or tables, authored or reviewed drafts of the article, and approved the final draft.

### Data Availability

The raw data is available in the Supplemental Files.

### New Species Registration

The following information was supplied regarding the registration of a newly described species:

Publication LSID: urn:lsid:zoobank.org:pub:D190F6B6-FB67-4F2B-AC24-145DF06D3FD3

*Olympicetus thalassodon*:  urn:lsid:zoobank.org:act:0CEE0B0A-D5C4-4387-A320-F063640B98CC

## Supplemental Information

Supplemental information for this article can be found online at http://dx.doi.org/10.7717/peerj.15576#supplemental-information.

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
