# Peer review of "New heterodont odontocetes from the Oligocene Pysht Formation in Washington State, U.S.A., and a reevaluation of Simocetidae (Cetacea, Odontoceti)"

_PeerJ, doi:10.7717/peerj.15576_

## Round 0.1 · original submission · Minor Revisions

This manuscript received three in-depth and favorable reviews, all of which converged on the same recommendation of minor revisions, which I support. They all agreed that this manuscript is an important contribution to the systematics of early odontocetes, filling in important information for a taxonomic group (Simocetidae) that bears on major questions about the evolution of key morphological complexes in toothed whales. In that way, the description of new fossil taxa is a welcome advance, as the reviewers point out. The bulk of the suggested revisions are minor in scope: for example, Reviewers 1 and 2 (Geisler and Lambert) noted some inconsistencies in anatomical terminology (singular vs. plural usage for individual element descriptions), which are easily addressed; both also noted some improvements that could be made in the reporting of phylogenetic results (i.e., trees). Reviewers 2 and 1 (the latter anonymous) both pointed out that it is worth lightening the touch on the functional interpretations of high-frequency hearing -- as Reviewer 1 pointed out, lacking adult material, it cannot be discounted that this taxon grew ontogenetically into high frequency hearing. Where relevant, note the reviewers' comments in the attached annotated versions of the manuscript. I concur with all reviewers that the illustrations are extensive and thorough, befitting the open format of this contribution in PeerJ. Lastly, it is likely that a revised manuscript will be sent out for further review.

Reviewer 1 ·

Basic reporting

Basic Reporting
The study is well written, presented in clear and unambiguous language, and is both concise but comprehensive. The author does an excellent job of describing key anatomical features across the specimens in question, and providing the necessary comparative context for the information to be relevant to the reader, without bloating the manuscript with needless details. The literature cited is comprehensive and correct. It provides the necessary background and context without consistent of a full literature review that would bloat the introduction. The figures are of excellent quality and enhance the manuscript in every way. The RAW data is accessible and available. The study is entirely self contained, presents a clear hypothesis, and addresses the hypothesis in a way that is supported by the data presented therein.

Experimental design

Experimental Design
This study represents primary research within the Aims and Scope of PeerJ. This manuscript describes several key specimens of fossil, ancestral toothed whales that collectively bear on our understanding of the evolution of toothed whales as a clade. This manuscript bears novel data on a key part of the evolutionary tree: Simocetids, which are a well-known but poorly understood clade. The author clearly lays out the gaps in our knowledge in the introduction, carefully contextualizes all of the descriptive and comparative data with an eye towards these gaps, and the advances our understanding in doing so. By the discussion and the conclusions, it is clear that the novel data has advanced our understanding of the topic. I have no qualms nor concerns about the methodology or the standard of the investigation performed.

Validity of the findings

Validity of the Findings
Overall, the author does an exceptional job of drawing a through line from the gaps in our knowledge to the specimens and novel data and back to the gaps, thereby filling them in. The study is easily replicable by external researchers; the specimens are all publicly available and the data is fully open. The conclusions are well stated and are logical to arrive at given the data presented. I think the author does an excellent job of detailing how the novel data advances our understanding of the geography and feeding ecology of this clade, while avoiding grandiose claims that would be unsupported by the specimens in question.
I do, however, disagree with the author on one key aspect: the nomenclatural acts. I applaud the author for their conservative approach to nomenclature, and acknowledge why they may be hesitant to name taxonomic units based on the less complete specimens. I do not disagree with this hesitancy; but I do disagree with their choice. In particular, naming “Simocetidae gen. et sp. A” and “Olympicetus sp. 1” feel like the wrong approach. If these specimens are, indeed, distinct and diagnostic beyond the existing nomenclature, and they are complete enough to warrant holotype status, then they should be named, for ease of future works. If they are not complete enough to warrant holotype status (and I applaud the author’s conservative approach here are discussed in line 1078), then they should be simply referred to the parent taxonomic unit, but still described.
It feels to me like the awkward middle choice of naming the taxon after a number or letter is borne from an underlying assumption that the specimen is important enough to describe but not complete enough to name. I would challenge this assumption: even if it is not complete enough to name it is valuable to describe because of the valuable data it has brought to this discussion. While the notion of naming after a number or letter is not unprecedented in marine mammals (Homiphoca comes to mind), those examples typically involve multiple specimens being referred to an undiagnosable taxonomic unit. Thus, I do not find it helpful or valuable in this instance. Ultimately, this type of nomenclature is semantic in nature and not providing a scientific value greater than simply referring to the specimens by their specimen number.
Critically, I do not think this should prevent the manuscript from publication. It is an excellent study overall and deserves publication. I simply disagree with this taxonomic approach and would urge the author to consider an alternative.

Additional comments

General Comments
I would like to commend the author for a well constructed, thoughtful, and high quality contribution to the literature. Besides the comments above, I have only a few other, minor comments they will hopefully find helpful:

Line 41: consider “most prolific” instead of best? Best implies a value of quality, not merely superlative abundance.

Line 68-69: I agree with the author here, but will not that the inclusion of CCNHM 1000 may challenge the biogeography. The author should avoid an appearance of “wanting their cake and eating it too” in that including CCNHM is valuable for their commentary on echolocation but excluding it is valuable for their comments on biogeography. Just something to consider.

Line 341: early = stem? Early diverging? Basal branching? Consider clarifying please.

Line 1130: I agree with the author about their interpretation of CCNHM 1000 and how it bears on the clade overall. However, they may want to note that “…it does not have the capacity for ultrasonic hearing…” requires the critical qualifier “as neonates”. It is theoretically possible for Simocetids to “grow in to” their high frequency hearing and, to my knowledge, the hearing capabilities have not been studied in an adult yet.

Line 1222: I strongly agree with the author’s interpretation of the tooth count. Though I applaud the author for taking the conservative approach and including the ? after I3, I think that three incisors is a very safe assumption. Perhaps the author would consider commentary in this regard; a deviation from 3 incisors would be quite non standard for toothed cetaceans. However, I can think of at least three exception: Kentriodon pernix; Wimahl chinookensis; and tusked cetaceans such as Monodon and Odobenocetops.

·

Basic reporting

- The English of all parts of this very fine work is clear and does not request any significant improvement. There is a minor, recurrent problem with the use of the singular and plural in the description of the cranial bones, but this can be easily fixed. I made a series of suggestions in the pdf of the ms, though it may be worth checking the whole text. There are some minor problems with the anatomical terminology (e.g. sagittal crest vs external occipital crest, infraorbital plate vs alveoloar/antorbital process of the maxilla, foramen ovale vs fenestra ovalis) that should be assessed.
- The introduction provides sufficient details on the state of the art, and the reference list is complete. I only made a couple of suggestions in the pdf.
- The article is organized in a logical way. I only suggest moving part of the discussion on the phylogenetic analysis in a short section before the discussion, in a way that the results of this analysis are better separated from their interpretation/discussion (see annotated pdf).
- All the figures are of excellent quality. The photos look great and the associated line drawings are clear and really informative. I only made a series of suggestions for minor corrections in the figure captions and some labels (a few mistakes for the side/orientation of the bones and some possible issues with terminology and abbreviations). Really nothing crucial.
- The .nex version of the matrix for the phylogenetic analysis is provided (though I could not try to re-run the analysis due to computer issues). I think that the complete version of the selected tree (with relationships detailed for all clades) should be provided as supplementary figure (not only in the .nex file), as well as the constraint molecular tree, for completeness.

Experimental design

- This work describes a series of beautifully preserved fossils that are new to science, providing important anatomical details on previously described and new taxa. The inclusion of this important new material in a phylogenetic analysis yields a tree that provides further support for the diagnosis of a poorly know family of ancient odontocetes. The objectives of the study and the gaps in knowledge are adequately explained.
- The description and interpretation of the fossils are done in a rigorous way, using up to date terminology. The phylogenetic analysis is performed following a simple but adequate and clearly explained methodology (though see my question below about a potential issue with the molecular constraint).

Validity of the findings

- The anatomical and phylogenetic results are convincingly supported by the associated data (descriptions and very fine illustrations). I made comments about some anatomical interpretations in the annotated pdf, but most should be easily dealt with.
- I would recommend gathering all the data supporting the ontogenetic stage attribution of each specimen in a short section at the beginning of each description, to facilitate access to that information and to allow for a clearer assessment of potentially ontogenetically-related differences between taxa/specimens.
- Most conclusions are very well supported. I suggest to done down somewhat the conclusions arising from the interpretation of a very young specimen as unable to hear high-frequency sounds. It would indeed be essential to investigate this aspect on fully grown individuals, and until then I would recommend remaining more cautious about subsequent evolutionary conclusions.

Additional comments

I made of series of comments and suggestions in the annotated pdf. Most deal with very minor issues. Other than the requests listed in the three sections above, find below a few, slightly more important issues or suggestions that should be dealt with:
1- There is a mention that a backbone constraint was used for the phylogenetic analysis based on molecular results on extant odontocetes. However, the most parsimonious tree does not show fully resolved relationships between extant families/superfamilies. Therefore, I am not sure that the constraint was enforced. I do not think that this would change much the topology for the stem odontocete part of the tree, but I would strongly recommend making sure that this constraint is applied (and reflected in the final tree).
2- The very detailed diagnoses may be made slightly easier to read if unambiguous synapomorphies were placed in a separate, first section before the other characters. Also, for the diagnosis of O. thalassodon, because you provide a diagnosis for the genus Olympicetus just above, highlighting the main differences and similarities with other simocetids and archaic odontocetes, I would recommend limiting the species diagnosis to differences/similarities with other species of Olympicetus, to avoid any repetition of characters also diagnostic at the genus level. This would make this part more concise, I think.
3- Many teeth are beautifully preserved (and figured) for several specimens. I did not find any mention of tooth wear (or its absence) and other dental damage in the related descriptions. As there is a part of the discussion dealing with feeding techniques/strategies (with mentions of taxa possibly more specialized towards combined or suction feeding, as well as benthic feeding), I would suggest adding some brief comments on this aspect, which can be informative to test for such techniques.
4- Could the Japanese specimens that are discussed quite in detail in the section on North Pacific simocetids benefit from a couple of illustrations? Even schematic line drawings would be useful, as the original publication may be difficult to find.
5- The use of 'middle Oligocene' throughout the text may lead to confusion, as this epoch is only split in two ages. I would suggest using something like 'late early to early late Oligocene' instead.

I am looking forwards to seeing this excellent work published.
O Lambert

·

Basic reporting

• There are some grammatical issues that should be checked. I caught as many as I could and noted them on the review pdf. Most problems relate to bilateral bones of the skull. Sometimes right and left sides of one bone are described in relation to either the right or the left side of another bone (not both), often in the same sentence. This sometimes results in subjective verb agreement issues. I find that sticking to one side is best, and only use both sides when a median element (i.e. premaxillae relative to mesorostral groove) is discussed.
• Figures 9 and 11, P3 vs. P4: The P3 in Fig. 11 is labeled as P4 in Fig. 9. This needs to be fixed, and it may have implications for the inferred tooth count and the identities of various tooth positions as described in the text.
• Figure 11 flipped teeth: The labels for G and I are reversed (the P4 is M2 and the M3 is P4).
• Figure 12G, unlabeled tooth: There is a tooth posterior to the m2 that is not labeled. It is unclear to me if that floated in from another part of the toothrow or is the adjacent upper. It should be labeled.

Experimental design

The caption for figure 20 mentions decay indices and bootstrap support, but these are not shown in the tree. These values should be added to the figure and the analyses used to estimate them described in the methods.

Validity of the findings

• One of the main conclusions of the study is that you have discovered a new species. I would tend to agree with you, but it is not as clear as some of cases; clearly your new taxon is very closely related to O. avitis. I suggest adding a section in discussion going over why this is not Olympicetus avitus and exploring other possibilities (e.g. differences are due to intraspecific variation, ontogeny). The orbits look pretty different in lateral view, so it might be helpful to really go into some detail about that in this section. Some of the differences are in the differential diagnosis, but they are sort of lost in a lot of comparisons.

• Please note whether there is any wear on the teeth and give specifics about the size of the wear and its locations.

Additional comments

This is an excellent study and most of my comments are quite minor. I have one comment that I see as more of a suggestion, rather than a requirement.

You discuss the teeth of Olympicetis avitus and how the holotype of the new species allows the loci of many of the teeth of the holotype of O. avitus to be inferred. While the text is clear enough, a figure showing the teeth of Olympicetus avitus, with their likely tooth positions (in the right order), would be really helpful. You correctly note how archaic these teeth are, and such a figure would increase the impact of the present study.

Jonathan Geisler

---

## Round 0.2 · Minor Revisions

The original Academic Editor is not available so I have taken over handling this submission.

Thank you for your close attention to the comments from the reviewers. One of the previous reviewers has seen the manuscript again, and provided some minor comments and requests for small clarifications. Once those are addressed, I should be able to move the manuscript forward.

·

Basic reporting

no further comments

Experimental design

no further comments

Validity of the findings

no further comments

Additional comments

Many thanks to the author for having taken the time to respond to the comments and suggestions, even for very minor issues. This made the second round of review even easier than the first. Again, I really enjoyed reading this text. I only found a series of typos and a few small formulation problems in the revised text (see annotated .doc version attached). This will be very easily solved, and obviously won't ask for another round of reviews. I am looking forwards to seeing this excellent work published.
O Lambert

PS. I see that I can only upload a pdf of the annotated text, but I would be pleased to send by direct email the .doc version, as some minor corrections may be more difficult to spot in the pdf. Feel free to ask!

---

## Round 0.3 · Minor Revisions

Thank you for your close attention to the comments from the most recent round of revisions. The manuscript is nearly ready for publication. A number of minor grammatical and stylistic edits should be made prior to acceptance; once these are completed, I should be able to return a final decision in short order. All edits are marked on a Word document; I have sent this to the main editorial office to pass along to you.

---

## Round 0.4 · accepted · Accept

Thank you for your close attention to the previous round of edits -- the manuscript is now ready to move forward to publication.